# Contribution of sea ice albedo and insulation effects to Arctic amplification in the EC-Earth Pliocene simulation

Jianqiu Zheng[1,2,3], Qiong Zhang[1], Qiang Li[1], Qiang Zhang[1], Ming Cai[4]

[1]Department of Physical Geography and Bolin Centre for Climate Research, Stockholm University, Stockholm, 10691, Sweden

[2]School of Earth and Space Sciences, University of Science and Technology of China, Hefei, 230026, China

[3]Key Laboratory of Meteorological Disaster of Ministry of Education, Nanjing University of Information Science and Technology, Nanjing, 210044, China

[4]Department of Earth, Ocean and Atmospheric Science, Florida State University, Tallahassee, Florida, 32306, USA

*Correspondence to: Jianqiu Zheng (qiu@ustc.edu.cn)*

**Abstract.** In the present work, we simulate the Pliocene climate with the EC-Earth climate model as an equilibrium state for the current warming climate induced by rising $CO_2$ in the atmosphere. The simulated Pliocene climate shows a strong Arctic amplification featuring pronounced warming sea surface temperature (SST) over the North Atlantic, in particular over Greenland Sea and Baffin Bays, which is comparable with geological SST reconstructions from the Pliocene Research, Interpretation and Synoptic Mapping group (PRISM, Dowsett et al., 2016). To understand the underlying physical processes, the air–sea heat flux variation in response to Arctic sea ice change is quantitatively assessed by a climate feedback and response analysis method (CFRAM) and an approach similar to equilibrium feedback assessment. Given the fact that the maximum SST warming occurs in summer while the maximum surface air temperature warming happens during winter, our analyses show that a dominant ice-albedo effect is the main reason for summer SST warming, and a 1% loss in sea ice concentration could lead to an approximate 1.8 $Wm^{-2}$ increase in shortwave solar radiation into open sea surface. During winter months, the insulation effect induces enhanced turbulent heat flux out of the sea surface due to sea ice melting in previous summer months. This leads to more heat released from the ocean to atmosphere, thus explaining why surface air temperature warming amplification is stronger in winter than in summer.

## 1 Introduction

As shown in the monitoring at Mauna Loa Observatory in Hawaii (https://www.esrl.noaa.gov/gmd/obop/mlo/), the $CO_2$ concentration in the atmosphere passed the 400 ppm threshold by September 2016. Accordingly, global mean temperature in 2016 increased by about 1.1 °C compared to the preindustrial period, as released by the World Meteorological Organization (https://public.wmo.int/en/media/press-release). One major consequence of this continuing and accelerating warming is the rapid melting of ice at high latitudes. The ten lowest minimum Arctic sea ice extents since satellite records were made available in 1979 have happened in the last decade except for 2005, as documented by the National Snow and Ice Data Centre. Moreover, an ice-free Arctic Ocean in September is estimated to emerge in around 2050 on the basis of climate

model projections (Overland et al., 2011). As sea ice retreats, the surface of the Arctic Ocean becomes less reflective and the enhanced open-ocean region leads to greater air–sea heat exchange due to the reduced insulating effect of sea ice. This leads to changes in the surface heat budget and changes in overlying cloud and water vapour, further amplifying Arctic warming

and sea ice melting. Many studies have shown that the accelerated Arctic sea ice retreat possibly results from the local ice-albedo positive feedback (Winton, 2008), meridional heat transport by atmospheric circulation and oceanic current (Alexeev et al., 2013), or sea ice drift out of the Fram Strait (Nghiem et al., 2007; Krumpen et al., 2016). In turn, Arctic sea ice decline can result in a variety of impacts on climate change, such as Arctic amplification (Serreze et al., 2009), change of cloud cover and precipitation (Liu et al., 2012; Bintanja and Selten, 2014), shift in atmospheric circulation pattern (Alexander et

al., 2004), and slow-down of the Atlantic Meridional Overturning Circulation (Sévellec et al., 2017). A detailed consequence of Arctic sea ice decline classified by local and remote effects has been reviewed by Vihma et al. (2014).

Such ongoing high $CO_2$ level and low ice concentration in the Arctic is not unique in Earth's history. Geological data show that during the Pliocene, the $CO_2$ concentration in the atmosphere reached 400 ppm or even more, and extreme warmth and Arctic amplification are recorded in multi-proxy evidence, including the longest and most complete record from Lake

El'gygytgyn, an undisturbed Siberian lake in northeast Arctic Russia (Brigham-Grette et al., 2013). Seasonally ice-free conditions existed in some Arctic regions in the mid-Pliocene until circulation through the Bering Strait reversed, at which point the excess freshwater supply might have facilitated sea ice formation (Matthiessen et al., 2009). Several climate models have simulated the Pliocene but failed to reproduce the strong Arctic amplification shown in geological proxy data (Dowsett et al., 2012). While most of the previous studies on the contributions of the sea ice effect to Arctic amplification focus on

contemporary trends or future projections, here the Pliocene simulation is selected for three reasons: (1) The Pliocene epoch (approximately 3 million years ago), the most recent warm period with $CO_2$ concentrations similar to today, is not only an analogue of future climate change but also an appropriate past time-slice to examine regarding sea ice effects of albedo and insulation (Haywood et al., 2016a). (2) The Pliocene simulation can be partly verified by proxy data reconstructed from deep-sea oxygen isotope analysis (Dowsett et al., 2012), while projecting the future from a climate model is of high

uncertainty owing to the lack of any validation. (3) Whereas the historical or undergoing climate variability is transient, the Pliocene simulation is obtained after the model integration reaches quasi-equilibrium. As inferred from Li et al. (2013), the equilibrium response is in principle reversible, while transient response is hysteretic, suggesting that the Pliocene simulation can better represent a steady climate response.

Two physical characteristics of sea ice are considered to affect the climate system. One is much higher surface reflectivity

of ice than that of open water, and the other is that ice can inhibit or reduce the exchange of momentum, heat, and mass between the atmosphere and ocean. Hereafter we refer to these two effects as "albedo" and "insulation," respectively. Most previous studies on the two effects are mainly carried out by sensitivity experiments with the atmospheric general circulation model (AGCM). For instance, Gildor et al. (2014) examined the role of sea ice in the hydrological cycle using the Community Atmosphere Model (CAM3). The two effects are separated by modifying the sea ice albedo to that of open-

water, or setting the sea ice thickness to zero and keeping albedo unchanged. Their results show that the insulation effect on

the hydrological cycle is larger than the albedo effect, and these two effects are not independent, i.e. their total effect is not the sum of their separate contributions. Lang et al. (2017) also pointed out that the sea ice thinning in recent years can lead to a 37% increase of Arctic amplification through the weakened insulation effect, as estimated by an AGCM. Note that sea surface temperature (SST) is prescribed in their AGCM simulation, while sea ice albedo or thickness is modified. In fact, the modification of sea ice does not closely match the fixed SST, which may lead to a bias in the sea ice effect estimation from the AGCM simulation. The climate system, in turn, reinforces sea ice loss while influenced by albedo or insulation effects, which are known as ice–albedo feedback or ice–insulation feedback. In addition, albedo and insulation interact in a nonlinear way (Gildor et al., 2014). These feedbacks and interactions add more challenges to understanding the effect of sea ice on climate. Recently, Burt et al. (2016) and Kim et al. (2016) addressed the relationship between sea ice loss and air–sea interface heat budget using the Community Earth System Model (CESM) simulation and cyclo-stationary empirical orthogonal function (CSEOF) analysis, respectively. However, the studies contain large uncertainties due to the hysteresis of transient processes (Li et al., 2013). Although the surface heat budget is the most fundamental aspect of air–sea interaction, it is still not clear to what extent heat flux responds to the change of Arctic sea ice. Therefore the present study aims to quantitatively assess the variation of each individual component of air–sea heat flux caused by the decrease of Arctic sea ice albedo and insulation. The analysis is based on the EC-Earth simulation of the Pliocene climate, which represents an analogue for a future climate at equilibrium with modern greenhouse gas levels, and the reference state is a preindustrial equilibrium climate state.

The remainder of the paper is organized as follows. Section 2 describes the EC-Earth model and experimental design, and introduces the climate feedback and response analysis method (CFRAM) as well as the approach to extract the impact of sea ice loss. In Section 3, we present several climate features simulated in the Pliocene experiment. The albedo and insulation effects of sea ice on air–sea interface heat flux are investigated in Sections 4 and 5, respectively, followed by summary and discussion in Section 6.

## 2 Model and method

### 2.1 Model description and experimental design

The model applied in the study is the global coupled climate model EC-Earth (version 3.1, Hazeleger et al., 2012). Its atmospheric component is the Integrated Forecast System (IFS, version cycle 36r4) developed at the European Centre for Medium-Range Weather Forecast (ECMWF), including the land model H-TESSEL (Balsamo et al., 2009). This atmospheric spectral model is run at T159 resolution (roughly 1.125°, approximately 125 km) with 62 vertical levels and coupled to an ocean component based on the Nucleus for European Modelling of the Ocean (NEMO, version 3.3, Madec, 2008) and the Louvain-la-Neuve sea ice Model (LIM, version 3, Vancoppenolle et al. 2009). NEMO was developed at the Institute Pierre Simon Laplace (IPSL) and has a resolution of about 1° and 46 vertical levels. In LIM3, the surface albedo parameterization follows Shine and Henderson-Sellers (1985) with the following values: thick dry snow 0.8, thick melting snow 0.65, thick

frozen bare ice 0.72, thick melting bare ice 0.53, and thin melting ice 0.47. The tuning of bare ice and snow albedo would affect whether the equilibrium ice thickness is reasonable and whether the ice is from a multi-year or seasonal ice zone. The coupling between the atmosphere and ocean/sea ice is through the Ocean Atmosphere Sea ice Soil coupler (OASIS, version 3.0, Valcke, 2006). EC-Earth has been used to examine the Arctic climate for the historical period and future scenarios in CMIP5. An evaluation of EC-Earth for the Arctic shows that the model simulates the 20th century Arctic climate reasonably well. EC-Earth simulated cloud variables with slightly larger cloud fraction and less cloud condensate than ERA-Interim, which led to similar longwave cloud radiative forcing. Moreover, total cloud forcing in EC-Earth is in good agreement with the APP-x satellite estimates (Koenigk et al., 2013). Koenigk et al. (2013) showed that the annual mean surface temperature in the Arctic increases by 12 K in the EC-Earth RCP8.5 scenario simulation, and the most pronounced warming is during autumn and winter in the lower atmosphere. A likely ice-free Arctic is indicated in September around 2040. The enhanced oceanic meridional heat flux into the Arctic (Koenigk et al., 2013) and the enhanced atmospheric northward latent energy transport (Graversen and Burtu, 2016) are suggested as major contributors to future Arctic warming in the EC-Earth simulation. The EC-Earth model has also been applied to understand past climates, such as changes in the Arctic climate (Muschitiello et al., 2015), African monsoons (Pausata et al., 2016; Gaetani et al., 2017), tropical cyclones (Pausata et al., 2017a), and ENSO activity (Pausata et al., 2017b) during the mid-Holocene. In this study we apply the model to the mid-Pliocene climate and focus on the effects of sea ice on Arctic climate change.

Two numerical experiments are performed with EC-Earth to facilitate this study. One is the preindustrial control run with the 1850 $CO_2$ concentration of 284.725 ppm, and the other is the mid-Pliocene warm period (3.264–3.025 Ma) sensitivity experiment in which the atmospheric $CO_2$ concentration is set to 400 ppm. The PRISM remains the only global-scale synoptic reconstruction of the Pliocene (Haywood et al., 2016a), and PRISM data are concentrated on the warm interval (3.264-3.025 Ma). Therefore the time slice (3.264-3.025 Ma) is selected for Pliocene simulation. Though the warm interval actually belongs to the late Pliocene, given that the term "mid-Pliocene warm period" have been frequently used for this period in literature, here we continue using mid-Pliocene for consistency. Following the protocol of the Pliocene Model Intercomparison Project, phase 2 (PlioMIP2, Haywood et al., 2016b), several configurations are modified in the Pliocene simulation: (1) in the Pliocene experiment, all trace gases other than $CO_2$, such as $CH_4$, $N_2O$, and aerosols, are specified as identical to the preindustrial run to account for the absence of proxy data. (2) Orbit forcing, including eccentricity, obliquity, and precession, remains same in the preindustrial run as in the mid-Pliocene warm period, which has a near-modern orbital forcing. (3) Enhanced boundary conditions from the Pliocene Research, Interpretation and Synoptic Mapping group (PRISM, Dowsett et al., 2016), including land–sea mask, topography, bathymetry, and ice-sheet, are applied in the Pliocene experiment. The global distributions of lake, soil, and biome are modified to match the new land–sea mask and ice reconstruction. These two experiments proposed in PlioMIP2 core experiments may assess the dependence of climate sensitivity on the radiative forcing and the boundary conditions. The integrations of the preindustrial control run and the Pliocene experiment are carried out for 500 years, and it takes approximately 300 years for the model to reach equilibrium. From our last 200 years of output in the Pliocene simulation (see Figure S1 in the Supplement), the mean top of the

atmosphere (TOA) net radiation is about –0.5 Wm$^{-2}$ and its trend is near zero. The trend of mean SST is about 0.027 K/century, which fulfils the PMIP4 criterion that the trend of mean SST should be less than 0.05 K/century (Kageyama et al., 2018). In this study, the last 100-year-mean of all variables are used for analysis, and the Pliocene climate anomalies are
135 calculated by subtracting the mean of the preindustrial simulation without trends removal, i.e., the term "anomalies" hereafter is the departure from the preindustrial mean. In the following analysis, the Arctic is defined as the region poleward of 70 °N.

## 2.2 Climate feedback and response analysis method (CFRAM)

Climate system warming in the Pliocene experiment is driven by variation in radiative forcing, which is in turn caused by
140 increased $CO_2$ concentration. In response to temperature change, factors such as surface albedo, cloud, water vapour, and air temperature will adjust and feedback until the climate system reaches equilibrium. The contribution from each factor can be quantitatively evaluated by climate feedback analysis. Traditional climate feedback analysis methods, such as partial radiative perturbation, is based on the TOA radiative budget (Wetherald and Manabe, 1988), while the radiative kernel method can be extended to the surface and remain computationally efficient (Soden and Held, 2006; Pithan and Mauritsen,
2014). However, none of them takes individual physical processes into account, particularly non-radiative processes. CFRAM, proposed by Lu and Cai (2009), overcomes this limitation.

CFRAM contains two parts: one is decomposing the radiative perturbation into individual contributions, including shortwave and longwave components, from $CO_2$, surface albedo, cloud, water vapour, and air temperature:

$$\Delta Q_{rad} = \Delta(S+R)_{CO_2} + \Delta S_{albedo} + \Delta(S+R)_{cloud} + \Delta(S+R)_{WV} + \Delta R_T, \quad (1)$$

where $\Delta Q_{rad}$ is total radiative flux perturbation at the surface (ice and ocean), $\Delta S$ and $\Delta R$ are the net shortwave and longwave radiative perturbations at the surface, respectively, and the subscripts $CO_2$, albedo, cloud, WV, and T represent the partial radiative perturbation due to changes in the $CO_2$ concentration, surface albedo, cloud properties, atmospheric water vapour, and air temperature, respectively. In this study, the perturbation means the difference between the Pliocene run and the preindustrial run. Note that here it is assumed that the interactions among the factors ($CO_2$, surface albedo, cloud, water
vapour, and air temperature) are negligible and the higher order terms of each factor are omitted. The assumption is validated by comparing the total radiative perturbation and the sum of all the partial radiative perturbation terms (Figure S4). The other part is calculating partial temperature perturbations due to individual radiative and non-radiative feedback processes, which is based on total energy balance and derived from the relationship between longwave radiation and temperature change. A more detailed description about CFRAM can be found in Lu and Cai (2009).
CFRAM is a practical diagnostic tool to analyse the role of various forcing and feedback agents and has been used widely in climate change research (e.g. Taylor et al., 2013; Song and Zhang, 2014; Hu et al., 2017). In the present study, total radiative flux perturbation is first calculated from the surface radiative flux difference between the Pliocene sensitivity experiment and the preindustrial control run. Then we apply the first part of CFRAM to compute each partial radiative

perturbation, which is performed by offline calculation using a radiative transfer model (Fu and Liou, 1993). The linear
approximation in Equation (1) should be verified with the output from the radiative transfer model. Finally, the partial
radiative perturbation due to albedo, cloud, and water vapour can be used to evaluate albedo or insulation effects of sea ice.

## 2.3 Approach to extract sea ice effects

As sea ice declines in the Pliocene warming climate, air–sea heat flux varies. However, the variation is not only due to the
impact of sea ice but also determined by other factors, such as atmospheric circulation. Therefore an approach capable of
quantifying the influence of a factor is indispensable for extracting the corresponding contribution of the sea ice effect from
the total heat flux change. To distinguish sea ice's contribution from the other processes, the linkage between sea ice and
heat flux needs to be identified through either temporal correlation or spatial correlation, if the effect of sea ice is assumed to
be linear. A canonical case of the former is equilibrium feedback assessment (EFA), which has been used to quantify the
influence of sea ice on cloud cover (Liu et al., 2012) and the heat flux response to SST (Frankignoul and Kestenare, 2002).

Here we adopt a method similar to EFA, but built on spatial correlation due to the limitation of data and computation. As a
high-temporal–resolution CFRAM calculation, such as 6-hourly or daily, is computationally expensive, monthly data are
used in the analysis. However, the monthly resolution is too coarse to explain the relationship between heat fluxes and sea-
ice concentration by temporal correlations. Therefore, spatial correlations are calculated. This method is used in Hu et al.
(2017) to correct cloud feedback. The response of heat flux to changes in sea ice concentration (SIC) is represented as

$$F(s) = \lambda I(s) + N(s), \quad (2)$$

where $F(s)$ is the heat flux anomaly at location $s$, $I(s)$ is anomalous SIC, $\lambda$ is the response coefficient of heat flux to SIC
change, and $N(s)$ is the climate noise independent of SIC variability. The response coefficient can be calculated as

$$\lambda = \frac{cov[F(s), I(s)]}{cov[I(s), I(s)]}, \quad (3)$$

where $cov[F(s), I(s)]$ is the spatial covariance between heat flux and SIC, and $cov[I(s), I(s)]$ is the spatial variance of
SIC.

The statistical significance of the response coefficient is tested using a two-sided Student's t-test, where the effective
degrees of freedom is estimated from the auto-correlation function (Bretherton et al., 1999) as

$$n = N \frac{1 - r_1 r_2}{1 + r_1 r_2}, \quad (4)$$

where $n$ is the effective degree of freedom, $N$ is the sample size, and $r_1$ is the lag-one auto-correlation of heat flux
(similarly $r_2$ for SIC). Note that auto-correlation of heat flux and SIC is so strong that $r_1$ and $r_2$ can approach 1, leading to a
drastic decrease of effective degree of freedom.

## 3 Mid-Pliocene climate features

Unlike the modern Earth observation system, the Pliocene climate proxy data are reconstructed mainly from the oxygen isotope analysis of deep-sea samples, such as forminifera, diatom, and ostracod assemblages. Several climate features have been revealed with the multi-proxy data (Haywood et al., 2016a). One of the most concerning is Arctic amplification — the warming in surface air temperature (SAT) in the Arctic region tends to be more than twice as warm as that in the low- and mid-latitude regions (Serreze and Barry, 2011). Furthermore, Arctic SAT and SST during the Pliocene is significantly warmer than today, despite comparable $CO_2$ concentrations (Ballantyne et al., 2013). This probably stems from the fact that the present transient process has not yet reached a steady state, or is due to the change of gateways that can affect the Atlantic meridional overturning circulation (AMOC) (Brierley and Fedorov, 2016; Otto-Bliesner et al., 2017; Feng et al., 2017).

In Figure 1, we show the annual mean warming and seasonal warming averaged over the Arctic Ocean for SST and SAT between the Pliocene and preindustrial simulations. The shaded circles in the SST change distribution (Figure 1a) represent the mean annual SST anomalies at 95% confidence-assessed marine sites from the Deep Sea Drilling Project and Ocean Drilling Program, which are available in the supplementary table of Dowsett et al. (2012). The overlay of proxy data over the filled contour maps does not show the difference well, so the difference of annual mean SST anomaly between EC-Earth simulation and the proxy data is shown in Figure S2. In contrast to the large underestimation of multi-model ensembles regarding the warming over the northern Atlantic sector of the Arctic Ocean (Dowsett et al., 2012), the warming amplitude and pattern in EC-Earth simulation is comparable with the high-confidence proxy data. This is consistent with the result of Koenigk et al. (2013), which suggests that the sea ice change in the EC-Earth is strong and that the EC-Earth simulations show a strong Arctic amplification compared to most CMIP3 models. According to Figure 1b, the Pliocene SAT north of 70 °N is as much as 10–18 °C higher than the preindustrial period, similar to the mid-Pliocene paleoclimate estimate by Robinson et al. (2008).

Figures 1a and 1b also show that the SST and SAT anomaly patterns are somewhat similar over low- and mid-latitude regions, different from over high-latitude regions, particularly over the Arctic Ocean, which was previously illustrated by Hill et al. (2014). This disparity results from the intense air–sea coupling over tropical and subtropical oceans, while the air–sea interaction is relatively weak over the Arctic Ocean owing to the albedo and insulation effects of sea ice. Notably, SST warming averaged over the Arctic Ocean shows a distinct seasonal evolution from that of SAT; the maximum warming in SST occurs in summer, while the maximum warming in SAT happens during winter (Figures 1c and 1d). Over the ice-covered regions, the SST is close to -1.8 °C in winter while the SAT is close to 0 °C in summer, which can explain the small SST change in winter as well as the small SAT change in summer. During the preindustrial period, the annual mean sea ice appears to cover the whole Arctic Ocean except for the Greenland Sea, the Norwegian Sea, and the Barents Sea, and it retreats to the western Arctic Ocean in the Pliocene, leading to a significant decrease of sea ice extent over the Fram Basin and Baffin Bay (Figures 2a–c). Consequently, the net heat exchange at the surface of ice or ocean varies greatly (Figure 2d–

f). The net heat flux and other flux terms mentioned hereafter are defined as positive downward. A positive value means that the ocean gains heat from the atmosphere and a negative value means oceanic heat loss. The net heat flux over the sea ice–covered area clearly shows net heat loss during both the preindustrial period and the Pliocene (Figures 2d and 2e). Thus, it can be expected that net heat gain will occur when the sea ice declines. However, the Fram Basin and Baffin Bay display pronounced heat loss, which might be linked to the disappearance of sea ice in the Pliocene (Figure 2b).

The net heat flux at the surface of ice or ocean can be represented as the sum of four terms: the net shortwave radiative flux, the net longwave radiative flux, the turbulent sensible heat flux, and the turbulent latent heat flux. Figure 3 compares the annual mean of the four flux terms to further illustrate the possible relationship between sea ice and net heat exchange (Figures 2c and 2f). The radiative and turbulent heat flux anomalies both are positive over the Chukchi Sea, indicating a marked net heat gain emerging there. Over the Beaufort Sea and East Siberian Sea, the positive change in the net shortwave

radiation anomalies are dominant over the other three negative components, yielding net heat gain. In contrast, the positive net shortwave radiation anomalies over the Fram Basin, the Greenland Sea, and Baffin Bay are less than the sum of net longwave radiation and turbulent heat flux anomalies, thus leading to net heat loss. The negative turbulent heat flux anomalies over Fram Basin, the Greenland Sea, and Baffin Bay are prominent, indicating the sea ice effect on turbulent heat flux anomalies in light of the transition to ice-covered or ice-free states, respectively. Note that the partition threshold of ice-

free and ice-covered conditions is 15% SIC, i.e., a grid point with an SIC of less than 15% is considered ice-free. In Figure 2c, the diagonal stripe represents the region with the transition from ice-covered to ice-free condition, and the diagonal crosshatch represents the region that retains its ice-covered status as the simulation shifts from the preindustrial period to the Pliocene. Only ice-covered regions are examined, as there appears to be large surface heat flux changes in regions that contain no sea ice in both periods, which could be contaminating the statistical relationships between sea ice and the

associated surface flux changes.

## 4 Albedo effect of sea ice

Arctic amplification has been demonstrated by significant SAT anomalies in the foregoing Pliocene simulation. Similar to the process-based decomposition of a climate difference in Hu et al. (2017), the SAT anomalies in the Pliocene simulation as compared to the preindustrial simulation can be thought of as the combination of partial temperature perturbations due to

radiative feedbacks (surface albedo, cloud, water vapour, and air temperature) and non-radiative feedbacks (surface sensible and latent heat fluxes, dynamical advection, ocean processes, etc.). That is to say, the albedo effect of sea ice and snow can be quantified by climate feedback analysis such as CFRAM. Surface albedo is defined as the proportion of the incident solar shortwave radiation that is reflected by the surface, therefore indicating that the albedo effect is relevant to net shortwave radiation rather than net longwave radiation and turbulent heat fluxes.

The annual mean net shortwave radiation change due to sea ice and snow albedo derived from CFRAM is presented in Figure 4. The largest net shortwave radiation change exceeding 50 Wm$^{-2}$ takes place over Fram Basin and Baffin Bay, and most of the Arctic Ocean, except for part of the North Atlantic and the Barents Sea, shows net shortwave radiative heat gain.

Compared with the SIC change (Figure 2c), the increase of annual mean net shortwave radiation absorbed by the ocean is in accordance with sea ice retreat, which can be clearly depicted in a scatter plot (Figure 5). The effective degrees of freedom is calculated from Formula (4) for testing statistical significance, and the correlation coefficient (r = –0.84) is significant at a 99% confidence level. This indicates that changes in sea ice extent can explain the approximate 71% (square of correlation coefficient) variance of total shortwave radiation change due to albedo, and the residual variance may be caused by changes in snow cover and sea ice/snow state as well as thickness. The statistically significant response coefficient calculated according to formula (3) is –43.0 $Wm^{-2}$, indicating that a 1% decrease in annual mean SIC leads to an approximate 0.43 $Wm^{-2}$ increase in net shortwave radiative heat flux at the surface.

As SIC and incoming solar radiation in the polar region vary with season, we examine the response of net shortwave radiation to sea ice change for every month. As shown in Figure 6, the response coefficient of net shortwave flux to the albedo effect of sea ice displays a seasonal variation, peaking in June with a maximum absolute value of 178.3 $Wm^{-2}$ (approximate 1.8 $Wm^{-2}$ increase in net shortwave radiation due to 1% decrease in SIC). The prominent oceanic heating in May and June seems consistent with the maximum SST warming in August, as the response of seawater lags about 2 months behind due to the great heat inertia and heat capacity of seawater (Venegas et al., 1997; Zheng et al., 2014). Even though Arctic sea ice itself has a great variability owing to melting and freezing processes, SIC anomalies do not exhibit a large variability in different seasons, ranging from 0.19 to 0.26 as shown in the standard deviation of SIC (Table 1). However, the standard deviation of net shortwave radiation anomalies (with respect to monthly mean) associated with the albedo effect varies from 52.45 $Wm^{-2}$ in May to 0 $Wm^{-2}$ in December, when the polar night occurs without any sunlight. Moreover, our correlation analysis indicates that sea ice has a statistically significant impact on surface shortwave radiation, except in November, December, and January, when there is low incident solar shortwave radiation during the Arctic winter. Overall, the seasonality of sea ice's albedo effect on surface shortwave radiation is attributed primarily to the seasonal cycle of net shortwave radiation, and the contribution of SIC variation is substantially small.

**5 Insulation effect of sea ice**

**5.1 Insulation effect of sea ice on surface radiation**

The insulating effect of sea ice has an indirect effect on the net surface shortwave and longwave fluxes. By separating the overlying atmosphere from the ocean, sea ice reduces evaporation from the ocean, resulting in a decrease in water vapour and cloud cover. This reduction plays a non-negligible role in the amount of downward shortwave and longwave radiation reaching the surface. However, remote moisture transport also affects water vapour and cloud amount. Thus, in order to address the insulation effect of sea ice, two steps have to be performed. First, we obtain the total influence of water and cloud on surface radiation by CFRAM. Second, we need to extract the contribution from a local source associated with sea ice.

Figure 7 shows the annual mean cloud feedback and water vapour feedback on net shortwave and longwave radiation, respectively, before removing the remote effects on clouds and water vapour. Even though an increase in cloud cover is expected with the diminishing Arctic sea ice (Liu et al., 2012), whether the increased cloud cover will heat or cool the surface depends on the cloud characteristics. The cloud feedback on shortwave radiation is nearly out of phase with that on longwave radiation, except in the Beaufort Sea and the East Siberian Sea (Figure 7a, 7b). The significant decrease of low cloud cover in the North Atlantic (Figure S3a) may enhance incoming shortwave radiation and weaken downwelling longwave radiation, thus contributing to the positive anomaly in shortwave radiation and negative anomaly in longwave radiation in the North Atlantic. Similarly, the increase of high cloud cover east and north of Greenland (Figure S3b) is responsible for the positive anomaly in longwave radiation over the related areas. In contrast, water vapour feedback tends to simultaneously cool and heat the surface by absorbing solar radiation and downwelling longwave radiation, respectively; the heating is one order of magnitude higher than the cooling (Figure 7c, 7d).

The approach to extract the local insulation effect due to changes in sea ice concentration is based on the premise that the insulation effect on surface radiation is linear with SIC. Like the steps performed to isolate the albedo effect, the response coefficient of shortwave and longwave radiation due to cloud and water vapour for annual mean and seasonal evolution can be calculated respectively, and the results are shown in Figure 8. In the annual mean, the main contributor comes from cloud feedback on longwave radiation ($-11.1$ Wm$^{-2}$), and the cloud feedback on shortwave radiation and water vapour feedback on longwave radiation are similar in magnitude, but opposite in sign. In addition, the annual mean absorption of incoming solar radiation by water vapour is negligible, and this is true for the individual months as well. The absorption and reflection of shortwave radiation by cloud shows a pronounced seasonal cycle, with a large effect in August. However, there is no statistically significant relationship between cloud feedback on shortwave radiation and SIC (Table 2). Compared to the seasonal variation of standard deviation of the net shortwave radiation anomalies, standard deviation of the net longwave radiation anomalies caused by cloud and water vapour associated with local SIC anomalies both show smaller seasonal variation, therefore leading to a relatively constant contribution of sea ice insulation to surface longwave radiation, except in summer months when there is a lack of linear relationship between SIC and longwave radiation (Table 2). Note that the longwave cloud forcing in September ($-17.6$ Wm$^{-2}$) is quite large relative to all the other months, which might result from the maximum cloud cover over the Arctic, as well as the fact that the linear relationship between sea ice concentration and longwave radiation changes due to cloud is strongest in September.

**5.2 Insulation effect of sea ice on turbulent heat fluxes**

Air–sea turbulent heat fluxes, including sensible and latent heat fluxes, have been widely studied with the bulk aerodynamic formula, which specifies that the turbulent heat fluxes are dependent on surface wind speed, sea surface and air temperature difference, specific humidity difference, and the bulk heat transfer coefficients. However, due to the existence of sea ice, the Arctic turbulent heat fluxes show distinctive features from ice-free conditions, which has been mentioned in Section 3. It is therefore essential to take the insulation effect of sea ice into account and differentiate fluxes from ice-covered versus ice-

free areas. This is demonstrated in Figure 9, which displays the Pliocene anomalies in annual mean sensible and latent heat fluxes as a function of SIC anomalies. There is a larger spread in the turbulent heat flux anomalies over the ice-free areas (grey symbols, corresponding to the diagonal hatched region in Figure 2c) than in anomalies from the ice-covered areas
(light blue symbols, cross-hatched region in Figure 2c) because the former is free from the constraint of sea ice. The constraint of sea ice can be apparently captured through the scatter plot of turbulent heat flux and changes in SIC (light blue symbols). For the ice-covered areas, SIC can explain approximate 59% and 74% (square of correlation coefficient) of the variance in the sensible heat flux and latent heat flux, respectively.

The linear regressions of sensible and latent heat flux anomalies on SIC are similar, but not exactly the same. The
response coefficient of sensible heat flux ($35.3\ \mathrm{Wm^{-2}}$) to SIC is larger than that of latent heat flux ($27.7\ \mathrm{Wm^{-2}}$) for the ice-covered areas, which means that the sensible heat flux is more sensitive to SIC change than the latent heat flux. Notably, this is different from the turbulent heat flux variability over low- and mid-latitude regions, where the variability of sensible heat flux is significantly less than that of latent heat flux, such as the trend of turbulent heat flux over the low- and mid-latitude North Pacific and North Atlantic oceans from 1984–2004 (Li et al., 2011). Following Clausius–Clapeyron equation, in the
colder Arctic, the saturated specific humidity is much smaller, resulting in the smaller latent heat flux and response. The positive intercept on the turbulent flux anomaly axis implies more heat gain at the sea surface, even without SIC change. Because the large specific heat capacity of seawater leads to less warming of the ocean than of the atmosphere, the sea surface and air temperature difference (the specific humidity difference) decreases during the cold season when the turbulent heat transport is the most pronounced, consequently resulting in a lower annual heat loss from the ocean to the atmosphere.

Figure 10 shows the seasonal response coefficient of the sensible and latent heat fluxes to the SIC. It appears that the turbulent heat fluxes have a similar seasonal evolution, peaking in November and showing a negative response in July. Therefore, the prompt atmospheric response to turbulent heating is an important contributing factor to the maximum SAT warming that occurs in November. The melting of sea ice due to warming by high levels of $CO_2$ can attenuate the insulation effect and result in more heat transfer through the processes of convection, conduction and evaporation from the ocean to the
atmosphere when SST is higher than SAT; therefore, the turbulent heat fluxes correlate positively with SIC in all seasons except summer (Table 3). If SAT is higher than SST (for instance, in July), sea ice will inhibit the heat transfer from the atmosphere to ocean; thus, the negative correlation emerges. However, the correlations between the turbulent heat fluxes and SIC in summer are not statistically significant (Table 3), indicating other factors rather than sea ice might be dominant.

**6 Summary and discussion**

Arctic amplification in the Pliocene has previously been addressed from reconstructed data (e.g. Robinson et al., 2008; Brigham-Grette et al., 2013); however these data tell only part of the story because of a scarcity of data sites. A model may be applied to investigate mechanisms and processes that help understanding. In contrast to the underestimation of multi-model ensembles documented in Dowsett et al. (2012), the EC-Earth Pliocene simulation can better display some characteristics that have been revealed by the paleoclimate proxy data from deep-sea oxygen isotope analysis. Thus the EC-

Earth coupled model is used in the present work to simulate the Pliocene climate and study the contribution of sea ice albedo and insulation to Arctic amplification.

Air–sea heat flux variation in response to Arctic sea ice change is quantitatively assessed by CFRAM and an EFA-like method in order to reveal important features of Arctic amplification. Table 4 summarizes the results presented in Sections 4 and 5, which separately illustrate the effects of changes in albedo and insulation of sea ice on surface heat exchange. Annual

mean and seasonal evolution of effects are both considered. These allow us to partly interpret the mechanisms of Arctic amplification because the results are merely the contribution from sea ice change. A complete energy budget, including dynamical and thermodynamical processes, is required to understand Arctic amplification comprehensively.

The Pliocene Arctic amplification compared to the preindustrial simulation represents a maximum SST warming in August and a maximum SAT warming in November, which might be associated with the albedo and insulation effects of sea

ice. Albedo only regulates the shortwave radiation, and its effect is primarily determined by the annual cycle of insolation. As sea ice melts starting in early spring, the enhanced insolation through open sea surface makes the ocean warmer, with the most pronounced heating anomalies in May and June. Because of the great heat inertia and heat capacity of seawater, the SST anomaly peaks in August. As a result of the albedo effect of sea ice, ocean heat content increases and more heat is stored in the upper ocean, which is the potential for the later enhanced heat release from ocean to atmosphere. The insulation

effect of sea ice can modulate shortwave and longwave radiation anomalies indirectly through cloud and water vapour as well as directly modulate sensible and latent heat flux anomalies, since sea ice serves as a barrier. Averaged over the year, the absorption of longwave radiation due to the insulation effect is about 4 times stronger than the reflected shortwave radiation by cloud, while the contribution of water vapour to shortwave radiation is almost negligible. The longwave radiation anomalies in response to cloud and water vapour is attributed to downwelling longwave radiation, as upwelling

longwave radiation depends solely on the surface temperature according to the Stefan–Boltzmann law, and its seasonal variation is relatively small compared to the significant seasonality in shortwave radiation. The Pliocene sea ice decline, as compared to the preindustrial period, amplifies the turbulent exchange between the ocean and atmosphere, and the annual sum of sensible and latent heat flux anomalies exceeds radiation flux anomalies. In particular, heat is released to the atmosphere by the prominent enhanced turbulent heat flux anomalies in November, contributing to the formation of the

maximum SAT anomaly in November.

A synthesis of Arctic amplification given by Serreze and Barry (2011) has introduced some of the physical processes mentioned above, including sea ice loss, albedo feedback, cloud cover, and water vapour. Unlike Serreze and Barry (2011), in this work we apply CFRAM and an EFA-like method to untangle these physical processes and obtain a quantitative understanding of sea-ice effects, which would help to directly evaluate the impact on heat exchange once the sea-ice

concentration variation within the Arctic is given. The EC-Earth simulation shows a stronger Arctic amplification than multi-model ensembles (Dowsett et al., 2012). However, an underestimation of Arctic warming as compared to proxy data remains in the EC-Earth simulation, implying less warmth produced by the EC-Earth model from oceanic heat transport,

which yields a clue for improving the simulation. Furthermore, caution should be exercised when discussing sea-ice effects on heat flux, as underestimating Arctic warming might affect the interface heat exchange.

Though significant albedo and insulation effects of sea ice have been studied, the possible nonlinear response of heat flux to sea ice can not be captured in this work. In addition, this approach to extracting sea ice effects is based on spatial correlation; whether the corresponding conclusion is consistent with that from EFA remains uncertain. The consistency check is computationally expensive for CFRAM calculation, as EFA requires high temporal resolution. The present study is based on the Pliocene simulation with the EC-Earth, and the results may be model-dependent. Further work is needed to

compare our results with other PlioMIP models.

*Acknowledgements.* This work was supported by the Swedish Research Council VR for the Swedish–French project GIWA, the China Scholarship Council (Grant 201606345010), and the Opening Project of Key Laboratory of Meteorological Disaster of Ministry of Education of Nanjing University of Information Science and Technology (Grant KLME1401). The

EC-Earth mid-Pliocene simulation was performed at ECMWF's computing and archive facilities, and the analysis was performed on resources provided by the Swedish National Infrastructure for Computing (SNIC) at Linköping University.

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

**Table 1.** The spatial standard deviation of SIC anomalies $\sigma_{SIC}$ and net shortwave radiation anomalies due to the albedo effect $\sigma_{SW\text{-}albedo}$ (Wm$^{-2}$) over the Arctic Ocean. $r_{SW\text{-}albedo}$ is the correlation coefficient between SIC and shortwave radiation anomalies. Those significant at a 99% confidence level are bolded.

| | Jan | Feb | Mar | Apr | May | Jun | Jul | Aug | Sep | Oct | Nov | Dec |
|---|---|---|---|---|---|---|---|---|---|---|---|---|
| $\sigma_{SIC}$ | 0.25 | 0.26 | 0.26 | 0.26 | 0.25 | 0.23 | 0.23 | 0.20 | 0.19 | 0.25 | 0.25 | 0.25 |
| $\sigma_{SW\text{-}albedo}$ | 0.01 | 0.75 | 5.81 | 25.34 | 52.45 | 48.79 | 26.28 | 12.39 | 6.85 | 2.16 | 0.21 | 0 |
| $r_{SW\text{-}albedo}$ | –0.22 | –0.37 | **–0.63** | **–0.77** | **–0.80** | **–0.85** | **–0.85** | **–0.83** | **–0.57** | **–0.53** | –0.11 | / |

**Table 2.** The spatial standard deviation of shortwave and longwave radiation anomalies due to cloud change ($\sigma_{SW\text{-}cloud}$, $\sigma_{LW\text{-}cloud}$) (Wm$^{-2}$) and water vapour change ($\sigma_{SW\text{-}WV}$, $\sigma_{LW\text{-}WV}$) (Wm$^{-2}$) over the Arctic Ocean. $r_{SW\text{-}cloud}$, $r_{LW\text{-}cloud}$, $r_{SW\text{-}WV}$, and $r_{LW\text{-}WV}$ are correlation coefficients between SIC and shortwave and longwave radiation anomalies due to cloud and water change, respectively. Those significant at a 99% confidence level are bolded. Here, the cloud and water vapour change is specified as the part caused by sea ice decrease.

| | Annual | Jan | Feb | Mar | Apr | May | Jun | Jul | Aug | Sep | Oct | Nov | Dec |
|---|---|---|---|---|---|---|---|---|---|---|---|---|---|
| $\sigma_{SW\text{-}cloud}$ | 4.76 | 0.01 | 0.16 | 1.11 | 3.86 | 5.97 | 11.71 | 19.61 | 13.86 | 3.21 | 0.50 | 0.04 | 0 |
| $r_{SW\text{-}cloud}$ | 0.18 | 0.14 | 0.22 | 0.36 | 0.36 | 0.16 | 0.01 | 0.05 | 0.24 | 0.26 | 0.25 | 0.32 | / |
| $\sigma_{LW\text{-}cloud}$ | 8.02 | 9.13 | 9.29 | 8.25 | 7.64 | 10.20 | 11.91 | 15.11 | 13.56 | 11.96 | 10.01 | 10.18 | 9.86 |
| $r_{LW\text{-}cloud}$ | **–0.46** | **–0.59** | **–0.56** | **–0.56** | **–0.51** | –0.36 | 0.06 | 0.04 | –0.23 | **–0.54** | **–0.41** | **–0.60** | **–0.56** |
| $\sigma_{SW\text{-}WV}$ | 0.29 | 0.001 | 0.03 | 0.14 | 0.40 | 0.59 | 0.85 | 0.85 | 0.63 | 0.33 | 0.09 | 0.01 | 0 |
| $r_{SW\text{-}WV}$ | –0.02 | –0.05 | 0.02 | 0.06 | 0.05 | 0.02 | 0.11 | –0.07 | **–0.57** | **–0.62** | **–0.43** | –0.22 | / |
| $\sigma_{LW\text{-}WV}$ | 2.27 | 3.45 | 3.53 | 3.11 | 2.84 | 2.57 | 2.72 | 2.15 | 1.73 | 1.77 | 2.31 | 2.89 | 3.54 |
| $r_{LW\text{-}WV}$ | **–0.56** | **–0.45** | **–0.43** | **–0.50** | **–0.58** | **–0.57** | **–0.46** | –0.13 | 0.38 | 0.13 | –0.36 | **–0.58** | **–0.49** |

**Table 3.** The spatial standard deviation of sensible and latent heat flux anomalies $\sigma_{SH}$, $\sigma_{LH}$ (Wm$^{-2}$) over the Arctic Ocean. $r_{SH}$ and $r_{LH}$ are correlation coefficients between SIC and sensible and latent heat flux anomalies, respectively. Those significant at a 99% confidence level are bolded.

|  | Jan | Feb | Mar | Apr | May | Jun | Jul | Aug | Sep | Oct | Nov | Dec |
|---|---|---|---|---|---|---|---|---|---|---|---|---|
| $\sigma_{SH}$ | 28.53 | 29.44 | 21.64 | 12.87 | 7.94 | 9.46 | 9.55 | 2.63 | 2.11 | 7.02 | 31.11 | 26.80 |
| $r_{SH}$ | **0.57** | **0.64** | **0.67** | **0.66** | **0.76** | 0.26 | −0.36 | 0.03 | **0.65** | **0.80** | **0.71** | **0.56** |
| $\sigma_{LH}$ | 18.70 | 19.00 | 14.75 | 9.46 | 5.64 | 5.84 | 8.75 | 1.93 | 1.69 | 5.77 | 19.87 | 17.44 |
| $r_{LH}$ | **0.74** | **0.77** | **0.78** | **0.76** | **0.71** | 0.14 | **−0.42** | −0.37 | **0.69** | **0.90** | **0.79** | **0.72** |

**Table 4.** The response coefficients (Wm$^{-2}$) of radiation and turbulent heat fluxes to the albedo and insulation effects of sea ice. Those significant at a 99% confidence level are bolded.

| $\lambda$ (Wm$^{-2}$) | flux | | Ann | Jan | Feb | Mar | Apr | May | Jun | Jul | Aug | Sep | Oct | Nov | Dec |
|---|---|---|---|---|---|---|---|---|---|---|---|---|---|---|---|
| albedo | SW | | **−43.0** | 0.0 | −1.1 | **−13.8** | **−75.0** | **−169.2** | **−178.3** | **−97.0** | **−52.0** | **−20.2** | **−4.5** | −0.1 | 0 |
| insulation | SW | cloud | 2.6 | 0.0 | 0.1 | 0.9 | 3.1 | 2.3 | 0.4 | 3.1 | 9.6 | 2.3 | 0.4 | 0.0 | 0 |
|  |  | WV | −0.0 | 0.0 | 0.0 | 0.0 | 0.0 | 0.0 | 0.2 | −0.2 | **−1.0** | **−0.5** | −0.1 | 0.0 | 0 |
|  | LW | cloud | **−11.1** | **−12.1** | **−11.7** | **−10.4** | **−8.9** | −8.6 | 1.7 | 1.9 | −9.0 | **−17.6** | **−11.6** | **−15.8** | **−13.0** |
|  |  | WV | **−3.9** | **−3.5** | **−3.4** | **−3.5** | **−3.7** | **−3.4** | **−3.2** | −0.9 | 1.9 | 0.6 | −2.3 | **−4.4** | **−4.1** |
|  | SH | | **35.3** | **53.4** | **59.0** | **46.4** | **29.6** | **24.2** | 10.4 | −13.8 | 0.4 | **7.1** | **22.3** | **79.2** | **54.0** |
|  | LH | | **27.7** | **45.3** | **46.0** | **36.6** | **25.0** | **16.1** | 3.5 | **−15.0** | −3.6 | **6.0** | **20.5** | **56.7** | **45.7** |

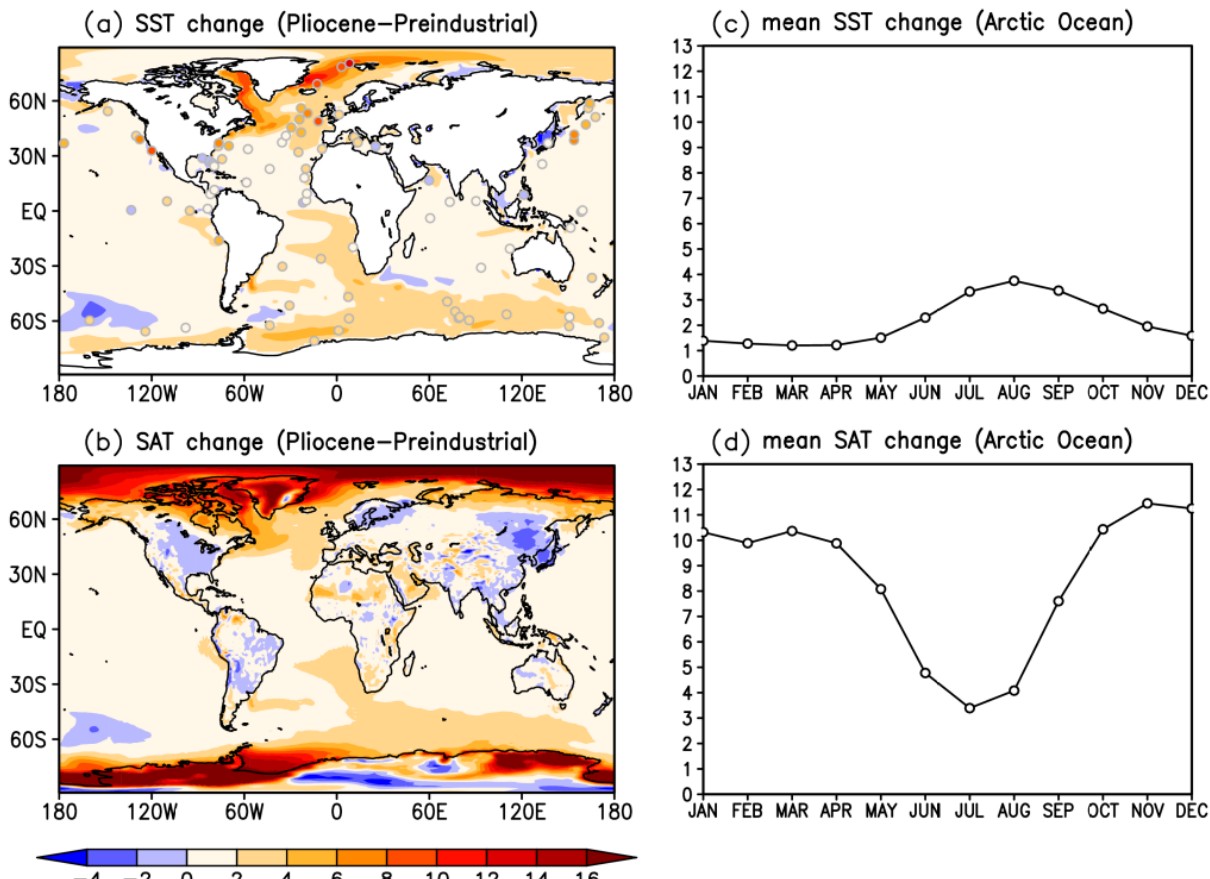

**Figure 1.** The annual mean warming (K) for (a) sea surface temperature (SST) and (b) surface air temperature (SAT), and seasonal warming (K) averaged over the Arctic Ocean for (c) SST and (d) SAT between the Pliocene and preindustrial simulations. The shaded circles in (a) represent the annual mean SST anomalies at 95% confidence-assessed marine sites from the Deep Sea Drilling Project (DSDP) and Ocean Drilling Program (ODP).



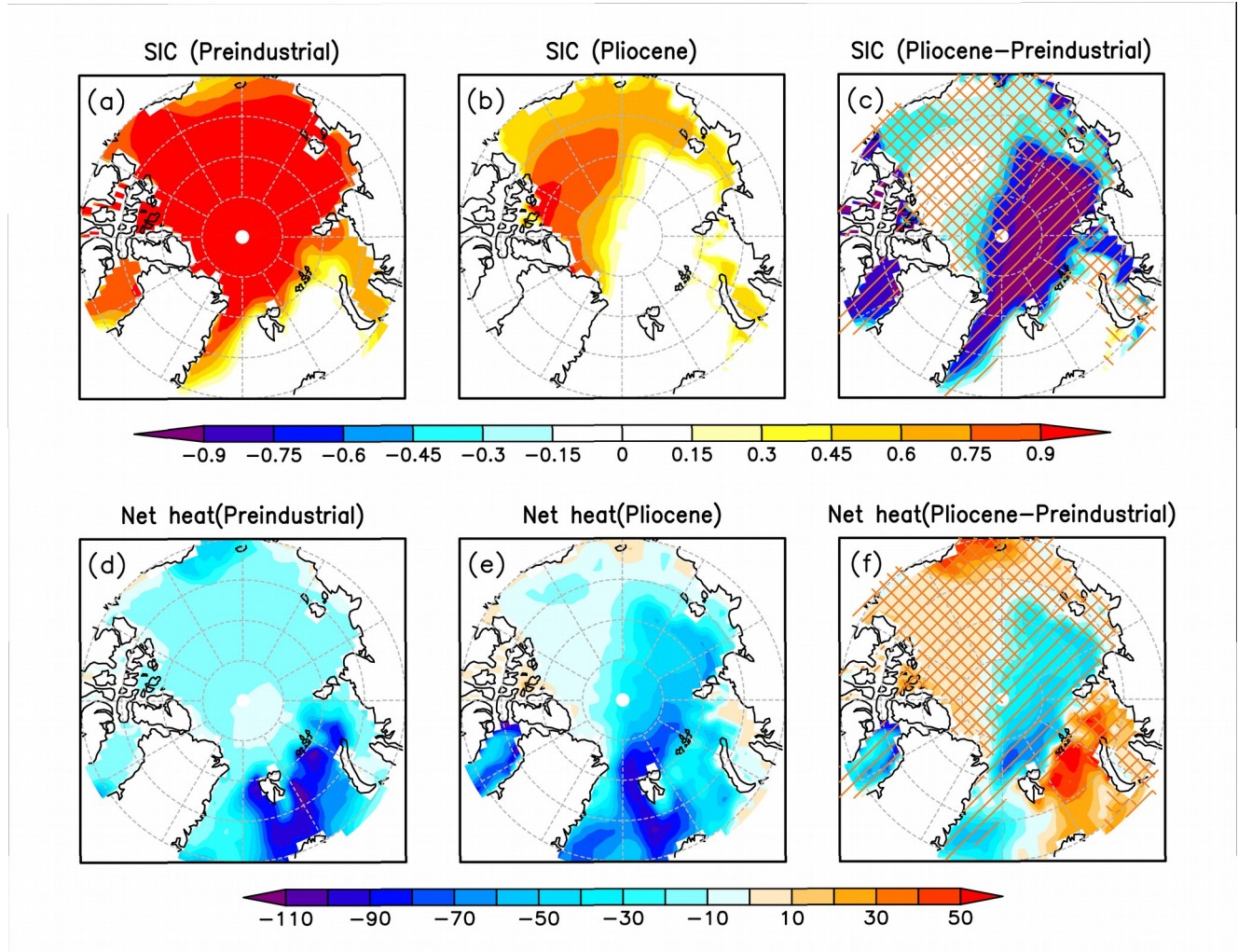

**Figure 2.** Spatial distributions of the annual mean sea ice concentration (SIC) and net heat flux at the surface of ice and ocean (Wm$^{-2}$, positive downward) over the Arctic Ocean. (a) SIC in the preindustrial period, (b) SIC in the Pliocene, (c) the Pliocene SIC change with respect to the preindustrial period, (d) net heat flux in the preindustrial period, (e) net heat flux in the Pliocene, and (f) the Pliocene net heat flux change with respect to the preindustrial period. The diagonal stripe in (c) and (f) represents the regions from ice-covered to ice-free, and the diagonal crosshatch represents the regions from ice-covered to ice-covered.


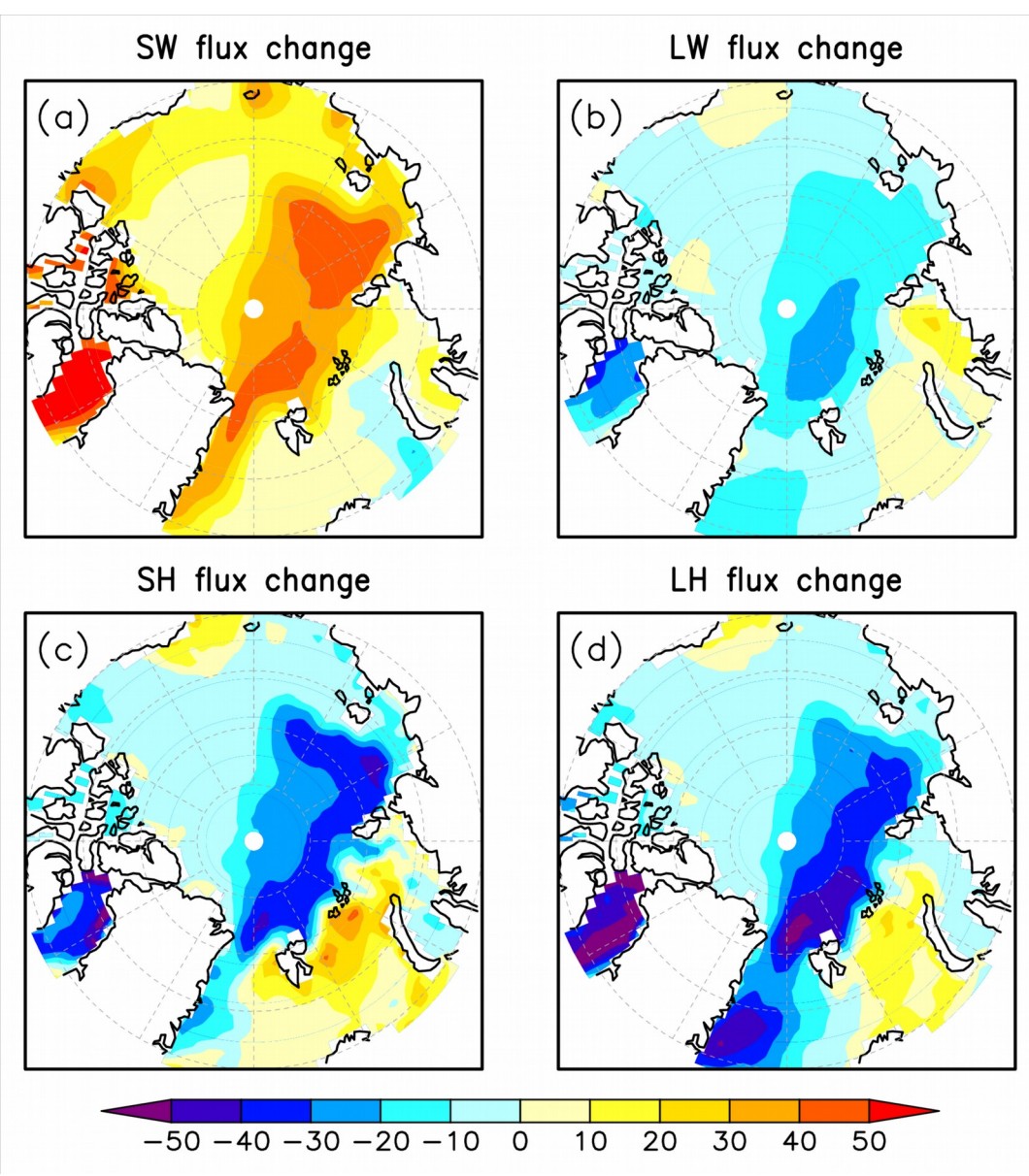

**Figure 3.** Spatial distributions of the Pliocene annual mean heat flux change (Wm$^{-2}$, positive downward) with respect to the preindustrial period over the Arctic Ocean. (a) net shortwave flux, (b) net longwave radiation flux, (c) sensible heat flux, and (d) latent heat flux.

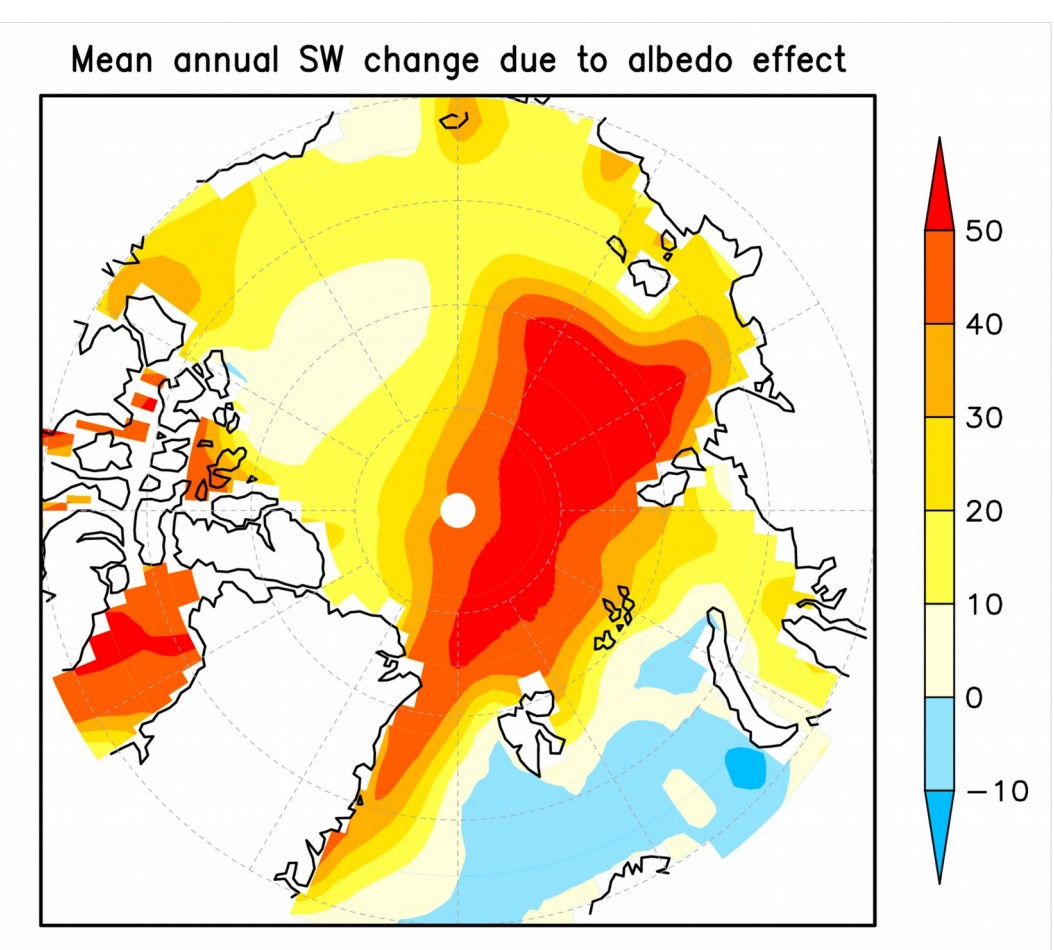

Mean annual SW change due to albedo effect

**Figure 4.** Spatial distributions of the Pliocene annual mean net shortwave flux change (Wm$^{-2}$, positive downward) at the surface over the Arctic Ocean caused by albedo effect of sea ice change with respect to the preindustrial period.


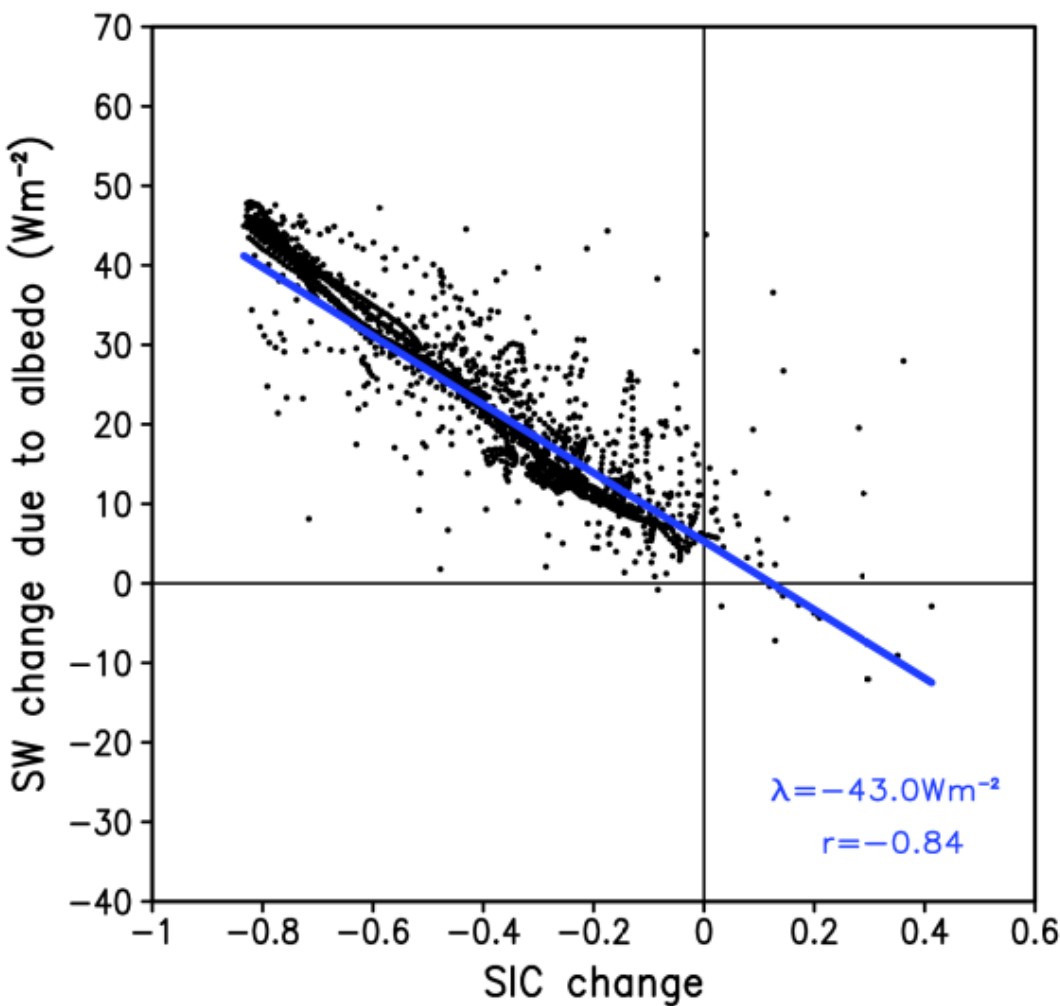

**Figure 5.** The annual mean net shortwave flux change (Wm⁻², positive downward) caused by the albedo effect of sea ice change averaged over the Arctic Ocean as a function of SIC change. All the change is with respect to the preindustrial period, and each dot represents one grid point value over the Arctic Ocean.


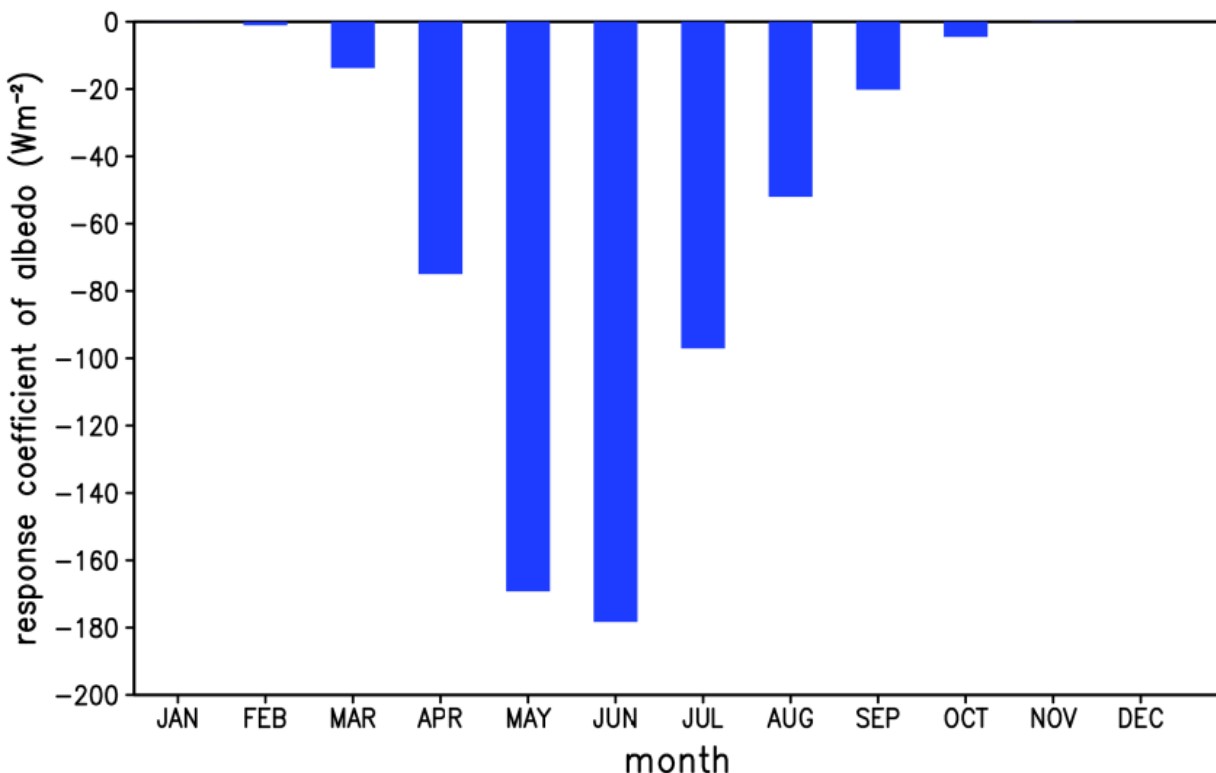

**Figure 6.** The monthly response coefficients (Wm$^{-2}$) of net shortwave flux to the albedo effect of sea ice.




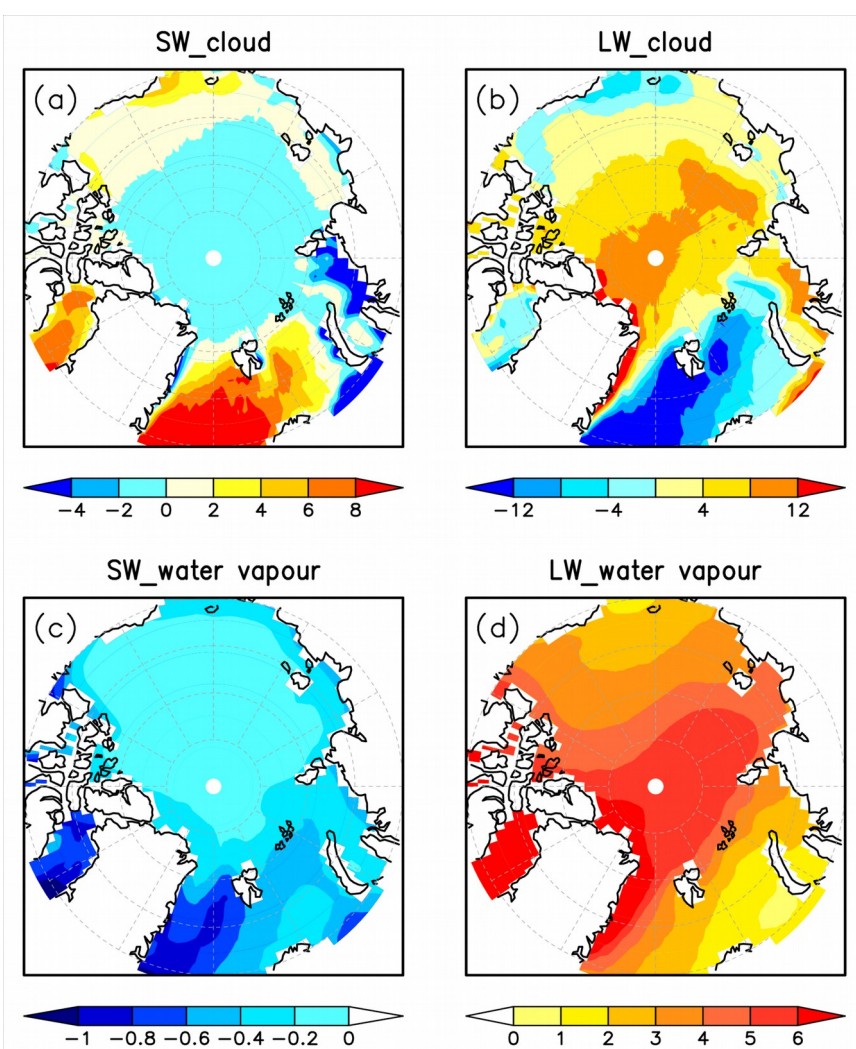

**Figure 7.** Spatial distributions of the Pliocene annual mean radiation fluxes change (Wm⁻², positive downward) with respect to the preindustrial period over the Arctic Ocean. (a) shortwave radiation due to cloud change, (b) longwave radiation due to cloud change, (c) shortwave radiation due to water vapour change, (d) longwave radiation due to water vapour change. Here, cloud and water vapour change is the value before removing the remote effects of clouds and water vapour.



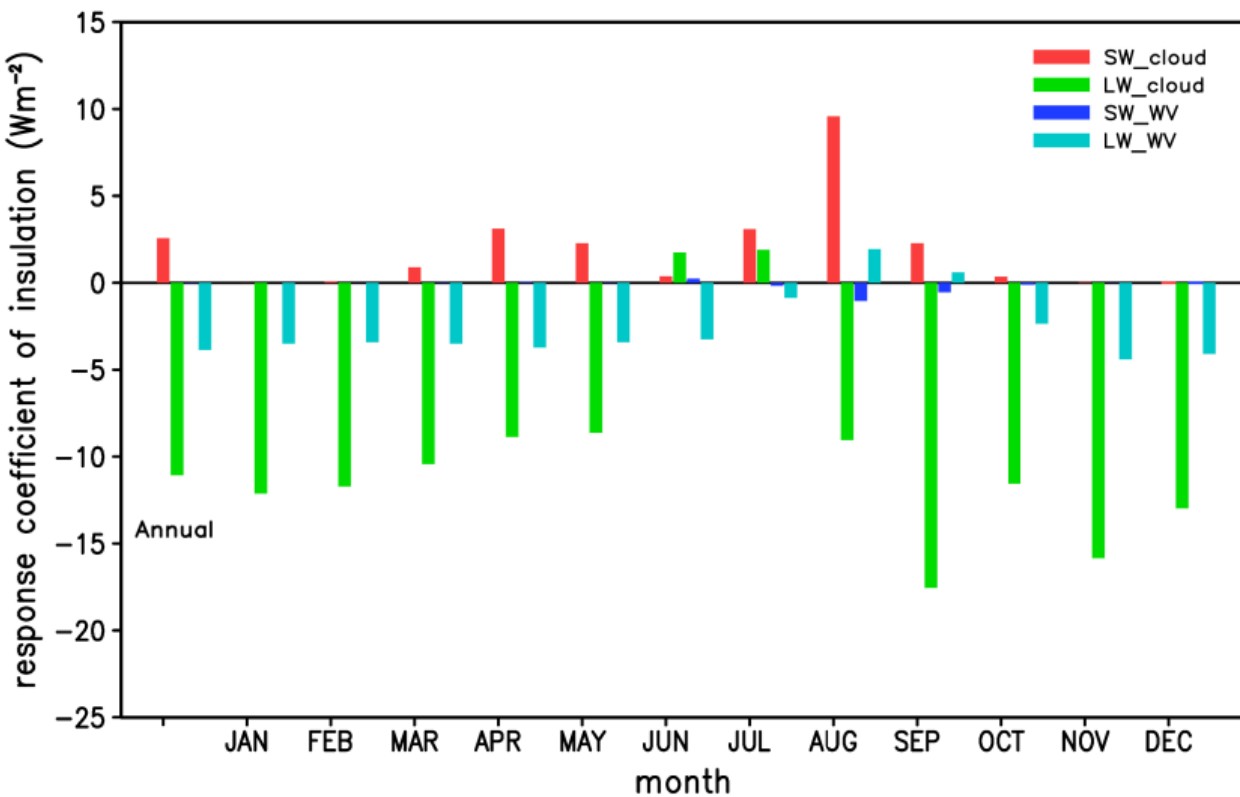

**Figure 8.** The annual and monthly response coefficients (Wm$^{-2}$) of net shortwave and longwave radiation flux related to cloud and water vapour change due to the insulation effect of sea ice. Here, the cloud and water vapour change is specified as the part related to sea ice
decrease.


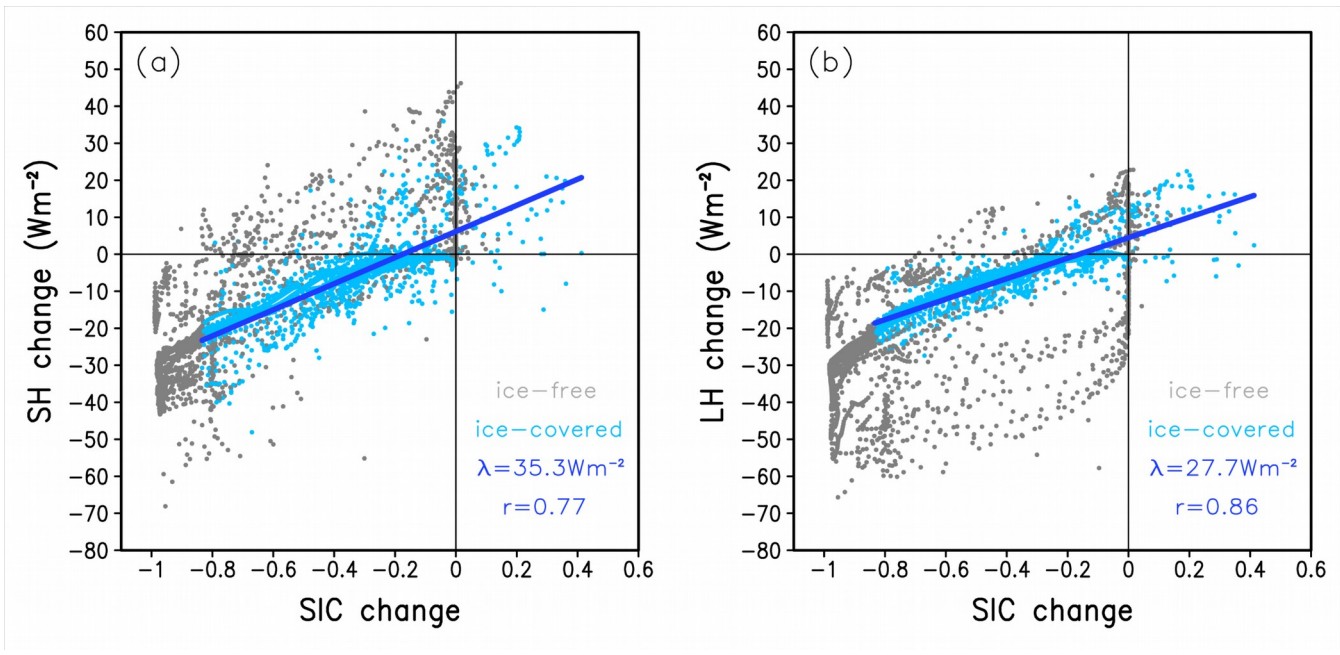


**Figure 9.** The annual mean sensible and latent heat flux change (Wm⁻², positive downward) related to the insulation effect of sea ice change averaged over the Arctic Ocean as a function of SIC change. Pliocene changes shown are computed relative to the preindustrial simulation. The ice-free and ice-covered regions here refer to the diagonal hatched and cross-hatched regions in Figure 2c, respectively. The blue line is the linear regression on the ice-covered scatter points, and the response coefficient (λ) and correlation coefficient (r) are
just for the ice-covered areas.


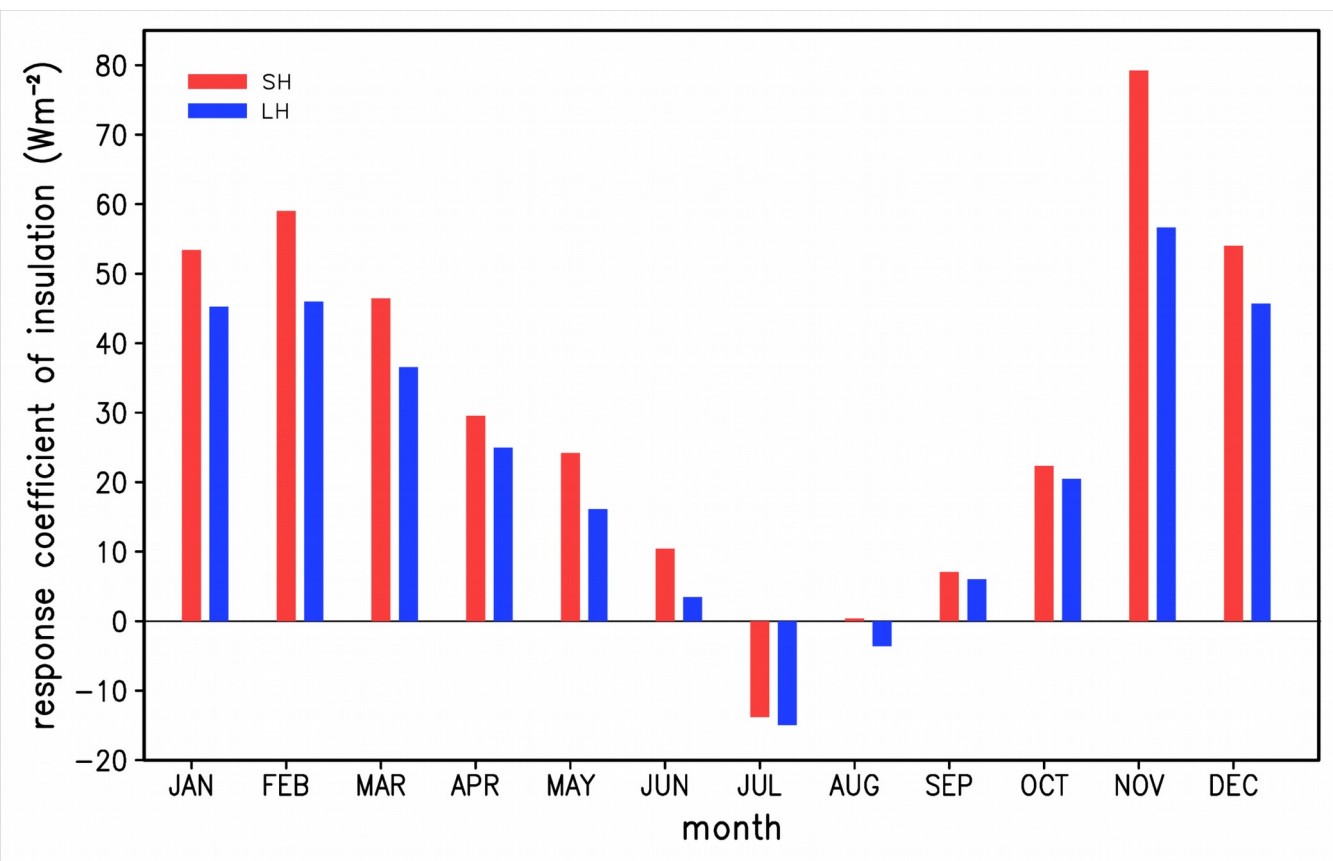

**Figure 10.** The monthly response coefficients (Wm$^{-2}$) of sensible and latent heat fluxes to the insulation effect of sea ice.