# Peer review of "Contribution of sea ice albedo and insulation effects to Arctic amplification in the EC-Earth Pliocene simulation"

_Climate of the Past, 2018_

## Referee Comment (RC1) · D. Bailey (Referee) · 20 Jul 2018

I have read the manuscript titled "Contribution of sea-ice albedo and insulation effects due to Arctic amplification in the EC-Earth Pliocene simulation." by Zheng et al. This manuscript describes two simulations of the EC-Earth in pre-industrial (1850) and Pliocene climates. The authors use two statistically based techniques known as the equilibrium feedback assessment (EFA) and the climate feedback and response analysis method (CFRAM). I am not familiar with these techniques and so this might be part of my misunderstanding of the analysis. I believe this manuscript may be acceptable for Climate of the Past Discussions, however it does require substantial revision to get

to this point. Here are my main issues:

1. The English language usage is problematic. While, it does not necessarily make the results incomprehensible, it still was difficult to interpret some of the results. It wasn't clear if it was the explanation from the authors or if it was a fundamental issue in the analysis. I started to correct some of the grammar, but it was taking too much time. So, I would encourage the authors to contact a native English speaker to check the usage.

2. One of my scientific issues is around the results in Figure 1. For one, the Y axis in panels (C) and (D) is different which provides the mistaken impression that the seasonal cycle of SST difference is much larger than it really is. Also, the discussion in the text does not make it clear why the seasonal cycle of SST difference is out of phase with the seasonal cycle of SAT. More is needed here.

3. Related to point 2, what is the variable in the model used to get SST? i.e. is this the first level of the ocean model? Is it the surface temperature in the atmosphere model? I am mainly concerned about the SST when there is ice present. This value should be very close to -1.8C when there is ice. Perhaps the authors could plot the absolute SST and SAT fields instead of the differences. I believe this might help explain part of the issue with the seasonal cycles being out of phase.

4. In Figure 2, I am very surprised that the Pliocene ice concentration is so low in the annual mean. You are using a present day value of CO2 of 400ppm I believe? Have you done the present day control to compare here? What is the top of the atmosphere imbalance in your runs? The sun still goes away in winter I presume, so I would expect more ice in the annual mean. Can you compare the seasonal cycle of extent in your Pliocene simulation to your pre-industrial and even perhaps a present-day control?I can't find the reference off hand, but Gerald Meehl has done some work looking at control runs versus transient runs.

5. As I mentioned, I am not familiar with the CFRAM/EFA techniques, so a bit more clarification here I think would be helpful for the readers of the journal. One concern

I have is how do you do the calculation when there is no ice? In other words, would your results change if you only computed the shortwave difference with respect to the sea ice difference at points where there was a nonzero ice concentration in both simulations? What about the relationship to the SST change?

---

## Referee Comment (RC2) · Anonymous Referee #2 · 23 Jul 2018

**Review of "Contribution of sea-ice albedo and insulation effects to Arctic amplification in the EC-Earth Pliocene simulation" for *Climate of the Past*.**

**Summary:**

This manuscript examines the impact of sea ice changes on the surface air and sea temperatures in the Arctic during the Pliocene, as simulated by EC-Earth. Specifically, they examine the energy flux impact of differences in surface albedo and surface ocean insulation spatially correlated with the changes in sea ice between the Pliocene and pre-industrial eras. They found that a reduction in albedo allows for much stronger short-wave heating during May, causing the biggest SST difference between the two eras to be during August. They also found that this extra stored heat was released back to the atmosphere via enhanced surface heat fluxes due to the decrease in sea ice insulation. This resulted in SAT differences between the two eras being largest during winter, and with an inverse seasonal timing relative to SST.

**Paper recommendation:**

Understanding how sea ice influences Arctic climate, particularly for higher-CO2 forcing scenarios, is of great importance in terms of understanding what our future climate state will be. The Pliocene also provides a unique test bed to examine sea ice in a warm climate, given that it was in near-equilibrium with a similar forcing compared to now, along with enough proxy data to validate model results. Thus I do believe this paper is a beneficial contribution to the literature. However, I do have some concerns in regards to the interpretation of their results, particularly sea ice's influence on clouds and water vapor. I also believe there needs to be some grammatical improvements to the text itself, to help make the manuscript easier to read. Given this, I am recommending **Major Revisions**. Although I hope these revisions are not too difficult to implement, I do believe they will significantly improve the final manuscript.

**Major issues:**

1. I am worried that some of the changes, especially related to water vapor and cloud cover, may not be directly related to changes in sea ice, but instead are related to a third variable that is also correlated with the sea ice (which would produce the correlation you found). The reason for this concern is that (I believe) you are using multi-year temporal averages to calculate the spatial correlation. However, over those time-scales potentially significant changes in the large-scale circulation could be present, which would impact clouds, water vapor, and sea ice through, for example, changes in atmospheric and oceanic heat transport. I sadly don't know of a great way to untangle all of these effects, but I could imagine calculating the correlation between sea ice and, say, vertically-integrated atmospheric moisture and heat flux, to see if it changed substantially in regions with substantial sea ice loss, or regions with noticeable changes in water vapor/cloud cover.

If that correlation analysis produces a relationship that is difficult to interpret, then I would at least add a plot showing the annual average differences between atmospheric heat and moisture fluxes, or at least wind fields, to help the reader understand how he atmospheric circulation over the Arctic is different in the Pliocene relative to the pre-industrial.

Related to this, are your correlations and analyses including the entire Arctic, or just the regions that have or had sea ice? If you are examining the entire Arctic, then I might recommend examining only the sea ice regions, as there appears to be large surface heat flux changes in regions that contain little-to-no sea ice in both eras, which could be contaminating the statistical relationships between sea ice and the associated surface flux changes.

2. I would strongly recommend having a native or very-proficient English speaker edit this document, as there are a large number of minor but noticeable grammatical errors, such as missing articles (a, an, the), and misuses of pluralization, combined with some strange phrasing that made the document difficult to understand sometimes. By improving the grammar/wording, I believe this manuscript would be made much stronger and more accessible to a wider audience.

**Minor issues:**

1. Are all the maps shown in this manuscript annual averages? If so this should be stated in the text (you mention it in a few locations in the text, just not everywhere).

2. It is interesting in your supplemental figure that the net TOA is negative (implying energy loss), yet all the surface variables show increasing warming. Does this imply a cooling higher up in the atmosphere, and thus a change in the atmospheric lapse rate (which could impact clouds and water vapor)?

3. The modeled SSTs do well in the Atlantic, but quite poor in the (North) Pacific. Could this imply a bias in the Pacific basin, which could impact heat fluxes coming from the Pacific into the Arctic? Just a short sentence or two on it would be all that I would recommend.

4. It would help to include the striping and cross-hatching in Figure 2f as well, so the reader can see how the surface heat flux changes with respect to the sea ice transitions.

5. How does sea ice thickness change in the regions where sea ice is still present in the Pliocene?

6. I assume the correlation shown in Figure 5 is significant, but it would still be good to state that somewhere in the text (as a high r-squared value doesn't always imply significance).

7. Line 252: If sea ice insulation reduces local water vapor and cloud cover, then it could certainly impact the local surface radiation budget, at least in the long-wave. Not sure if this sentence was just not worded correctly?

8.  It would help to show how the actual cloud cover is changing over the Arctic in the Pliocene (just showing cloud fraction by low and high clouds would help).  That would put the changes shown in Figure 7 into better context.

9.  It would be beneficial to see a spatial map of the latent and sensible heat flux changes, similar to Figures 2 and 3.

10.  Figure 8:  Any idea as to why the longwave cloud forcing in September is so large relative to all the other months?  Again, just a sentence or two would probably be all that is needed.

---

## Referee Comment (RC3) · Anonymous Referee #3 · 25 Jul 2018

Zheng et al. examined the contribution of sea ice albedo effects and insulation effects on the Arctic amplification using EC-Earth climate model simulations of preindustrial and Pliocene forcing. The attribution of different effects is assessed by the climate feedback and response analysis method (CFRAM) and the equilibrium feedback assessment (EFA)-like approach. Their analysis showed that the ice albedo effects were related to the summer SST warming and the insulation effects were related to the surface air temperature warming in winter. The results of this study are important to the understanding of the Pliocence warming as well as to improve the projection for the present warming trend. The organization of this manuscript is clear and easy to follow. The clarity of presentation is one of my main concerns. Some confusion or ambiguity are likely due to language issues. I suggest the authors either carefully examine the manuscript for grammar issues sentence by sentence or consult a native speaker to improve the writing of the manuscript. The assumption of linearity is very important for the attribution assessment. The authors have not provided physical reasons to support this assumption. It would be useful to provide some uncertainty estimates due to the inherent limitation of the analysis methods. Therefore, I recommend a major revision to the manuscript before it can be accepted for publication.

Other main concern:

The experiment setup. (1) If the objective of this study is to understand the contribution of ice albedo effects and insulation effects on the Arctic warming, why is it necessary to use the Pliocene simulation in the current setup? In this case, there are lots of publicly available model output from CMIP3 and CMIP5 to provide more robust results. (2) As is stressed by the authors, the justification of performing the Pliocene warming simulation is the availability of the proxy data constraint from paleo-records. If the objective is to understand the Pliocene warming, however, the orbital forcing adopts preindustrial conditions. Does it make the justification of this Pliocene simulation irrelevant? The Pliocene is a long time period and authors have not stated clearly what period are simulated. Are the fixed $CO_2$ with PRISM surface conditions the default setting for the PlioMIP2 simulations? If so, please give a brief description of it and the justification for this choice. It will be very helpful for readers who are not familiar with the PlioMIP2.

Specific comments:

Line 14: Define PRISM.

Line 24: "CO2" to "$CO_2$".

Line 24-25: This sentence seems to suggest the $CO_2$ increase is due to the monitoring. Please rephrase.

Line 25-28: Break up this sentence to two sentences and rewrite.

Line 26: "increased 1.1°C to" change to "increased 1.1°C compared to".

Line 31: "its reflectivity". Do you mean the sea ice reflectivity changes is responsible to changes in surface energy budget, etc.? Please rephrase to clarify.

Line 47-54: Some justification for Pliocene simulation is provided here. But the simulation in this study use mixed Pliocene and preindustrial conditions. Can you still use proxy data to verify the simulation? Some comparison with proxy data are shown in Fig. 1 but performance constraint by proxy data is still qualitative.

Line 146-147: It is not clear what "an approach" is referring to.

Line 150: "linear": do you have any reason to believe it should be linear. Please provide some argument to support this assumption or reference previous studies to support your statement.

Line 162: The choice of student-t test assumes Gaussian distribution. However, it is usually not the case for geophysical parameters. The spatial and temporal autocorrelation are not taken into account. The bootstrap method adopted by Liu et al. (2012) is more relevant, especially when "moving-block" is used to represent the length scale of the spatial or temporal features (van de Poll et al., 2006; Z. Liu et al., 2010).

Line 174-179: It is not clear what message these sentences are trying to convey. Is the Arctic amplification stronger in Pliocene simulation or not?

Line 180: Not exactly the SST and SAT in "Pliocene epoch". They are from the Pliocene simulations using some of the Pliocene conditions.

Fig. 1: The overlay of proxy data over the filled contour maps does not show the difference well. An additional map showing the differences might be more informative. Maybe add it to supplement?

Line 187: "off America". Do you mean USA or the America continents? The coastal warming is not very obvious or extensive in either. The warming off the west coast of Africa is more obvious. By "permanent El Nino-like feature", I expect to see extensive warming in equatorial eastern Pacific, which is not present in Fig. 1. Am I missing anything?

Line 197: "very sensitive" to what? "preindustrial" is used noun here and in several other occasions. Please add proper noun after it so that it is clear what it is referring to.

Line 200-201: Please rephrase this sentence.

Line 201: The definition of heat loss or heat gain is not given here. Consider move up the definition in line 204-208 before they are discussed.

Fig. 5: What are these dots? Are they differences between corresponding years in Pliocene and preindustrial runs? Does it matter how you pair up the years in Pliocene and preindustrial runs? Would it change the response coefficients? How much uncertainty would it introduce?

Line 242: How do you define "anomalies"? Are they with respect to the 100-year Arctic averages, Arctic monthly mean, or something else? Please describe clearly. Otherwise, it is impossible to interpret Table 1 to Table 4.

Line 252-253: This sentence is confusing. Please rewrite.

Line 263: Again, do you have any reason to believe they should be linear?

Line 274: What do you mean by "significant interaction"?

Line 276-284: The partition of ice-free and ice-covered conditions is confusing. How do you have SIC changes when it is ice-free?

Fig. 9: Is the regression line with respect to the ice-covered only? Would the regression line be different between ice-covered and ice-free? You would expect so because it is why you separate them in the first place.

Line 289-293: I am confused with the definition of ice-free and ice-covered changes relative to SIC. So I cannot follow the argument here. I will have to revisit after the authors' clarification.

Line 297-299: Does the melting of sea ice refer to the sea ice melt in spring or due to warming by high $CO_2$? Are you sure it is conduction not convection?

References:

Liu, Y., Key, J. R., Liu, Z., Wang, X., & Vavrus, S. J. (2012). A cloudier Arctic expected with diminishing sea ice. *Geophysical Research Letters*, *39*(5). https://doi.org/10.1029/2012GL051251

Liu, Z., Marchand, R., & Ackerman, T. (2010). A comparison of observations in the tropical western Pacific from ground-based and satellite millimeter-wavelength cloud radars. *Journal of Geophysical Research: Atmospheres*, *115*(D24), D24206. https://doi.org/10.1029/2009JD013575

van de Poll, H. M., Grubb, H., & Astin, I. (2006). Sampling uncertainty properties of cloud fraction estimates from random transect observations. *Journal of Geophysical Research: Atmospheres*, *111*(D22), D22218. https://doi.org/10.1029/2006JD007189

---

## Referee Comment (RC4) · Anonymous Referee #4 · 2 Aug 2018

Review of Clim. Past. Discussion: doi.org/10.5194/cp-2018-59:
**Contribution of sea-ice albedo and insulation effects to Arctic amplification in the EC-Earth Pliocene simulation**
J. Zheng, Q. Zhang, Q. Li, Q. Zhang, M. Cai

General Comments:

I recommend this work being accepted only after major revisions and rewriting for improved English composition. The late Pliocene is a unique warm period, representing a climate in equilibrium to modern levels of atmospheric greenhouse gas concentrations. This study makes a contribution to understanding the impact on sea ice response to Arctic amplification in a past warm climate state for which there is a large body of climate proxy data and other model studies. Most of the results shown here concerning the decomposition of the effects of sea-ice change on Arctic amplification are not new, as discussed in Serreze and Barry (2011), though the methodology, of using CFRAM may be new to this particular application. As this work pertains to a Pliocene simulation that compares well to the proxy-reconstruction, it would make a good contribution to the literature. More details of the CFRAM methodology used in this application would be beneficial to the readers.

Specific comments:

This manuscript needs to be thoroughly edited by someone with more English proficiency in order to improve the grammar and many awkward sentence structures. The result would be greater clarity for understanding the methodology and results. As written, many sections are not written with the necessary clarity for communicating the authors' intent. I point out some examples in my detailed comments by line number, below.

A TOA net energy imbalance of about -0.5Wm$^{-2}$ (Fig. S1) is large and suggests the simulation is not at a near equilibrium. In addition, the weak negative trend in the TOA energy imbalance suggests the model is moving away from equilibrium. Is this not globally integrated? A net loss of energy at the top of the model is inconsistent with both a positive SST trend and a negative sea ice concentration trend, both of which suggest the model is warming, unless a negative TOA flux is directed downward. This figure needs more explanation.

Overall, there is a general lack of specific details of the analyses presented. For example, how are the anomalies computed, i.e. are trends first removed? A more physical explanation is needed for why you are using spatial correlations because as written, "limitation of data and computation" is quite vague. More detail is also needed to explain how CFRAM was applied to the surface energy balance in the Arctic. For example, what is the "first part of CFRAM" (line 143) used to obtain the surface radiative fluxes. Some equations would be useful, as would a citation to the same specific use of CFRAM as applied here.

More detailed comments by line number: (Note: this is not a complete list of every grammatical issue in the text.)

11: "current warming climate" suggests transient climate change.

14: Define PRISM and give a citation.

17: "Given the facts…"

19: Run-on sentence structure.

20: During winter months…

29: Either "…in the recent decade…" or "…in recent decades…" "Moreover, an ice-free Arctic…"

31: "As the sea ice retreats…" Isn't it the average reflectivity of the surface that decreases due to a decrease in the fractional sea ice coverage and a decrease in the sea-ice albedo due to melting snow and ice. Are these two effects separated out here? For improved clarity it might be better to say "As sea ice retreats, the surface Arctic Ocean becomes less reflective and the enhanced open ocean region leads to greater air-sea heat exchange due to the reduction in the insulating effect of sea ice." However, as sea ice melts, its reflectivity changes and as the sea ice concentration changes the surface albedo is impacted. Both of these changes affect the net shortwave radiation at the surface.
"This leads to changes in the surface heat budget and changes in…"

33: "…possibly results from…"

38: "…consequence…has been reviewed"

46: shown

57: First "attributions" should perhaps be "characteristics" or "properties". Second "attributions" in line 59 should perhaps be "effects" or "mechanisms" here.

59: Run on sentence. I suggest cutting into two separate sentences or using a semi-colon instead of the comma after "…atmosphere and ocean"

69: should be "effects"

78: "represents"

79: Should amend "future climate at equilibrium with modern ghg levels"

138: "temperature"

143: "first part of CFRAM" is vague. More details on how CFRAM is applied should be given here.

169: Replace "present" with "modern", omit "benthic,"

170-174: Make a stronger statement linking the Arctic amplification statement starting on line 174, because that is the focus in this paper. To do this, I suggest rewriting here. I also suggest that you omit describing the tropical anomalies mainly because it is irrelevant to this work. Also, see Scroxton et al., Paleoceanography (2011); Brierley, PAGES News, 21(2), (2013); Watanabe et al., Nature, 471, 209-211, (2012); for recent papers discussing evidence for a robust Pliocene ENSO.

176-177: "…even though they have comparable $CO_2$ concentration…" replace with "…despite comparable $CO_2$ concentrations…" I also suggest breaking this sentence into two after the Ballantyne citation. Then in the next sentence suggest possible reasons why there is enhanced Arctic warming or amplification compared to today. I would add these newer citations for the amplified response to closed gateways: Otto-Bliesner et al. GRL, 44, 2017 and Feng et al. EPSL 466, 2017.

187: Omit the brief, sentence starting in line 187 with, "Meanwhile…" because the paper's focus is on the Arctic response.

192: "region" should be "regions" and omit "but they are apparently" for improved conciseness.

194: "Noteworthily…" Awkward. A potential replacement is "Notably"

194-195: Suggest replacing "…SAT, and the maximum…" with "SAT; the maximum…" using a semi-colon to join the two clauses instead of the conjunction "and".

197: The first line of this paragraph contains little meaning.

204:208: An equation for the net air-sea heat flux with the components presented symbolically is written in equation (4), but then these symbols are never used again, and reference to (4) is never made. I suggest removing the equation and stating this decomposition elsewhere. Also, what about the ice-ocean heat flux? A surface heat budget for the Arctic ocean should include the ice-ocean heat fluxes associated with the freezing and melting of sea-ice.

209-218: This paragraph discusses the anomalies or differences in the heat flux components in the Pliocene simulation as compared to the Preindustrial simulation, not heat fluxes. Every mention of a flux in this paragraph should reflect the fact that what is discussed are differences in the flux.

210: "The radiative and turbulent heat fluxes…" These are differences or anomalies.

211: "…the positive shortwave radiation is dominant…" should be "…the positive change in the shortwave radiation is dominant…"

212: "On the contrary" should be "In contrast"

220: "accounted as the synergy…" Awkward phrase. Suggest a rewrite of the first paragraph of this section for better clarity and conciseness. The definition of albedo needs to be more precise, to distinguish from planetary albedo.

224: relevant to net shortwave…

225: Be more specific: net shortwave flux at the surface… Also, is this due to changes in sea-ice and snow albedo (that is, changes in the albedo due to changes in the state of sea-ice or snow, or to the change in albedo over the ocean grid box due to both albedo change and change in sea-ice concentration?

226-227: "…most…shows…"

227: net shortwave

229: changes in sea ice extent

231: changes in snow cover … (and, I presume, any change affecting the actual sea-ice albedo which could be changes in the sea-ice or snow state, such as melting, because the albedo is a function of the sea-ice state as well as thickness according to earlier descriptions of LIM.)

233: net shortwave

234: "Regarding the…" This opening sentence is awkwardly stated and vague.

235: *net* shortwave radiation, i.e. shortwave radiation absorbed?

238: "The prominent oceanic heating in May and June seems inconsistent with the maximum SST warming in August,…" Second clause of sentence seems to explain why the SST warming lags the SW heating response, thus it is not "inconsistent" but "consistent."

240: About the SIC anomalies: Shouldn't this be something "like the mean spatial variance over the Arctic of the Pliocene SIC anomalies is not strongly variable over the mean annual cycle."

This is a curious result.  A nice additional supplemental figure could show the Pliocene anomaly pattern at the minimum and maximum of the SIC annual cycle.

243: "our correlation analysis indicates that…

246: Needs to be more specific: …seasonal cycle of incident shortwave or net shortwave? …sea ice concentration variation, or some other sea ice property variation?

250: …insulation effect of sea-ice…

251: omit "In fact,"

251: "insulation effect"

250-255:  This first paragraph should be rewritten.  The insulating effect of sea-ice has an indirect effect on the net surface shortwave and longwave fluxes. By separating the overlying atmosphere from the ocean, sea-ice reduces evaporation from the ocean resulting in a decrease of water vapour and cloud cover.  This reduction plays a non-negligible role in the amount of downward shortwave and longwave radiation reaching the surface. However, remote moisture transport also affects water vapour and cloud amount. Thus, …

256:262:  It is not clear in this paragraph whether the discussion is about the SW and LW feedbacks after the remote effects on clouds and water vapour have been removed. This is suggested in the Figure 7 caption however.

258: …cloud characteristics…

263:274:  Then this paragraph discusses just the local effect due to changes in sea ice concentration?  I don't know what is meant by "counterpart of sea-ice insulation."

264: "Like the steps performed to isolate the albedo effect…"

266: "In the annual mean…"

270:  "…shows a pronounced…"

271:  "Compared to…"  And this is being compared to the standard deviation of the shortwave anomalies due to clouds ?   Also, should be "SIC anomalies" and

272: Net shortwave radiation change and net longwave radiation change?

274:  …when there is a lack of…

279: ice-free conditions

280: the insulation effect

280: …and differentiate fluxes from ice-covered versus ice-free areas, not "ice-covered fluxes"

281: displays the Pliocene anomalies in …heat fluxes…as a function of SIC anomalies.

282: There is a larger spread in the turbulent heat flux anomalies over the ice-free area (grey symbols, corresponding to the diagonal hatched region in Figure 2c) than compared to anomalies from the ice-covered areas (light blue symbols, cross-hatched region in Fig. 2c) because the former is free from the constraint of sea-ice.

284: …and changes in SIC…

285:  Are these estimates of variance explained from the regression lines shown in Figure 9? Is this and the response coefficient shown in the figure just for the ice-covered region? Be specific in both the Figure caption and the main text.

286:293:  This paragraph jumps all over the place and is very unclear. First it discusses annual mean response coefficients vs. trends elsewhere, to the y-intercept of the regression line, then jumping to explaining seasonal variation.

287:  Noteworthily—a better choice would be "Notably" as mentioned previously.

288: "trend of sensible heat flux"  this comes out of nowhere, to what does this refer? Is this "trend" referring to 20th century trends observed? Describe accurately.

289: …turbulent flux anomaly axis?

290: "even without SIC change" for improved conciseness.

294: …to the sea-ice concentration? Or to sea-ice changes in general (thickness, albedo, concentration, etc.)? Also, replace "two" with "the".

295: "…have a similar…" and "…showing a negative response…"

296: "…maximum warming of SAT occurs in November as a consequence…"  It looks like changes in the net LW due to the response of clouds and water vapour is also a contributing factor to the warming in fall. As a complete budget for SAT is not presented, it would require adding heat transports and other fluxes, one can only suggest contributing factors.

Section 6  Summary and Discussion;

This section is mostly a summary of results. Additional discussion could compare these results to previous results (see Serreze and Barry, 2011), could compare to the other Pliocene simulations which showed weaker Arctic amplification, highlight what is new here, etc.

304-309: Paragraph should be rewritten as it contains many awkward phrases.  Also, I disagree that a model ever reveals a complete picture, but a model may be applied to investigate mechanisms and processes that help in understanding.

312: "…the effects of changes in"

314: "…expected to partly interpret the variability of heat flux"  Very unclear as to what this is supposed to mean.

315-328:  This paragraph appears to summarize the albedo and the insulation effects of sea ice on surface heat fluxes over the annual cycle, but doesn't seem to say anything about the Arctic amplification noted in the comparison of the Pliocene to preindustrial climate simulations. This section needs to be more specific.

326: sea ice decline…Is this the decline of Pliocene sea ice as compared to the preindustrial, or over a seasonal cycle? It is not clear whether anomalies are being discussed. Also, "accelerates" should be "amplifies" as "accelerates" suggests time evolution, and here equilibrium runs are being discussed.

Comments on Figures:

2) Please make the hatching in 2c more visible with another color or thickness. Be specific about describing the heat flux. "Net heat flux at the surface" Is this net heat flux at the surface (ice and ocean), or net heat flux at the surface of the ocean (air-sea and ice-ocean interfaces)?

3) State when the flux is a "net" flux change, that is for the sw and lw fluxes.

4) Be more specific.  Does the figure show the change in mean annual net shortwave flux at the surface?

5) and 6) net shortwave flux, also in 5) and 9) "All changes  are" or "All change is"

7) More clarification is needed in the figure caption. Also, "caused by" should be "related to", because causality is difficult to attribute in feedback processes.

8) Specify "net" again. Also, "caused by…" should be "related to…".

9) Define ice-free vs ice-covered regions here referring to Fig. 2c. Also…"Pliocene changes shown are computed relative to the preindustrial simulation." Describe the regression lines, i.e. which set of scatter points are being regressed. "caused by" should be "related to".

S1) Are all of these quantities global averages?

---

## Author Comment (AC1) · 5 Sep 2018

**Point-to-point response to reviewers' comments**

The comments are in black, and our answers are in blue.

**Reviewer #1:**

I have read the manuscript titled "Contribution of sea-ice albedo and insulation effects due to Arctic amplification in the EC-Earth Pliocene simulation." by Zheng et al. This manuscript describes two simulations of the EC-Earth in pre-industrial (1850) and Pliocene climates. The authors use two statistically based techniques known as the equilibrium feedback assessment (EFA) and the climate feedback and response analysis method (CFRAM). I am not familiar with these techniques and so this might be part of my misunderstanding of the analysis. I believe this manuscript may be acceptable for Climate of the Past Discussions, however it does require substantial revision to get to this point.

We are grateful for the positive evaluation and constructive comments that follow.

Here are my main issues:

1. The English language usage is problematic. While, it does not necessarily make the results incomprehensible, it still was difficult to interpret some of the results. It wasn't clear if it was the explanation from the authors or if it was a fundamental issue in the analysis. I started to correct some of the grammar, but it was taking too much time. So, I would encourage the authors to contact a native English speaker to check the usage. We have checked the grammatical errors, and a native English speaker has proofread the revised manuscript.

2. One of my scientific issues is around the results in Figure 1. For one, the Y axis in panels (C) and (D) is different which provides the mistaken impression that the seasonal cycle of SST difference is much larger than it really is. Also, the discussion in the text does not make it clear why the seasonal cycle of SST difference is out of phase with the seasonal cycle of SAT. More is needed here.

The Y axis in panels (c) and (d) in Figure 1 has been changed for clarity. The third paragraph of section 6 has been rephrased to explain why the Pliocene Arctic warming compared to the preindustrial simulation represents a maximum warming of SST in August and a maximum warming of SAT in November.

3. Related to point 2, what is the variable in the model used to get SST? i.e. is this the first level of the ocean model? Is it the surface temperature in the atmosphere model? I am mainly concerned about the SST when there is ice present. This value should be

very close to -1.8C when there is ice. Perhaps the authors could plot the absolute SST and SAT fields instead of the differences. I believe this might help explain part of the issue with the seasonal cycles being out of phase.

SST is the temperature at the first level of the ocean model. It is the surface temperature in the atmosphere model only in ice-free regions. We agree that SST is close to -1.8°C when there is ice.

The seasonal cycles of absolute SST and SAT are shown below. However, we focus on the changes in SST and SAT in this paper. SST is close to -1.8°C when there is ice, therefore the SST difference doesn't peak in winter when the SAT difference reaches a maximum. This does help explain part of the issue with the seasonal cycles being out of phase.

4. In Figure 2, I am very surprised that the Pliocene ice concentration is so low in the annual mean. You are using a present day value of  $CO_2$  of 400ppm I believe? Have you done the present day control to compare here? What is the top of the atmosphere imbalance in your runs? The sun still goes away in winter I presume, so I would expect more ice in the annual mean. Can you compare the seasonal cycle of extent in your Pliocene simulation to your pre-industrial and even perhaps a present-day control? I can't find the reference off hand, but Gerald Meehl has done some work looking at control runs versus transient runs.

The Pliocene simulation is a Core PlioMIP2 experiment, and the atmospheric  $CO_2$  concentration is set to 400 ppm according to the protocol of PlioMIP2 (similar to a

present day value of  $CO_2$ ). The present day control has not been carried out in this study.

Strictly speaking, the TOA net radiation should balance after reaching an equilibrium. However, a small imbalance generally remains associated with numerical errors, such as  $-1.5 \text{ Wm}^{-2}$  displayed in ERA-Interim (Hazeleger et al., 2011) and 0.9 Wm-2 shown in Trenberth et al. (2009). From our last 200 years output in the Pliocene simulation, the mean TOA net radiation (globally integrated) is about -0.5 Wm-2 and its trend close to zero. The trend of mean SST is about 0.02 K/century, which fulfils the PMIP4 equilibrium criterion that the trend of mean SST should be less than 0.05 K/century (Kageyama et al., 2018).

The mid-Pliocene Warm Period (3.264–3.025 Ma) has a near-modern orbital forcing and the orbital forcing in Pliocene simulation adopts preindustrial conditions, thus the impact of orbital forcing is not considered in the study.

The seasonal cycles of sea ice concentration in Pliocene simulation and in preindustrial simulation are shown below. The two simulations both are equilibrium runs and other transient runs will be discussed in the further study.

5. As I mentioned, I am not familiar with the CFRAM/EFA techniques, so a bit more clarification here I think would be helpful for the readers of the journal. One concern I have is how do you do the calculation when there is no ice? In other words, would your results change if you only computed the shortwave difference with respect to the sea ice difference at points where there was a nonzero ice concentration in both simulations? What about the relationship to the SST change?

One of the most important, but difficult to understand, aspects of the analysis methods is the decomposition of partial radiative perturbations, so more details including a partial radiative perturbation equation are added in Section 2.2 in the revised version. We have checked the response coefficients of the annual mean net shortwave flux change caused by albedo change due to SIC change, according to the category of ice-free or ice-covered. Here the threshold of ice-free and ice-covered conditions is 15% sea ice concentration, as commonly used in sea ice study. It can be inferred that the results would change if sea ice status is considered, thus the response coefficients may depend not only on the SST change but also on the SST itself.

In the revised version only ice-covered regions are examined, as there appears to be large surface heat flux changes in regions that contain little-to-no sea ice in both eras, which could be contaminating the statistical relationships between sea ice and the associated surface flux changes.

| The preindustrial status | The Pliocene status    | response coefficients   |
|--------------------------|------------------------|-------------------------|
| ice-covered              | ice-covered            | $-43.0 \text{ Wm}^{-2}$ |
| ice-covered              | ice-free               | $-49.9 \text{ Wm}^{-2}$ |
| ice-free                 | ice-free               | -38.8 Wm -2  |
| ice-free                 | ice-covered            | $-43.5 \text{ Wm}^{-2}$ |
| ice-covered & ice-free   | ice-covered & ice-free | $-46.5 \text{ Wm}^{-2}$ |

[revised manuscript text omitted]
. Recently the The EC-Earth model ishas also been applied to understand the past climateclimates, such as changes in the change of Arctic climate (Muschitiello et al., 2015), African monsoonmonsoons (Pausata et al., 2016; Gaetani et al., 2017), tropical evelone cyclones (Pausata et al., 2017a), and ENSO activity (Pausata et al., 2017a) al., 2017b) during the mid-Holocene. In this study we apply the model to the mid-Pliocene climate and focus on the effects of sea -ice on Arctic climate change.
- 120

Two numerical experiments are performed with EC-Earth to facilitate this study. One is the preindustrial control run with the 1850 CO2 concentration of 284.725 ppm, and the other is the mid-Pliocene warm period (3.264-3.025 Ma) sensitivity experiment in which the atmospheric  $CO_2$  concentration is set to 400 ppm. Following the protocol of the Pliocene Model Intercomparison Project, phase 2 (PlioMIP2, Haywood et al., 2016b), several configurations are modified in the Pliocene simulation: (1) in the Pliocene experiment, all other trace gases exceptother than CO2, such as CH4-and, N2O, and aerosols-in 125 the Pliocene experiment, are specified to be identical to the preindustrial run- to account for the absence of proxy data. (2) Orbit forcing, including eccentricity, obliquity, and precession, remains same within the preindustrial run- as in the mid-

130

4

Pliocene warm period, which has a near-modern orbital forcing. (3) Enhanced boundary condition from the Pliocene Research, Interpretation and Synoptic Mapping group (PRISM, Dowsett et al., 2016)), including land-sea mask, topography, bathymetry, and ice-sheet, are applied in the Pliocene experiment where the land sea mask, orography,

bathymetry, vegetation. The global distributions of lake, soil, and ice-sheetbiome are modified accordingly to match the new land-sea mask and ice reconstruction. The integrations of the preindustrial control run and the Pliocene experiment are carried out for 500 years, and it takes approximate approximately 300 years for the model to reach equilibrium. From our last 200 years of output in the Pliocene simulation (see Figure S1 in the Supplement), the mean top of the atmosphere (TOA) net radiation is about -0.5 Wm-2 and its trend is near zero. The trend of mean SST is about 0.027 K/century, which fulfils the PMIP4 criterion that the trend of mean SST should be less than 0.05 K/century (Kageyama et al., 2018). In this study, the last 100-year-mean of all variables are used for analysis, and the Pliocene climate anomalies are calculated with respect toby subtracting the mean of the preindustrial control run. The Arctic insimulation without trends removal. In the following analysis, the Arctic is defined as the region poleward of 70 °N.

**2.2 Climate feedback and response analysis method (CFRAM)**

140 Radiative forcing varies as CO2 concentration increasesClimate system warming in the Pliocene experiment is driven by variation in radiative forcing, which drives climate system warming is in turn caused by increased CO2 concentration. In response to temperature change, factors such as surface albedo, cloud, water vapour, and air temperature will adjust and feedback until the climate system reaches equilibrium. The contribution from each factor can be quantitatively evaluated by climate feedback analysis. The traditional Traditional climate feedback analysis methodmethods, such as partial radiative 145 perturbation-(PRP) technique, is based on TOA radiative budget (Wetherald and Manabe, 1988), while the radiative kernel method can be extended to the surface and remain computationally efficient (Soden and Held, 2006; Pithan and Mauritsen, 2014). However, none of them takes individual physical processes into account, particularly non-radiative processes. The

CFRAM contains two parts: one is decomposing radiative perturbation into individual contribution, including shortwave and longwave components, from CO2, surface albedo, cloud, water vapour, and air temperature. It:

elimate feedback and response analysis method (CFRAM), proposed by Lu and Cai (2009)), overcomes this limitation.

$$\Delta Q_{rad} = \Delta \left( S + R \right)_{co_2} + \Delta S_{albedo} + \Delta \left( S + R \right)_{cloud} + \Delta \left( S + R \right)_{WV} + \Delta R_T ,$$
(1)

where  $\Delta Q_{red}$  is performed by offline calculation usingtotal radiative transfer model (Fu and Liou, 1993) with flux perturbation at the output from 
[revised manuscript text omitted]
 highconfidence proxy data. This is consistent with the resultsresult of Koenigk et al. (2013), which pointed outsuggests that the sea ice change in EC-Earth is strong and that the EC-Earth simulations show a strong Arctic amplification compared to most

- CMIP3 models. Meanwhile, a warming can be seen along the coastal upwelling zones off the America, which implies a permanent El Niño-like feature. According to Figure 1b, the Pliocene SAT north of 70 °N is as much as 10–18 °C higher than the preindustrial period, similar to the mid-Pliocene paleoclimate estimate by Robinson et al. (2008).
- FigureFigures 1a and 1b also show that the SST and SAT anomaly patterns are somewhat similar over low- and midlatitude region, but they are apparentlyregions, different from over high-latitude regionregions, particularly over the Arctic Ocean, which iswas previously illustrated by Hill et al. (2014). This disparity results from the intense air—sea coupling over tropical and subtropical oceanoceans, while the air—sea interaction is relatively weak over the Arctic Ocean owing to the albedo and insulation effects of sea -ice. Noteworthily, theNotably, SST warming-of SST averaged over the Arctic Ocean shows a distinct seasonal evolution from that of SAT, and; the maximum warming in SST occurs in summer, while the maximum warming in SAT happens during winter (FigureFigures 1c and 1d).

The SIC is very sensitive during the different period as shown in Figure 2a-e. During the preindustrial period, the annual mean sea -ice appears to cover the whole Arctic Ocean except for the Greenland Sea, the Norwegian Sea, and the Barents Sea, and it retreats to the western Arctic Ocean in the Pliocene, leading to a significant decrease of sea -ice extent over the Fram Basin and Baffin Bay- (Figures 2a-c). Consequently, the net air-sea interface heat exchange at the surface of ice or ocean varies greatly (Figure 2d-f). The sea-ice f). The net heat flux and other flux terms mentioned hereafter are defined as positive downward. A positive value means that the ocean gains heat from the atmosphere and a negative value means oceanic heat loss. The net heat flux over the sea ice covered area seems to beclearly shows net heat loss during both the

preindustrial period and the Pliocene- (Figures 2d and 2e). Thus, it can be expected that net heat gain will occur when the sea
-ice declines. However, the Fram Basin and Baffin Bay displaysdisplay pronounced heat loss, which might be linked to the disappearance of sea -ice in the Pliocene (Figure 2b).

The net heat flux at the air-sea interfacesurface of ice or ocean can be writtenrepresented as

 $Q_{net} = Q_{sw} + Q_{lw} + Q_{sh} + Q_{lh}, \quad (4)$

- 240 Where  $Q_{sw}$  and  $Q_{tw}$  are the sum of four terms: the net solar-shortwave and radiative flux, the net 
[revised manuscript text omitted]
 flux changefluxes as a function of SIC anomalies. There are-is a larger spreads of spread in the turbulent heat flux changeanomalies over the ice-free areaareas (grey symbols, corresponding to the diagonal hatched region in Figure 2c) than that of in anomalies from the ice-covered, areas (light blue symbols, crosshatched region in Figure 2c) because the former is free from the constraint of sea -ice. The constraint of sea -ice can be 345 apparently captured through the scatter plot of turbulent heat flux and changes in SIC change (the (light blue plot in Figure 9,
- corresponding to the diagonal crosshatchsymbols). For the ice-covered areas-in Figure 2c), and, SIC can explain approximate 59% and 74% (square of correlation coefficient) of the variance in the sensible heat flux and latent heat flux, respectively.
- The linear regressions of sensible and latent heat flux anomalies on SIC are similar but different. The response coefficient of sensible heat flux (35.3 Wm-2) to SIC is larger than that of latent heat flux (27.7 Wm-2), for the ice-covered areas, which 350 means that the sensible heat flux is more sensitive to SIC change than the latent heat flux. NoteworthilyNotably, this is different from the turbulent heat flux variability over low- and mid-latitude regions, where the trendyariability of sensible heat flux is significantly less than that of latent heat flux-(e.g., such as the trend of turbulent heat flux over the low- and mid-latitude North Pacific and North Atlantic oceans from 1984-2004 (Li et al., 2011). The positive intercept on the turbulent flux anomaly axis implies more heat gain at the sea surface, even if there is nowithout SIC change. Because the large specific heat capacity of seawater leads to less warming of the ocean than of the atmosphere, therefore the sea surface and air temperature difference or (the specific humidity difference) decreases induring the cold season when the turbulent
- 355

heat transport is the most pronounced, and consequently resulting in the lessa lower annual heat loss from the ocean to the atmosphere.

360

Figure 10 shows the seasonal response coefficient of the sensible and latent heat fluxes to the sea-ice. Apparently two SIC. It appears that the turbulent heat fluxes have thea similar seasonal evolution, peaking in November and showing a negative response in July. Therefore the maximum warming of SAT occurs in November as a consequence of, the prompt atmospheric prompt response to turbulent heating- is an important contributing factor to the maximum SAT warming that occurs in November. The melting of sea-ice ice due to warming by high levels of CO2 can attenuate the insulation effect and result in more heat transfer through the processes of conduction or evaporation from the ocean to the atmosphere when SST 365 is higher than SAT $\frac{1}{2}$  therefore, the turbulent heat fluxes correlate positively with SIC in all seasons except summer (Table 3). If SAT is higher than SST, (for instance, in July-the), sea -ice will inhibit the heat transfer from the atmosphere to ocean; thus, the negative correlation emerges. However, the correlations between the turbulent heat fluxes and SIC in summer are not statistically significant (Table 3), indicating other factors might be dominant rather than sea -ice might be dominant.

**6 Summary and discussion**

- 370 In the present work we attempt to understand the albedo and insulation effects of sea-ice, on a warm Arctic climate during Pliocene simulated by EC-Earth coupled model. In contrast to Arctic amplification in the Pliocene has previously been addressed from reconstructed data (e.g. Robinson et al., 2008; Brigham-Grette et al., 2013); however these data tell only part of the story because of a scarcity of data sites. A model may be applied to investigate mechanisms and processes that help understanding. In contrast to the underestimation of multi-model ensembles documented in Dowsett et al. (2012), the EC-
- 375 Earth Pliocene simulation can better display some main features manifested in the characteristics that have been revealed by the paleoclimate proxy data from deep-sea oxygen isotope analysis. Thus the EC-Earth coupled model is used in the present work to simulate the Pliocene climate and study the contribution of sea ice albedo and insulation to Arctic amplification-in Pliocene had been confirmed by reconstructed data (e.g. Robinson et al., 2008; Brigham Grette et al., 2013). Proxy data, however, tell only part of the story. Thus a model is applied and it can reveal the complete picture with reasonable 380 explanation.

385

- As a key to reveal the important features of Arctic amplification, the air-Air-sea heat flux variation in response to Arctic sea -ice change is quantitatively assessed by CFRAM and an EFA-like method-in order to reveal important features of Arctic amplification. Table 4 summarizes the results presented in sectionSections 4 and 5, which separately illustratedillustrate the effects of changes in albedo and insulation of sea -ice on surface heat exchange. Annual mean and seasonal evolution of effects are both considered, and. These allow us to partly interpret the mechanisms of Arctic amplification because the results are merely the contribution from sea -ice change. A complete energy budget, including dynamical and thermodynamical processes, is required to understand Arctic amplification comprehensively.

The Pliocene Arctic amplification compared to the preindustrial simulation represents a maximum SST warming in August and expected to partly interpret the variability of heat flux.

- 390 The albedoa maximum SAT warming in November, which might be asso

---

## Author Comment (AC2) · 5 Sep 2018

**Point-to-point response to reviewers' comments**

*The comments are in black, and our answers are in blue.*

**Reviewer #2:**

**Summary:**
This manuscript examines the impact of sea ice changes on the surface air and sea temperatures in the Arctic during the Pliocene, as simulated by EC-Earth. Specifically, they examine the energy flux impact of differences in surface albedo and surface ocean insulation spatially correlated with the changes in sea ice between the Pliocene and pre-industrial eras. They found that a reduction in albedo allows for much stronger short-wave heating during May, causing the biggest SST difference between the two eras to be during August. They also found that this extra stored heat was released back to the atmosphere via enhanced surface heat fluxes due to the decrease in sea ice insulation. This resulted in SAT differences between the two eras being largest during winter, and with an inverse seasonal timing relative to SST.

**Paper recommendation:**
Understanding how sea ice influences Arctic climate, particularly for higher-CO2 forcing scenarios, is of great importance in terms of understanding what our future climate state will be. The Pliocene also provides a unique test bed to examine sea ice in a warm climate, given that it was in near-equilibrium with a similar forcing compared to now, along with enough proxy data to validate model results. Thus I do believe this paper is a beneficial contribution to the literature. However, I do have some concerns in regards to the interpretation of their results, particularly sea ice's influence on clouds and water vapor. I also believe there needs to be some grammatical improvements to the text itself, to help make the manuscript easier to read. Given this, I am recommending **Major Revisions**. Although I hope these revisions are not too difficult to implement, I do believe they will significantly improve the final manuscript.

We are grateful for the positive evaluation and constructive comments that follow.

**Major issues:**
1. I am worried that some of the changes, especially related to water vapor and cloud cover, may not be directly related to changes in sea ice, but instead are related to a third variable that is also correlated with the sea ice (which would produce the correlation you found). The reason for this concern is that (I believe) you are using multi-year temporal averages to calculate the spatial correlation. However, over those

time-scales potentially significant changes in the large-scale circulation could be present, which would impact clouds, water vapor, and sea ice through, for example, changes in atmospheric and oceanic heat transport. I sadly don't know of a great way to untangle all of these effects, but I could imagine calculating the correlation between sea ice and, say, vertically-integrated atmospheric moisture and heat flux, to see if it changed substantially in regions with substantial sea ice loss, or regions with noticeable changes in water vapor/cloud cover.

If that correlation analysis produces a relationship that is difficult to interpret, then I would at least add a plot showing the annual average differences between atmospheric heat and moisture fluxes, or at least wind fields, to help the reader understand how the atmospheric circulation over the Arctic is different in the Pliocene relative to the pre-industrial.

Related to this, are your correlations and analyses including the entire Arctic, or just the regions that have or had sea ice? If you are examining the entire Arctic, then I might recommend examining only the sea ice regions, as there appears to be large surface heat flux changes in regions that contain little-to-no sea ice in both eras, which could be contaminating the statistical relationships between sea ice and the associated surface flux changes.

The differences of sea ice concentration, total column water vapour and wind fields at 10m over the Arctic in the Pliocene relative to the preindustrial are shown below.

[Figure]

There is no significant correlation between sea ice concentration and wind fields at 10m ($r_{sic,uwnd}$=-0.06, $r_{sic,vwnd}$=-0.22). The total column water vapour is the combination of local source and large-scale circulation transport, and the former is correlated with sea ice concentration, which might lead to a weak correlation (r=-0.44) between SIC and total column water vapour. As you mentioned, a variety of processes are tangled, so in the study we attempted to decompose into individual contribution by CFRAM and EFA-like method.

Our correlations and analyses are including the entire Arctic except for turbulent heat fluxes. Note that ice-free and ice-covered are classified by the threshold of 15% sea ice concentration as commonly used in sea ice study. The ice-covered regions in both eras are examined in the revised manuscript as suggested.

2. I would strongly recommend having a native or very-proficient English speaker edit this document, as there are a large number of minor but noticeable grammatical errors, such as missing articles (a, an, the), and misuses of pluralization, combined with some strange phrasing that made the document difficult to understand sometimes. By improving the grammar/wording, I believe this manuscript would be made much stronger and more accessible to a wider audience.

We have corrected the grammatical errors. A native English speaker has proofread the revised manuscript.

**Minor issues:**
1. Are all the maps shown in this manuscript annual averages? If so this should be stated in the text (you mention it in a few locations in the text, just not everywhere).

Yes, all the maps shown are annual averages. We have stated "annual mean" in the text and figure captions.

2. It is interesting in your supplemental figure that the net TOA is negative (implying energy loss), yet all the surface variables show increasing warming. Does this imply a cooling higher up in the atmosphere, and thus a change in the atmospheric lapse rate (which could impact clouds and water vapor)?

We agree. The negative TOA flux and the positive net surface energy fluxes in the EC-Earth simulation (also shown in ERA-Interim, Hazeleger et al., 2011) might possibly affect the atmospheric lapse rate. It is interesting but beyond the scope of this paper and could be investigated in the further study.

3. The modeled SSTs do well in the Atlantic, but quite poor in the (North) Pacific. Could this imply a bias in the Pacific basin, which could impact heat fluxes coming

from the Pacific into the Arctic? Just a short sentence or two on it would be all that I would recommend.

The overlay of proxy data over the filled contour maps does not clearly show the difference, therefore we show the difference of annual mean SST anomaly (Pliocene minus preindustrial) between EC-Earth simulation and the proxy data in Figure S2. The underestimation in high latitude regions, similar to multi-model mean in Dowsett et al. (2012) but significantly less in bias, is found both in the Atlantic and in the North Pacific. Thus it is difficult to confirm the impact on heat fluxes coming from the Pacific into the Arctic.

4. It would help to include the striping and cross-hatching in Figure 2f as well, so the reader can see how the surface heat flux changes with respect to the sea ice transitions.

Done.

5. How does sea ice thickness change in the regions where sea ice is still present in the Pliocene?

We found that sea ice thickness would show a decline and reduce the surface albedo due to the warming in Pliocene. The decrease in the surface albedo due to a decrease in the fractional sea ice coverage (sea ice concentration) is focused on in this paper, and the decrease in the sea-ice albedo due to melting ice (sea ice thickness change) is mentioned in section 4. It can be found from section 4 that the former is dominant when they affect the net shortwave radiation at the surface.

6. I assume the correlation shown in Figure 5 is significant, but it would still be good to state that somewhere in the text (as a high r-squared value doesn't always imply significance).

The statistical significance of the correlation coefficient is stated in the revised version.

7. Line 252: If sea ice insulation reduces local water vapor and cloud cover, then it could certainly impact the local surface radiation budget, at least in the long-wave. Not sure if this sentence was just not worded correctly?

The sentences are confusing and they have been rewritten in the revised version as "By separating the overlying atmosphere from the ocean, sea ice reduces evaporation from the ocean, resulting in a decrease in water vapour and cloud cover. This reduction plays a non-negligible role in the amount of downward shortwave and longwave radiation reaching the surface. However, remote moisture transport also affects water vapour and cloud amount.".

8. It would help to show how the actual cloud cover is changing over the Arctic in the Pliocene (just showing cloud fraction by low and high clouds would help). That would put the changes shown in Figure 7 into better context.

The low and high cloud cover changes are shown below (See also Figure S3). The significant decrease of low cloud cover in North Atlantic may enhance incoming shortwave radiation and weaken downwelling longwave radiation, thus contributing to the positive anomaly in shortwave radiation (Figure 7a) and negative anomaly in longwave radiation (Figure 7b) in North Atlantic. Similarly, the increase of high cloud cover east and north of Greenland is responsible for the positive anomaly in longwave radiation (Figure 7b) over the related areas. These analyses are added in the revised text and Figure S3.

[Figure]

9. It would be beneficial to see a spatial map of the latent and sensible heat flux changes, similar to Figures 2 and 3.

A spatial map of the latent and sensible heat flux changes has been shown in Figure 3c and 3d.

10. Figure 8: Any idea as to why the longwave cloud forcing in September is so large relative to all the other months? Again, just a sentence or two would probably be all that is needed.

The mean cloud cover over the Arctic peaks in September, which contributes to large longwave radiation changes. In addition, there is a significant linear relationship between sea ice concentration and longwave radiation changes due to cloud in

September. Therefore the the longwave cloud forcing in September is much larger than in the other months. We have added this explanation in the last paragraph of Section 5.1.

[revised manuscript text omitted]
 asthe value before removing the part caused by sea-ice decreaseremote effects of clouds and water vapour.

690

695

[Figure]

[Figure]

**Figure 8.** The annual and monthly response coefficients (Wm$^{-2}$) of  shortwave and longwave radiation flux related to cloud and water vapour change to the insulation effect of sea -ice. Here, the cloud and water vapour change is specified as the part related to sea -ice decrease.

705

[Figure]

710

**Figure 9.** The annual mean sensible and latent heat flux change (Wm$^{-2}$, positive downward) related to insulation effect of sea - ice change averaged over the Arctic Ocean as a function of SIC change. Pliocene changes shown are computed relative to the preindustrial simulation. The ice-free and ice-covered regions here refer to the diagonal hatched and cross-hatched regions in Figure 2c, respectively. The blue line is the linear regression on the ice-covered scatter points, and the response coefficient (λ) and
715 correlation coefficient (r) are just for the ice-covered areas.

720

[Figure]

725  **Figure 10.** The monthly response coefficients (Wm$^{-2}$) of sensible and latent heat fluxes to the insulation effect of sea -ice.

**Supplementary Information**

[Figure]

**Figure S1.** The global annual mean of last 200 model years output in the Pliocene simulation (The negative TOA net radiation represents a heat loss of the earth-atmosphere system.)

[Figure]

**Figure S2.** The difference of annual mean SST anomaly (Pliocene minus preindustrial, K) between EC-Earth simulation and the proxy data at 95% confidence-assessed marine sites from Deep Sea Drilling Project (DSDP) and Ocean Drilling Program (ODP).

[Figure]

**Figure S3.** The difference of annual mean low cloud cover (a) and high cloud cover (b) anomaly in Pliocene with respect to the preindustrial.

---

## Author Comment (AC3) · 5 Sep 2018

**Point-to-point response to reviewers' comments**

*The comments are in black, and our answers are in blue.*

**Reviewer #3:**

Zheng et al. examined the contribution of sea ice albedo effects and insulation effects on the Arctic amplification using EC-Earth climate model simulations of preindustrial and Pliocene forcing. The attribution of different effects is assessed by the climate feedback and response analysis method (CFRAM) and the equilibrium feedback assessment (EFA)-like approach. Their analysis showed that the ice albedo effects were related to the summer SST warming and the insulation effects were related to the surface air temperature warming in winter. The results of this study are important to the understanding of the Pliocene warming as well as to improve the projection for the present warming trend. The organization of this manuscript is clear and easy to follow. The clarity of presentation is one of my main concerns. Some confusion or ambiguity are likely due to language issues. I suggest the authors either carefully examine the manuscript for grammar issues sentence by sentence or consult a native speaker to improve the writing of the manuscript. The assumption of linearity is very important for the attribution assessment. The authors have not provided physical reasons to support this assumption. It would be useful to provide some uncertainty estimates due to the inherent limitation of the analysis methods. Therefore, I recommend a major revision to the manuscript before it can be accepted for publication.

We are grateful for the positive evaluation and detailed comments that follow.

1) We have corrected the grammar issues, a native English speaker has proofread the revised manuscript.

2) For the attribution assessment, CFRAM can linearly decompose radiative perturbation into individual contribution, including shortwave and longwave components, from CO2, surface albedo, cloud, water vapour, and air temperature:

$$\Delta Q_{rad} = \Delta\left(S+R\right)_{co_2} + \Delta S_{albedo} + \Delta\left(S+R\right)_{cloud} + \Delta\left(S+R\right)_{WV} + \Delta R_T$$,

which can be validated by summing all the partial radiation perturbation from CFRAM output and comparing with the radiative perturbation output. They are almost identical (The figure is shown below), thus explaining the linearity.

[Figure]

Other main concern:

The experiment setup. (1) If the objective of this study is to understand the contribution of ice albedo effects and insulation effects on the Arctic warming, why is it necessary to use the Pliocene simulation in the current setup? In this case, there are lots of publicly available model output from CMIP3 and CMIP5 to provide more robust results. (2) As is stressed by the authors, the justification of performing the Pliocene warming simulation is the availability of the proxy data constraint from paleo-records. If the objective is to understand the Pliocene warming, however, the orbital forcing adopts preindustrial conditions. Does it make the justification of this Pliocene simulation irrelevant? The Pliocene is a long time period and authors have not stated clearly what period are simulated. Are the fixed $CO_2$ with PRISM surface conditions the default setting for the PlioMIP2 simulations? If so, please give a brief description of it and the justification for this choice. It will be very helpful for readers who are not familiar with the PlioMIP2.

(1) The Pliocene simulation is selected because of three reasons:

    1) The Pliocene epoch, the most recent warm period with the similar $CO_2$ concentration as today, is an analogue of current or future climate change.

2) The Pliocene simulation can be partly verified by proxy data reconstructed from deep-sea oxygen isotope analysis, while the future projection from climate model is of high uncertainty owing to the lack of any validation.

3) Whereas the historical or undergoing climate variability is transient, the Pliocene simulation is obtained after the model integration reaches a quasi-equilibrium state. The equilibrium response is in principle reversible while transient response is hysteretic, suggesting that the Pliocene simulation can better represent a steady climate response.

(2) The Pliocene is a long time period, and mid-Pliocene Warm Period (3.264–3.025 Ma) is selected for Pliocene simulation in PlioMIP2, which has been stated clearly in the revised version. The orbital forcing adopts preindustrial conditions because mid-Pliocene Warm Period (3.264–3.025 Ma) has a near-modern orbital forcing. The $CO_2$ concentration 400ppmv is set in the Core PlioMIP2 experiment, and other details of the experiment, i.e., PRISM4 Pliocene palaeogeography reconstruction including topography, bathymetry, ice sheets and land–sea mask, etc., are proposed in PlioMIP2. A brief description is specified in Section 2.1 of the revised version.

Specific comments:

Line 14: Define PRISM.
The definition of PRISM is added.

Line 24: "CO2" to "$CO_2$".
Done.

Line 24-25: This sentence seems to suggest the $CO_2$ increase is due to the monitoring. Please rephrase.
Changed to "As shown in the monitoring at …"

Line 25-28: Break up this sentence to two sentences and rewrite.
Done.

Line 26: "increased 1.1°C to" change to "increased 1.1°C compared to".
Done.

Line 31: "its reflectivity". Do you mean the sea ice reflectivity changes is responsible to changes in surface energy budget, etc.? Please rephrase to clarify.

Both the sea ice reflectivity changes and the reduction in the insulating effect of sea ice are responsible to changes in surface energy budget. The sentence has been rephrased for clarity.

Line 47-54: Some justification for Pliocene simulation is provided here. But the simulation in this study use mixed Pliocene and preindustrial conditions. Can you still use proxy data to verify the simulation? Some comparison with proxy data are shown in Fig. 1 but performance constraint by proxy data is still qualitative.

The Pliocene simulation is set following the PlioMIP2 protocol. The orbital forcing adopts preindustrial conditions because mid-Pliocene Warm Period (3.264–3.025 Ma) is specified and has a near-modern orbital forcing. Referring to the review of Haywood et al. (2016) that "The temporal focus for PlioMIP2 has been matched to the data collection strategy adopted by PRISM4 to further enhance the connection between models and data and to improve the validity of future data/model comparisons.", it is available to use proxy data (PRISM) to verify the simulation.

Line 146-147: It is not clear what "an approach" is referring to.

"an approach" has been specified.

Line 150: "linear": do you have any reason to believe it should be linear. Please provide some argument to support this assumption or reference previous studies to support your statement.

The heat flux depends not only on sea ice, but also on other variables, e.g., wind speed. In addition to considering the interaction at the interface, the heat flux is not strictly linear with sea ice. Therefore "linearity" assumed here is to investigate the linear impact of sea ice on the heat flux. The following correlation analysis can be used to evaluate to what extent it can be linear, e.g., the high correlation coefficient between shortwave radiation change due to albedo and sea ice concentration change indicates the albedo effect of sea ice on shortwave radiation is nearly linear.

Line 162: The choice of student-t test assumes Gaussian distribution. However, it is usually not the case for geophysical parameters. The spatial and temporal autocorrelation are not taken into account. The bootstrap method adopted by Liu et al. (2012) is more relevant, especially when "moving-block" is used to represent the length scale of the spatial or temporal features (van de Poll et al., 2006; Z. Liu et al., 2010).

The spatial correlation and student-t test applied in the study are based on the effective number of spatial degrees of freedom (Bretherton et al., 1999), in which non-normally distributed data could be estimated and the spatial autocorrelation was

taken into account. Moreover, Bretherton et al. (1999) also found close agreement between the actual PDF and a fitted distribution based on Monte Carlo simulation with the effective number of spatial degrees of freedom.

Line 174-179: It is not clear what message these sentences are trying to convey. Is the Arctic amplification stronger in Pliocene simulation or not?
These sentences are written to introduce the Arctic amplification in Pliocene based on proxy data, as well as some explanations for the stronger Arctic amplification in Pliocene. Yes, the Arctic amplification is stronger in Pliocene simulation.

Line 180: Not exactly the SST and SAT in "Pliocene epoch". They are from the Pliocene simulations using some of the Pliocene conditions.
The Pliocene simulation is a Core PlioMIP2 experiment, and the temporal focus for PlioMIP2 is the mid-Pliocene warm period, so "Pliocene epoch" is changed to "the mid-Pliocene warm period".

Fig. 1: The overlay of proxy data over the filled contour maps does not show the difference well. An additional map showing the differences might be more informative. Maybe add it to supplement?
Thanks for your suggestion. The difference of annual mean SST anomaly (Pliocene minus preindustrial) between EC-Earth simulation and the proxy data is shown in Figure S2. The underestimation in high latitude regions, similar to multi-model mean in Dowsett et al. (2012) but significantly less in bias.

Line 187: "off America". Do you mean USA or the America continents? The coastal warming is not very obvious or extensive in either. The warming off the west coast of Africa is more obvious. By "permanent El Nino-like feature", I expect to see extensive warming in equatorial eastern Pacific, which is not present in Fig. 1. Am I missing anything?
The comment is no longer applicable as the description of the tropical anomalies has been removed because the paper's focus is on the Arctic response (as suggested by Reviewer #4).

Line 197: "very sensitive" to what? "preindustrial" is used noun here and in several other occasions. Please add proper noun after it so that it is clear what it is referring to.
Referring to the comment of Reviewer #4 ("The first line of this paragraph contains little meaning"), we have removed the sentence, thus "very sensitive" is no longer applicable. A proper noun is added after "preindustrial" throughout the paper.

Line 200-201: Please rephrase this sentence.

The sentence has been rewritten.

Line 201: The definition of heat loss or heat gain is not given here. Consider move up the definition in line 204-208 before they are discussed.

The definition of heat loss or heat gain has been moved up as suggested.

Fig. 5: What are these dots? Are they differences between corresponding years in Pliocene and preindustrial runs? Does it matter how you pair up the years in Pliocene and preindustrial runs? Would it change the response coefficients? How much uncertainty would it introduce?

In this study the last 100-year-mean of all variables are used for analysis, and the Pliocene climate anomalies are calculated by subtracting the mean of the preindustrial simulation without trends removal. Thus each dot in this figure represents one grid point value of the Pliocene sea ice concentration anomaly and shortwave anomaly due to albedo over the Arctic Ocean, which is specified in the revised version for clarity. They are not the differences between corresponding years in Pliocene and preindustrial runs, and it doesn't matter how you pair up the years in Pliocene and preindustrial runs. The response coefficients depend on the linear regression coefficient of these dots. The uncertainty can be expressed as a confidence interval, i.e., 1.96 standard errors either side of the estimate.

Line 242: How do you define "anomalies"? Are they with respect to the 100-year Arctic averages, Arctic monthly mean, or something else? Please describe clearly. Otherwise, it is impossible to interpret Table 1 to Table 4.

The "anomalies" are defined in Section 2. (In this study the last 100-year-mean of all variables are used for analysis, and the Pliocene climate anomalies are calculated by subtracting the mean of the preindustrial simulation without trends removal.) Thus here they are with respect to Arctic monthly mean while discussing seasonal variation, and this is specified in the revised version.

Line 252-253: This sentence is confusing. Please rewrite.

The sentence is rewritten.

Line 263: Again, do you have any reason to believe they should be linear?

As mentioned above, the heat flux depends not only on sea ice, but also on other variables. In addition to considering the interaction at the interface, the heat flux is not strictly linear with sea ice. Therefore "linearity" assumed here is to investigate the

linear impact of sea ice on the heat flux. The following correlation analysis can be used to evaluate to what extent it can be linear, e.g., the high correlation coefficient between shortwave radiation change due to albedo and sea ice concentration change indicates the albedo effect of sea ice on shortwave radiation is nearly linear.

Line 274: What do you mean by "significant interaction"?
The phrase "significant interaction" is unclear and is changed to "linear relationship".

Line 276-284: The partition of ice-free and ice-covered conditions is confusing. How do you have SIC changes when it is ice-free?
The partition threshold of ice-free and ice-covered conditions is 15% sea ice concentration as commonly used in sea ice study, which is specified in the revised version. The "ice-free" shown in Figure 9 represents that the sea ice of the grid points varies from the preindustrial ice-covered status to the Pliocene ice-free status, so SIC change can be calculated as the SIC difference between the Pliocene and the preindustrial period.

Fig. 9: Is the regression line with respect to the ice-covered only? Would the regression line be different between ice-covered and ice-free? You would expect so because it is why you separate them in the first place.
Yes, the regression line is with respect to the ice-covered only, and it is specified in the figure caption. It can be inferred from Figure 9 that the regression line is different between ice-covered and ice-free conditions.

Line 289-293: I am confused with the definition of ice-free and ice-covered changes relative to SIC. So I cannot follow the argument here. I will have to revisit after the authors' clarification.
The paragraph intends to explain the linear regressions of sensible and latent heat flux anomalies on SIC, including slope and intercept. As mentioned above, the ice-free (ice-covered) shown in Figure 9 represents that the sea ice of the grid points varies from the preindustrial ice-covered status to the Pliocene ice-free (ice-covered) status.

Line 297-299: Does the melting of sea ice refer to the sea ice melt in spring or due to warming by high $CO_2$? Are you sure it is conduction not convection?
The melting of sea ice refers to the sea ice melts due to warming by high $CO_2$, which is specified in the revised version.
We think that conduction moves the heat from sea/ice to the atmosphere at the surface and then convection moves the heat upward to higher level of atmosphere.

[revised manuscript text omitted]

**Figure 2.** Spatial distributions of the annual mean sea ice concentration (SIC) and  net heat flux at the surface of ice and ocean (Wm$^{-2}$, positive downward) over the Arctic Ocean. (a) SIC in the preindustrial period, (b) SIC in the Pliocene, (c) the Pliocene SIC change with respect to the preindustrial period, (d) net heat flux in the preindustrial period, (e) net heat flux in the Pliocene, and (f) the Pliocene net heat flux change with respect to the preindustrial period. The diagonal stripe in (c) represents the regions from ice-covered to ice-free, and the diagonal crosshatch represents the regions from ice-covered to ice-covered.

640

[Figure]

645 **Figure 3.** Spatial distributions of the Pliocene annual mean heat flux change (Wm$^{-2}$, positive downward) with respect to the preindustrial period over the Arctic Ocean. (a) net shortwave radiation flux, (b) net longwave radiation flux, (c) sensible heat flux, and (d) latent heat flux.

650

[Figure]

**Figure 4.** Spatial distributions of the Pliocene annual mean net shortwave  flux change (Wm$^{-2}$, positive downward) at the surface
655   over the Arctic Ocean caused by albedo effect of sea ice change with respect to the preindustrial period.

660

[Figure]

[Figure]

**Figure 5.** The annual mean net shortwave  flux change (Wm$^{-2}$, positive downward) caused by the albedo effect of sea -ice change averaged over the Arctic Ocean as a function of SIC change. All the change is with respect to the preindustrial period, and each dot represents one grid point value over the Arctic Ocean.

670

[Figure]

[Figure]

675 **Figure 6.** The monthly response coefficients ($Wm^{-2}$) of net shortwave  flux to the albedo effect of sea ice.

680

685

[Figure]

**Figure 7.** Spatial distributions of the Pliocene annual mean radiation fluxes change (Wm$^{-2}$, positive downward) with respect to the preindustrial period over the Arctic Ocean. (a) shortwave radiation due to cloud change, (b) longwave radiation due to cloud change, (c) shortwave radiation due to water vapour change, (d) longwave radiation due to water vapour change. Here the, cloud and water vapour change is specified asthe value before removing the part caused by sea-ice decreaseremote effects of clouds and water vapour.

690

695

[Figure]

[Figure]

**Figure 8.** The annual and monthly response coefficients (Wm$^{-2}$) of  shortwave and longwave radiation flux related to cloud and water vapour change to the insulation effect of sea -ice. Here, the cloud and water vapour change is specified as the part related to sea -ice decrease.

[Figure]

710

**Figure 9.** The annual mean sensible and latent heat flux change (Wm⁻², positive downward) related to insulation effect of sea -
ice change averaged over the Arctic Ocean as a function of SIC change. Pliocene changes shown are computed
relative to the preindustrial simulation. The ice-free and ice-covered regions here refer to the diagonal hatched and cross-hatched regions
in Figure 2c, respectively. The blue line is the linear regression on the ice-covered scatter points, and the response coefficient (λ) and

715 correlation coefficient (r) are just for the ice-covered areas.

720

[Figure]

725 **Figure 10.** The monthly response coefficients (Wm$^{-2}$) of sensible and latent heat fluxes to the insulation effect of sea -ice.

**Supplementary Information**

[Figure]

**Figure S1.** The global annual mean of last 200 model years output in the Pliocene simulation (The negative TOA net radiation represents a heat loss of the earth-atmosphere system.)

[Figure]

**Figure S2.** The difference of annual mean SST anomaly (Pliocene minus preindustrial, K) between EC-Earth simulation and the proxy data at 95% confidence-assessed marine sites from Deep Sea Drilling Project (DSDP) and Ocean Drilling Program (ODP).

[Figure]

**Figure S3.** The difference of annual mean low cloud cover (a) and high cloud cover (b) anomaly in Pliocene with respect to the preindustrial.

---

## Author Comment (AC4) · 5 Sep 2018

**Point-to-point response to reviewers' comments**

*The comments are in black, and our answers are in blue.*

**Reviewer #4:**

General Comments:

I recommend this work being accepted only after major revisions and rewriting for improved English composition. The late Pliocene is a unique warm period, representing a climate in equilibrium to modern levels of atmospheric greenhouse gas concentrations. This study makes a contribution to understanding the impact on sea ice response to Arctic amplification in a past warm climate state for which there is a large body of climate proxy data and other model studies. Most of the results shown here concerning the decomposition of the effects of sea-ice change on Arctic amplification are not new, as discussed in Serreze and Barry (2011), though the methodology, of using CFRAM may be new to this particular application. As this work pertains to a Pliocene simulation that compares well to the proxy-reconstruction, it would make a good contribution to the literature. More details of the CFRAM methodology used in this application would be beneficial to the readers.

We are grateful for the overall positive evaluation and detailed comments that follow. Below are our responses and we have revised the manuscript accordingly.

Specific comments:

This manuscript needs to be thoroughly edited by someone with more English proficiency in order to improve the grammar and many awkward sentence structures. The result would be greater clarity for understanding the methodology and results. As written, many sections are not written with the necessary clarity for communicating the authors' intent. I point out some examples in my detailed comments by line number, below.

A native English speaker has proofread the revised manuscript. Thanks a lot for your detailed comments! All these comments have been taken into account in the revised manuscript.

A TOA net energy imbalance of about -0.5Wm$^{-2}$ (Fig. S1) is large and suggests the simulation is not at a near equilibrium. In addition, the weak negative trend in the TOA energy imbalance suggests the model is moving away from equilibrium. Is this not globally integrated? A net loss of energy at the top of the model is inconsistent with both a positive SST trend and a negative sea ice concentration trend, both of

which suggest the model is warming, unless a negative TOA flux is directed downward. This figure needs more explanation.

Strictly speaking, the TOA net radiation should balance after reaching an equilibrium. However, a small imbalance generally remains associated with numerical errors, such as -1.5 $Wm^{-2}$ displayed in ERA-Interim (Hazeleger et al., 2011) and 0.9 $Wm^{-2}$ shown in Trenberth et al. (2009). From our last 200 years output in the Pliocene simulation, the mean TOA net radiation (globally integrated) is about -0.5 $Wm^{-2}$ and its trend is near zero. The trend of mean SST is about 0.02 K/century, which fulfils the PMIP4 equilibrium criterion that the trend of mean SST should be less than 0.05 K/century (Kageyama et al., 2018). The negative TOA flux is directed upward, meaning a heat loss of the earth-atmosphere system, which seems inconsistent with a positive SST trend and a negative sea ice concentration trend. However, the positive net surface energy fluxes (also shown in Hazeleger et al., 2011) can explain the inconsistency. For clarity, the explanation about the direction of energy flux is added in the revised text and Figure S1.

Overall, there is a general lack of specific details of the analyses presented. For example, how are the anomalies computed, i.e. are trends first removed? A more physical explanation is needed for why you are using spatial correlations because as written, "limitation of data and computation" is quite vague. More detail is also needed to explain how CFRAM was applied to the surface energy balance in the Arctic. For example, what is the "first part of CFRAM" (line 143) used to obtain the surface radiative fluxes. Some equations would be useful, as would a citation to the same specific use of CFRAM as applied here.

More detail of the analyses presented is specified in the revised version. The last 100-year-mean of all variables in preindustrial control run and Pliocene sensitivity experiment are used for analysis, and the Pliocene anomalies are computed by subtracting the mean of the preindustrial simulation without trends removal. As the CFRAM calculation of high temporal resolution, such as 6-hourly or daily, is computationally expensive, monthly data are used in the analysis. However, the monthly resolution is too coarse to explain the relationship between heat fluxes and sea ice concentration by temporal correlations. Therefore spatial correlations are calculated. More detail including a partial radiative perturbation equation is added in the CFRAM introduction and the "first part of CFRAM" is specified in the revised version.

More detailed comments by line number: (Note: this is not a complete list of every grammatical issue in the text.)

11: "current warming climate" suggests transient climate change.

Replace "analogue" with "equilibrium state".

14: Define PRISM and give a citation.
Done.

17: "Given the facts…"
Done.

19: Run-on sentence structure.
Done.

20: During winter months…
Done.

29: Either "…in the recent decade…" or "…in recent decades..." "Moreover, an ice-free Arctic…"
Done.

31: "As the sea ice retreats…" Isn't it the average reflectivity of the surface that decreases due to a decrease in the fractional sea ice coverage and a decrease in the sea-ice albedo due to melting snow and ice. Are these two effects separated out here? For improved clarity it might be better to say "As sea ice retreats, the surface Arctic Ocean becomes less reflective and the enhanced open ocean region leads to greater air-sea heat exchange due to the reduction in the insulating effect of sea ice." However, as sea ice melts, its reflectivity changes and as the sea ice concentration changes the surface albedo is impacted. Both of these changes affect the net shortwave radiation at the surface.
The sentence was rewritten as suggested. The decrease in the sea-ice albedo due to a decrease in the fractional sea ice coverage is the focus in this paper, and the decrease in the sea-ice albedo due to melting snow and ice is mentioned in section 4. These two effects are not separated out here, and it can be found in section 4 that the former is dominant when they affect the net shortwave radiation at the surface.

"This leads to changes in the surface heat budget and changes in…"
Done.

33: "…possibly results from…"
Done.

38: "…consequence…has been reviewed"
Done.

46: shown
Done.

57: First "attributions" should perhaps be "characteristics" or "properties". Second "attributions" in line 59 should perhaps be "effects" or "mechanisms" here.
Done.

59: Run on sentence. I suggest cutting into two separate sentences or using a semi-colon instead of the comma after "…atmosphere and ocean"
Done.

69: should be "effects"
Done.

78: "represents"
Done.

79: Should amend "future climate at equilibrium with modern ghg levels"
Done.

138: "temperature"
Done.

143: "first part of CFRAM" is vague. More details on how CFRAM is applied should be given here.
More details on how CFRAM is applied have been given here.

169: Replace "present" with "modern", omit "benthic,"
Done.

170-174: Make a stronger statement linking the Arctic amplification statement starting on line 174, because that is the focus in this paper. To do this, I suggest rewriting here. I also suggest that you omit describing the tropical anomalies mainly because it is irrelevant to this work. Also, see Scroxton et al., Paleoceanography (2011); Brierley, PAGES News, 21(2), (2013);

Watanabe et al., Nature, 471, 209-211, (2012); for recent papers discussing evidence for a robust Pliocene ENSO.

Rewrite the sentences and remove the description of the tropical anomalies as suggested.

176-177: "…even though they have comparable CO2 concentration…" replace with "…despite comparable CO2 concentrations…" I also suggest breaking this sentence into two after the Ballantyne citation. Then in the next sentence suggest possible reasons why there is enhanced Arctic warming or amplification compared to today. I would add these newer citations for the amplified response to closed gateways: Otto-Bliesner et al. GRL, 44, 2017 and Feng et al. EPSL 466, 2017.

The sentences have been rephrased as suggested and the newer citations are added.

187: Omit the brief, sentence starting in line 187 with, "Meanwhile…" because the paper's focus is on the Arctic response.

Done.

192: "region" should be "regions" and omit "but they are apparently" for improved conciseness.

Done.

194: "Noteworthily…" Awkward. A potential replacement is "Notably"

Replace "Noteworthily" with "Notably".

194-195: Suggest replacing "…SAT, and the maximum…" with "SAT; the maximum…" using a semi-colon to join the two clauses instead of the conjunction "and".

Done.

197: The first line of this paragraph contains little meaning.

Remove the first sentence of this paragraph.

204-208: An equation for the net air-sea heat flux with the components presented symbolically is written in equation (4), but then these symbols are never used again, and reference to (4) is never made. I suggest removing the equation and stating this decomposition elsewhere. Also, what about the ice-ocean heat flux? A surface heat budget for the Arctic ocean should include the ice-ocean heat fluxes associated with the freezing and melting of sea-ice.

The equation for the net air-sea heat flux has been removed as suggested. And the decomposition is stated before the illustration of four flux terms in Figure 3. The ocean-atmosphere and ice-atmosphere interface heat exchanges are concerned in this study to understand the impact of sea-ice on Arctic amplification. The ice-ocean heat flux can affect the freezing and melting of sea-ice and then affect Arctic amplification indirectly, which would be investigated in the further study.

209-218: This paragraph discusses the anomalies or differences in the heat flux components in the Pliocene simulation as compared to the Preindustrial simulation, not heat fluxes. Every mention of a flux in this paragraph should reflect the fact that what is discussed are differences in the flux.
The word "anomalies" is added for clarity.

210: "The radiative and turbulent heat fluxes…" These are differences or anomalies.
The word "anomalies" is added for clarity.

211: "…the positive shortwave radiation is dominant…" should be "…the positive change in the shortwave radiation is dominant…"
Done.

212: "On the contrary" should be "In contrast"
Done.

220: "accounted as the synergy…" Awkward phrase. Suggest a rewrite of the first paragraph of this section for better clarity and conciseness. The definition of albedo needs to be more precise, to distinguish from planetary albedo.
The first paragraph of this section including the definition of albedo has been rewritten.

224: relevant to net shortwave…
Done.

225: Be more specific: net shortwave flux at the surface… Also, is this due to changes in sea-ice and snow albedo (that is, changes in the albedo due to changes in the state of sea-ice or snow, or to the change in albedo over the ocean grid box due to both albedo change and change in sea-ice concentration?
The word "net" is added. The albedo effect here is due to the combination of various albedo changes, including melting snow or ice and change in sea-ice concentration.

226-227: "…most…shows…"
Done.

227: net shortwave
Done.

229: changes in sea ice extent
Done.

231: changes in snow cover … (and, I presume, any change affecting the actual sea-ice albedo which could be changes in the sea-ice or snow state, such as melting, because the albedo is a function of the sea-ice state as well as thickness according to earlier descriptions of LIM.)
Done. We agree and revise the possible causes in the manuscript.

233: net shortwave
Done.

234: "Regarding the…" This opening sentence is awkwardly stated and vague.
The sentence is revised.

235: *net* shortwave radiation, i.e. shortwave radiation absorbed?
The word "net" is added. Shortwave radiation are defined positive downward, i.e. shortwave radiation absorbed.

238: "The prominent oceanic heating in May and June seems inconsistent with the maximum SST warming in August,…" Second clause of sentence seems to explain why the SST warming lags the SW heating response, thus it is not "inconsistent" but "consistent."
We agree and replace "inconsistent" with "consistent".

240: About the SIC anomalies: Shouldn't this be something "like the mean spatial variance over the Arctic of the Pliocene SIC anomalies is not strongly variable over the mean annual cycle." This is a curious result. A nice additional supplemental figure could show the Pliocene anomaly pattern at the minimum and maximum of the SIC annual cycle.
The SIC anomalies show different patterns at the maximum and minimum of the SIC annual cycle as the figure below. However, the relative rate of change of spatial variance over the Arctic of the Pliocene SIC anomalies has been checked and

demonstrated to be less as compared with that of net shortwave radiation anomalies due to albedo effect.

[Figure]

243: "our correlation analysis indicates that…
Done.

246: Needs to be more specific: …seasonal cycle of incident shortwave or net shortwave? …sea ice concentration variation, or some other sea ice property variation?
Here it's "seasonal cycle of net shortwave". It should be pointed out that the seasonal cycle of net shortwave depends not only on incident shortwave, but also on sea ice concentration variation and some other factors that can change surface albedo.

250: …insulation effect of sea-ice…
Done.

251: omit "In fact,"
Done.

251: "insulation effect"
Done.

250-255: This first paragraph should be rewritten. The insulating effect of sea-ice has an indirect effect on the net surface shortwave and longwave fluxes. By separating the overlying atmosphere from the ocean, sea-ice reduces evaporation from the ocean resulting in a decrease of water vapour and cloud cover. This reduction plays a non-negligible role in the amount of downward shortwave and longwave radiation

reaching the surface. However, remote moisture transport also affects water vapour and cloud amount. Thus, …

Thanks! The paragraph has been rewritten in the revised version as suggested.

256:262: It is not clear in this paragraph whether the discussion is about the SW and LW feedbacks after the remote effects on clouds and water vapour have been removed. This is suggested in the Figure 7 caption however.

The phrase "before removing the remote effects on clouds and water vapour" has been added in the manuscript and the Figure 7 caption for clarity.

258: …cloud characteristics…

Done.

263:274: Then this paragraph discusses just the local effect due to changes in sea ice concentration? I don't know what is meant by "counterpart of sea-ice insulation."

Yes. The phrase "counterpart of sea-ice insulation" is not clear, so it is changed to "the local insulation effect due to changes in sea ice concentration".

264: "Like the steps performed to isolate the albedo effect…"

Done.

266: "In the annual mean…"

Done.

270: "…shows a pronounced…"

Done.

271: "Compared to…" And this is being compared to the standard deviation of the shortwave anomalies due to clouds ? Also, should be "SIC anomalies" and

Done. Yes, "the standard deviation of" and "associated with local SIC anomalies" have been added for clarity.

272: Net shortwave radiation change and net longwave radiation change?

The word "net" is added.

274: …when there is a lack of…

Done.

279: ice-free conditions

Done.

280: the insulation effect
Done.

280: …and differentiate fluxes from ice-covered versus ice-free areas, not
"ice-covered fluxes"
Done.

281: displays the Pliocene anomalies in …heat fluxes…as a function of SIC
anomalies.
Done.

282: There is a larger spread in the turbulent heat flux anomalies over the ice-free area
(grey symbols, corresponding to the diagonal hatched region in Figure 2c) than
compared to anomalies from the ice-covered areas (light blue symbols, cross-hatched
region in Fig. 2c) because the former is free from the constraint of sea-ice.
Done.

284: …and changes in SIC…
Done.

285: Are these estimates of variance explained from the regression lines shown in
Figure 9? Is this and the response coefficient shown in the figure just for the
ice-covered region? Be specific in both the Figure caption and the main text.
Yes, these estimates of variance explained are square of correlation coefficient shown
in Figure 9. Yes, they are just for the ice-covered region, which has been specified in
the figure caption and the text.

286:293: This paragraph jumps all over the place and is very unclear. First it discusses
annual mean response coefficients vs. trends elsewhere, to the y-intercept of the
regression line, then jumping to explaining seasonal variation.
This paragraph has been rewritten for clarity. The paragraph intends to explain the
linear regressions of sensible and latent heat flux anomalies on SIC, including slope
and intercept. Trends elsewhere are specified in the text, which refers to the different
variability of turbulent heat fluxes over different latitude regions. Here "cold season"
means the turbulent heat transport is the most pronounced in winter and can determine
the overall value of annual turbulent heat fluxes, and we do not intend to explain the
seasonal variation.

287: Noteworthily—a better choice would be "Notably" as mentioned previously.
Done.

288: "trend of sensible heat flux" this comes out of nowhere, to what does this refer? Is this "trend" referring to 20th century trends observed? Describe accurately.
The "trend" is specified in the revised version as the trend of turbulent heat flux over the low- and mid-latitude North Pacific and North Atlantic oceans from 1984–2004.

289: …turbulent flux anomaly axis?
Done.

290: "even without SIC change" for improved conciseness.
Done.

294: …to the sea-ice concentration? Or to sea-ice changes in general (thickness, albedo, concentration, etc.)? Also, replace "two" with "the".
The phrase "to the sea-ice concentration" is fine here. Replaced "two" with "the".

295: "…have a similar…" and "…showing a negative response…"
Done.

296: "…maximum warming of SAT occurs in November as a consequence…" It looks like changes in the net LW due to the response of clouds and water vapour is also a contributing factor to the warming in fall. As a complete budget for SAT is not presented, it would require adding heat transports and other fluxes, one can only suggest contributing factors.
The net longwave radiation change in response to cloud and water vapour is attributed to downwelling longwave radiation as upwelling longwave radiation depends solely on the surface temperature according to the Stefan–Boltzmann law. The more downwelling longwave radiation is in favour of the warming in SST rather than in SAT, therefore changes in the net LW due to the response of clouds and water vapour is not a contributing factor to the warming in fall, which is demonstrated by the opposite sign of the response coefficients in net LW radiation and turbulent heat fluxes. The phrase "contributing factor" is more appropriate than "as a consequence", the sentence is rewritten in revision.

Section 6 Summary and Discussion;

This section is mostly a summary of results. Additional discussion could compare these results to previous results (see Serreze and Barry, 2011), could compare to the other Pliocene simulations which showed weaker Arctic amplification, highlight what is new here, etc.

A paragraph to compare these results to previous results and the other Pliocene simulations is added in Section 6.

304-309: Paragraph should be rewritten as it contains many awkward phrases. Also, I disagree that a model ever reveals a complete picture, but a model may be applied to investigate mechanisms and processes that help in understanding.

The paragraph has been rewritten. We agree that a model can not reveal a complete picture and revise it as suggested in the text.

312: "…the effects of changes in"

Done.

314: "…expected to partly interpret the variability of heat flux" Very unclear as to what this is supposed to mean.

The sentences are rewritten for clarity. Here we would like to highlight that albedo and insulate effects of sea ice can only partly explain the mechanism of Arctic amplification and a complete energy budget is required for a full understanding of Arctic amplification.

315-328: This paragraph appears to summarize the albedo and the insulation effects of sea ice on surface heat fluxes over the annual cycle, but doesn't seem to say anything about the Arctic amplification noted in the comparison of the Pliocene to preindustrial climate simulations. This section needs to be more specific.

More descriptions associated with Arctic amplification are added, and anomalies (in the Pliocene as compared to the preindustrial) are specified for clarity.

326: sea ice decline…Is this the decline of Pliocene sea ice as compared to the preindustrial, or over a seasonal cycle? It is not clear whether anomalies are being discussed. Also, "accelerates" should be "amplifies" as "accelerates" suggests time evolution, and here equilibrium runs are being discussed.

"sea ice decline" here refers to the decline of Pliocene sea ice as compared to the preindustrial, which is specified in the text. Replace "accelerates" with "amplifies".

Comments on Figures:

2) Please make the hatching in 2c more visible with another color or thickness. Be specific about describing the heat flux. "Net heat flux at the surface" Is this net heat flux at the surface (ice and ocean), or net heat flux at the surface of the ocean (air-sea and ice-ocean interfaces)?

The hatching is made more visible by changing its thickness.
The net heat flux is at the surface (ice and ocean) and is specified in the text and figure caption.

3) State when the flux is a "net" flux change, that is for the sw and lw fluxes.
The word "net" is specified.

4) Be more specific. Does the figure show the change in mean annual net shortwave flux at the surface?
Yes. The phrase "at the surface" is added.

5) and 6) net shortwave flux, also in 5) and 9) "All changes are" or "All change is"
Done.

7) More clarification is needed in the figure caption. Also, "caused by" should be "related to", because causality is difficult to attribute in feedback processes.
The phrase "before removing the remote effects on clouds and water vapour" has been added in the figure caption for clarity.

8) Specify "net" again. Also, "caused by..." should be "related to…".
Done.

9) Define ice-free vs ice-covered regions here referring to Fig. 2c. Also…"Pliocene changes shown are computed relative to the preindustrial simulation." Describe the regression lines, i.e. which set of scatter points are being regressed. "caused by" should be "related to".
Done.

S1) Are all of these quantities global averages?
Yes. The phrase "global annual mean" is used.

[revised manuscript text omitted]

**Figure 2.** Spatial distributions of the annual mean sea -ice concentration (SIC) and  net heat flux at the surface of ice and ocean (Wm$^{-2}$, positive downward) over the Arctic Ocean. (a) SIC in the preindustrial period, (b) SIC in the Pliocene, (c) the Pliocene SIC change with respect to the preindustrial period, (d) net heat flux in the preindustrial period, (e) net heat flux in the Pliocene, and (f) the Pliocene net heat flux change with respect to the preindustrial period. The diagonal stripe in (c) represents the regions from ice-covered to ice-free, and the diagonal crosshatch represents the regions from ice-covered to ice-covered.

[Figure]

645 **Figure 3.** Spatial distributions of the Pliocene annual mean heat flux change (Wm$^{-2}$, positive downward) with respect to the preindustrial
period over the Arctic Ocean. (a) net shortwave radiation flux, (b) net longwave radiation flux, (c) sensible heat flux, and (d) latent heat
flux.

650

**Mean annual SW change due to albedo effect**

[Figure]

**Figure 4.** Spatial distributions of the Pliocene annual mean net shortwave  flux change (Wm$^{-2}$, positive downward) at the surface
655  over the Arctic Ocean caused by albedo effect of sea ice change with respect to the preindustrial period.

660

[Figure]

[Figure]

**Figure 5.** The annual mean net shortwave  flux change (Wm$^{-2}$, positive downward) caused by the albedo effect of sea -ice change averaged over the Arctic Ocean as a function of SIC change. All the change is with respect to the preindustrial period, and each dot represents one grid point value over the Arctic Ocean.

[Figure]

[Figure]

675 **Figure 6.** The monthly response coefficients (Wm$^{-2}$) of net shortwave  flux to the albedo effect of sea -ice.

680

685

[Figure]

**Figure 7.** Spatial distributions of the Pliocene annual mean radiation fluxes change (Wm$^{-2}$, positive downward) with respect to the preindustrial period over the Arctic Ocean. (a) shortwave radiation due to cloud change, (b) longwave radiation due to cloud change, (c) shortwave radiation due to water vapour change, (d) longwave radiation due to water vapour change. Here the , cloud and water vapour change is specified asthe value before removing the part caused by sea-ice decreaseremote effects of clouds and water vapour.

690

695

[Figure]

[Figure]

**Figure 8.** The annual and monthly response coefficients (Wm$^{-2}$) of  underline{net} shortwave and longwave radiation flux related to cloud and water vapour change to the insulation effect of sea -ice. Here, the cloud and water vapour change is specified as the part related to sea -ice decrease.

[Figure]

710

**Figure 9.** The annual mean sensible and latent heat flux change (Wm$^{-2}$, positive downward) related to insulation effect of sea -
ice change averaged over the Arctic Ocean as a function of SIC change. Pliocene changes shown are computed
relative to the preindustrial simulation. The ice-free and ice-covered regions here refer to the diagonal hatched and cross-hatched regions
in Figure 2c, respectively. The blue line is the linear regression on the ice-covered scatter points, and the response coefficient (λ) and
715  correlation coefficient (r) are just for the ice-covered areas.

720

[Figure]

725 **Figure 10.** The monthly response coefficients (Wm$^{-2}$) of sensible and latent heat fluxes to the insulation effect of sea -ice.

**Supplementary Information**

[Figure]

**Figure S1.** The global annual mean of last 200 model years output in the Pliocene simulation (The negative TOA net radiation represents a heat loss of the earth-atmosphere system.)

[Figure]

**Figure S2.** The difference of annual mean SST anomaly (Pliocene minus preindustrial, K) between EC-Earth simulation and the proxy data at 95% confidence-assessed marine sites from Deep Sea Drilling Project (DSDP) and Ocean Drilling Program (ODP).

[Figure]

**Figure S3.** The difference of annual mean low cloud cover (a) and high cloud cover (b) anomaly in Pliocene with respect to the preindustrial.

---

## Author Response (AR1)

**Editor Decision: Reconsider after major revisions** (20 Sep 2018) by Ran Feng

Comments to the Author:

Dear Authors,

Thank you for taking the initiative to improve the manuscript. I can see clear improvements to the clarity of the analysis and statistic robustness. Based on your current responses, I cannot help notice a few things waiting to be better addressed in your revised manuscript:

**Response:** We thank the editor for the thoughtful and constructive comments.

1. Reviewers pointed out that the simulated sea ice is low at 400 ppm $CO_2$ compared to published modeling studies. In your reference list, the Koenigk et al., (2013) paper suggested that for EC-Earth, September sea ice free threshold is around 500 ppm $CO_2$. In RCP4.5 and 2.6, EC-Earth did not reach September sea ice free by year 2100. I tend to agree with reviewers, a comparison with present-day control, and explanation of the differences are needed to validate your PlioMIP2 run.

**Response:** In this study we focus on understanding the warming in mid-Pliocene by comparing with the pre-industrial condition as the reference state. Following mid-Pliocene protocol, we setup our experiment as Pliocene-4-Pliocene by changing the geographical conditions, but not for Pliocene-4-future that keeping the present geographical configuration. We agree with the editor that the simulations of the warm mid-Pliocene period will provide more social relevant implications to run a Pliocene-4-future experiment, as it is regarded as an analogue to future scenario. We plan to do so in the future research using a CMIP6 model version. For current work we keep our focus on simulated mid-Pliocene climate.

We also wish to mention that the model version used for our simulations is EC-Earth 3.1, which differs from the EC-Earth CMIP5 version 2.3 as both the atmosphere model and ocean model as well as sea-ice are updated. The EC-Earth 2.3 was too cold and the cold biases are reduced in the new model. We will re-examine the sea-ice change in Arctic in new RCP4.5 and 2.6 with EC-Earth CMIP6 version.

2. Reviewers pointed out potential numeric errors in the model integration. One review noticed that this error may lead to global changes in atmospheric lapse rate. This sounds quite alarming. Please examine the lapse rate and explain whether or not this relates to different results from EC-Earth present-day and RCP runs. I wanted to point out that ERA-interim or any reanalysis data are using models to fit data, the priority of reanalysis model system is not to conserve energy. Scientific literatures were published to alarm the community not to use reanalysis blindly to perform energy balance analysis. In general, you cannot compared the energy imbalance of reanalysis data to performance of a model. A model, by design, should conserve energy.

**Response:** Per your suggestion, we evaluate the climatological mean and global mean of air temperature and its trend in the last 200 model years of the Pliocene run. The trend of the global

mean air temperature (right panel) shows a weak positive trend in the troposphere and a weak negative trend in the stratosphere. Because of the smallness of the trend of the global mean air temperature (in the range of +/- 0.04 degree per century), plus its nearly uniform vertical structure in the troposphere (meaning little changes in the global mean lapse rate), the lapse rate feedback is thought to play little role in causing "different results from EC-Earth present-day and RCP runs".

[Figure]

[Figure]

The issue of TOA and surface net radiation imbalance had been discussed in the technical report (Davini et al., 2014), which pointed out that the atmosphere loses radiation but does not cool, suggesting that the model has a "hidden" internal heating source of about 2.5 watts per square meter (in terms of global mean). It appears that the "hidden" internal heating source in the EC-Earth model is not sensitive to climate forcing, since the imbalance of the TOA radiative energy fluxes in the Pliocene run is about the same as that in the pre-industrial run. As a result, these differences in the TOA radiative energy fluxes between the two simulations actually is nearly balanced. For this reason, we think that the issue of energy imbalance at the TOA would little implication that would compromise our findings. The surface energy fluxes imbalance has been explicitly considered in our study as it is used to infer the oceanic heat uptake rate (if it is positive) or the heating release from the oceans to the atmosphere (when it is negative).

3. Reviewers pointed out that the time scale of interactions between ice and heat fluxes is below monthly. One reviewer is unsure about whether correlation at monthly scale can be used for causation arguments. Would it be possible to continue the run for another 30-yrs and run CFRAM code and spatial correlations at daily time scale? This will hopefully address

these concerns.

**Response:** We agree with the reviewers and Editor that "the time scale of interactions between ice and heat fluxes is below monthly" and it may be problematic to think "correlation at monthly scale can be used for causation arguments". We here merely apply the correlation/regression analyses to estimate the strength of various feedbacks that are "coupled" with ice melting. In particular, our correlation/regression analyses reported here are performed over (horizonal) space domain, instead of temporal domain (i.e., the correlations are evaluated from plots of A versus B at different grid points in a given calendar month). From such spatial correlation/regression analyses, one could not tell "who cause who", but infer the strength of change in A that is associated with B. Since our correlation/regression analyses are not performed over temporal domain, the temporal resolution in the data has no direct impact on the correlation/regression in terms of "degree of freedom" or "sample size".

Of course, one may argue that the monthly mean of spatial correlations using daily data may not have the same numerical value as the spatial correlations using monthly mean data. We believe that this would be a relevant question to ask when one uses such spatial correlations for phenomena at weather scales or short-time climate scales (less than 10 years). Recall that the fields before correlation/regression analyses are the differences between two equilibrium states (one is the "pre-industrial" state and the other is the "Pliocene" state) and each of the two equilibrium states is obtained by averaging the data over 100 years. In other words, the fields that go to our correlation/regression analyses are the differences between the climatological monthly annual cycles of the two equilibrium states. Should our correlation/regression analyses be made with the difference fields between the climatological daily annual cycles of the two equilibrium states, we would construct the climatological daily annual cycles by averaging daily mean data over 100 years at a given calendar day. Because (i) the climatological daily annual cycles of both equilibrium states are already very smooth fields and (ii) the differences between the climatological daily annual cycles of the equilibrium states are regarded as the seasonal cycle of the climate response to the climate forcing imposed to the system, day-to-day variation of such seasonal cycle within each calendar month is very gradual and smooth. In this sense, we don't expect the monthly mean of (spatial) correlations using daily annual cycle data would be different noticeably in terms of their numerical values from the (spatial) correlations using monthly mean annual cycle (which can be obtained by making monthly average of the daily annual cycle or constructing climatological monthly annual cycles from monthly mean data directly as both ways yield the same results).

Per your suggestion we have attempted to run CFRAM code at daily time scale, but it is not validated because the CFRAM is based on the energy balance of an atmosphere-surface column, and the balance is approximately maintained over the long term such as year, season and month.

4. The target time period of PlioMIP2 is mid-Piacenzian (at 3.205 Ma, belongs to the later part

of Pliocene). Pliocene epoch spans 5.3 to 2.6 Ma with varying $CO_2$, orbital parameters, and gateway configuration. The word "Pliocene" or even "mid-Pliocene" is inappropriate for discussing model results.

**Response:** The Pliocene Research Interpretation and Synoptic Mapping Project (PRISM) remains the only global-scale synoptic reconstruction of the Pliocene (Haywood et al., 2016), and PRISM data are concentrated on the warm interval (3.264-3.025 Ma). Therefore the time slice (3.264-3.025 Ma) is selected for Pliocene simulation in PlioMIP2, though the Pliocene epoch spans 5.33 to 2.58 Ma. We agree that the warm interval belongs to the late Pliocene. However, given that the mid-Pliocene Warm Period (mPWP) have been commonly used in most of the Pliocene studies, we continue using the word "mid-Pliocene" for consistency. A brief clarification has been added in Section 2.1 of the revised version.

Thank you! I am looking forward to the revised manuscript!

Davini, P., Filippi, L., and von Hardenberg, J.: Tuning EC-Earth from v3.01 to v3.1, Tech. Rep. 01/14, CNR-ISAC, UOS Torino, 2014.

[revised manuscript text omitted]

---

## Referee Report (RR1)

Zheng et al. has made substantial revision to their manuscript and improved the writing and the clarity of the presentation. I appreciate the authors' detailed reply to my comments. For example, the addition of the definition of "anomaly" is essential for the interpretation of Fig. 5 and Fig. 9. Some of my questions on the linearity of the attribution method and the Pliocene experiment setup are still not well explained. I suggest the authors to add an uncertainty estimate to the attribution method. The differences in boundary conditions between the preindustrial run and the Pliocene run will introduce perturbation fluxes not included in the attribution framework. The implication for current results should be considered. I recommend a substantial-minor revision of the current manuscript before it can be accepted for publication.

Major concerns:

1. The linearity of CFRAM

The author mentioned the "albedo and insulation interact in a nonlinear way" in line 72-73 and then apply an attribution method with linear assumption without giving clear estimate of the role of nonlinearity. The authors suggest the linearity of the CFRAM can be examined by comparing the total radiative perturbation and the sum of all the partial radiative perturbation terms. The provided figure has very confusing titles and does not seem to be related to the "total vs partial perturbation" comparison. Even if the figure is indeed a comparison of the total vs partial perturbations, it does not show the residual well. A comparison of the residual, total minus sum of partial perturbation, with the total is more informative of degree of nonlinearity. Maybe this can also be used as a measure of the uncertainty in the linearity assumption? Please show clear evidence to support the assumption of linearity and provide an uncertainty estimate of this assumption.

2. The boundary condition differences between preindustrial and Pliocene runs

I would like to thank the authors for providing clearer description of the Pliocene experiment setup. After reading the authors' clarification in the revision, I realized that it is important to note the boundary condition differences between preindustrial run and the Pliocene run. The PRISM conditions are not used in the preindustrial run. This choice is understandable for model evaluation purpose using proxy data. However, it does introduce additional uncertainties in your attribution analysis. The adoption of PRISM may introduce radiative perturbation that are not included in Eq (1). How important is this factor compared to albedo and insulation effects of sea ice?

Secondly, I understand that the authors followed the standard protocol in PlioMIP2, but a brief explanation of the justification of some setup choices are beneficial to readers who are not very familiar with the PlioMIP2. For example, the choice of equilibrium runs depends on the small $CO_2$ concentration and orbital forcing changes during this period.

Minor concerns:

1.  The Pliocene climate "anomalies"

It is very confusing to refer to the differences between the Pliocene run and the preindustrial run as "anomalies", which is usually used to describe the deviation from a climatological mean or spatial averages. I guess this is why another reviewer is asking whether the trend is removed when calculating the "anomalies". Please try to find another way to describe it or be sure to explicitly remind the readers that the "anomalies" is defined differently here.

2.  Tables.

This is just a suggestion. The tables listed these variables for all 12 months. It might be less intimidating and easier to digest the information if they are shown in plots instead, unless there are some particular numbers that are important to the readers. This is just my opinion though.

Specific comments:

Line 29-30: Something is not right about this statement. In addition, 2005 is not in the last decade.

Line 31: "ice-free Arctic Ocean" in summer or all year around?

Line 47-48: Is this a target for this study to produce a more significant Arctic amplification?

Line 52: It is not clear what the "sea ice effect" is referring to.

Line 68: Why does the sea ice thinning lead to enhanced insulation? This is confusing. Please rephrase.

Line 72-73: So nonlinearity is important. Does this contradict the choice of linearity for attribution later?

Line 118-120: This statement is confusing. It gives the impression that although it is actually late Pliocene but called mid-Pliocene because the mid-Pliocene is studied more. Do you mean this term "mid-Pliocene warm period" is frequently used for this period in literature although it is actually late Pliocene? Please rephrase.

Line 146: Again, does the differences in boundary conditions contribute in this equation?

Line 147: Is the "perturbation" here refer to the differences between the Pliocene run and the preindustrial run? Please clarify.

Line 208-213: As is mentioned by another reviewer, the constant SST under sea ice is also a contributing factor here. Please add it here.

Fig. 5: Because each dot represents a grid point in the model, then the latitude can be very important for the total SW insulation and the SW changes with albedo. Will you get a different

regression coefficient if subsetting the data by latitude? Can you calculate similar contribution to variance for the other factors in Eq (1)? Will they add up to 1?

Line 260: What is the "response coefficient of albedo to SIC"?

Line 324-327: Following Clausius–Clapeyron, in the colder Arctic, the saturating mixing ratio of water vapor is much smaller, so the latent heating and the responses are also smaller.

Line 336: I don't think it is conduction. Heat transfer by conduction does not rely on the wind speed or the amount of turbulence.

---

## Referee Report (RR2)

**Second review of "Contribution of sea-ice albedo and insulation effects to Arctic amplification in the EC-Earth Pliocene simulation" for *Climate of the Past*.**

**Summary:**

This manuscript examines the impact of sea ice changes on the surface air and sea temperatures in the Arctic during the Pliocene, as simulated by EC-Earth. Specifically, they examine the energy flux impact of differences in surface albedo and surface ocean insulation spatially correlated with the changes in sea ice between the Pliocene and pre-industrial eras. They found that a reduction in albedo allows for much stronger short-wave heating during May, causing the biggest SST difference between the two eras to be during August. They also found that this extra stored heat was released back to the atmosphere via enhanced surface heat fluxes due to the decrease in sea ice insulation. This resulted in SAT differences between the two eras being largest during winter, and with an inverse seasonal timing relative to SST.

**Paper recommendation:**

The authors have adequately responded to my concerns, both in the main document and in the supplement, and have substantially improved the manuscript. A few very minor issues and grammatical corrections still need to be made (which are listed below), but once they have been dealt with then I believe the paper will be ready for publication. Given this, I am recommending "**Accept after technical corrections**", and look forward to seeing the final, published version of the manuscript.

**Minor issues:**

1. What units are the "SIC change" values in for figures 5 and 9?

2. Line 312: I would replace "bulk heat transfer coefficient" with "bulk transfer coefficients", as the coefficient for sensible heat is not necessarily the same as the coefficient for evaporation/latent heat.

**Grammatical and cosmetic issues:**

Line 12: Replace "current warming climate" with "the current warming climate".

Line 30: Replace "by National Snow" with "by the National Snow".

Line 35: Replace "results from local" with "results from the local".

Line 52: Replace "sea ice effect" with "sea ice effects".

Line 59: Replace "affect climate system" with "affect the climate system".

Line 61: Replace "refer these two effects" with "refer to these two effects".

Line 64: Replace "Community Atmospheric General Circulation Model" with "Community Atmosphere Model".

Line 67: Replace "separate contribution" with "separate contributions".

Line 81: Replace "is preindustrial" with "is a preindustrial".

Line 96: Replace "In LIM3, surface" with "In LIM3, the surface".

Line 140: Replace "is based on TOA" with "is based on the TOA".

Line 144: Replace "is decomposing radiative perturbation into individual contribution" with "is decomposing the radiative perturbation into individual contributions".

Line 151: Replace "each factors" with "each factor".

Line 152: Replace "perturbation" with "perturbations".

Line 165: Replace "of sea ice" with "of the sea ice".

Line 181: Replace "of response" with "of the response".

Line 181: Replace "degree" with "degrees".

Line 192: Replace "during Pliocene" with "during the Pliocene".

Line 203: Replace "in EC-Earth" with "in the EC-Earth".

Line 246: Replace "that albedo" with "that the albedo".

Line 252: Replace "degree of freedom" with "degrees of freedom".

Line 267: Replace "with albedo effect" with "with the albedo effect".

Line 299: Replace "individual month" with "individual months".

Line 322: Replace "similar but different" with "similar, but not exactly the same".

Line 357: Replace "by annual" with "by the annual".

Line 362: Remove one of the two "indirectly" words from this sentence.

Line 364: Replace "to insulation effect" with "to the insulation effect".

Line 377: Replace "within Arctic" with "within the Arctic".

Line 550:  Replace "to albedo effect" with "to the albedo effect".

Line 597:  Replace "The diagonal stripe in (c)" with "The diagonal stripe in (c) and (f)".

Line 658:  Replace "vapour change to" with "vapour change due to".

Line 670:  Replace "to insulation effect" with "to the insulation effect".

---

## Author Response (AR2)

**Editor Decision: Publish subject to minor revisions (review by editor)** (19 Dec 2018) by Ran Feng

Comments to the Author:

Dear authors,

Thanks for taking time revising the paper. After two rounds of reviews, two out of three reviewers still hold strong opinion about the study. In particular, one reviewer questions of your application of CFRAM method and explanation of out of phase behavior in SST and SAT:

"The fundamental issue I have with this work is that the authors look at a 400ppm Pliocene run and conclude that the surface albedo feedback is stronger and hence contributes to Arctic amplification. This is not new. What is new here is the use of this CFRAM method, but it is not clear to me that the authors have applied this correctly over ice covered regions and open ocean regions. With regards to the seasonal cycle of SST/SAT, this can simply be explained by the fact that SST cannot get colder in winter than -1.8C, while the SAT does not get warmer than 0C in summer over ice. However, the SST can warm substantially in summer and the SAT can get much colder in winter. This by itself explains the out of phase behaviour. "

The present work pertains to a Pliocene simulation that compares well to reconstructed data, it might make a good contribution to the literature. The application of CFRAM method can take individual physical processes into account and quantify the contribution of sea ice. To avoid the contamination of open ocean regions, only the ice-covered regions are examined in the revised version.

The fact that SST cannot get colder in winter than -1.8℃, while the SAT does not get warmer than 0℃ in summer over ice can only explain the small SST change in winter and the small SAT change in summer. The complete explanation of the out of phase behavior of SST and SAT must solve the question why the maximum SST change happens in summer and the maximum SAT change happens in winter. The former is due to albedo effect of sea ice and the latter results from insulation effect of sea ice, and both are quantitatively analyzed in this study.

One other reviewer is unsatisfied with your explanation of linearity of the attribution method in particular the potential perturbation fluxes introduced by switching boundary conditions from PI to MPWP:

"Some of my questions on the linearity of the attribution method and the Pliocene experiment setup are still not well explained. I suggest the authors to add an uncertainty estimate to the attribution method. The differences in boundary conditions between the preindustrial run and the Pliocene run will introduce perturbation fluxes not included in the attribution framework. The implication for current results should

be considered."

1) The linearity of the attribution method

The figure below shows the total radiative perturbation (a), the sum of all the partial radiative perturbation terms (b) and the residual (c) over the Arctic Ocean. It appears that the total radiative perturbation and the sum of all the partial radiative perturbation terms are almost identical. The residual can explain about 2.6% of the variance of the total radiative perturbation. Therefore the nonlinearity might be negligible and it is reasonable to linearly decompose radiative perturbation into individual contribution with CFRAM. The validation of linearity is added in the text and the Supplement (Figure S4).

[Figure]

2) The Pliocene experiment setup

The application of the boundary conditions from PRISM may affect various climatic variables, such as temperature and precipitation, so the influence of boundary conditions can be captured in the changes of air temperature, cloud, water vapor and other factors. That is to say, the radiative perturbations in Eq (1) contain not only the effects of sea ice but also effects of the boundary conditions. In this study we focus on the effects of sea ice, which can explain part of the variance of the partial radiative perturbations in Eq (1). The residual variance includes the contribution of the boundary conditions and other factors, and it is difficult to obtain the contribution of the boundary conditions and compare with the effects of sea ice.

I lean towards accepting your work eventually, however, you do need to address the above concerns, as well as other detailed comments/suggestions by all three reviewers published in the discussion forum. Thank you!

We are very grateful for the positive evaluation! The point-to-point responses to the three reviewers' comments are attached.

I am looking forward to your response. Hope you have a great holiday!

Ran

**Point-to-point response to reviewers' comments**

*The comments are in black, and our answers are in blue.*

**Reviewer #1:**

The fundamental issue I have with this work is that the authors look at a 400ppm Pliocene run and conclude that the surface albedo feedback is stronger and hence contributes to Arctic amplification. This is not new. What is new here is the use of this CFRAM method, but it is not clear to me that the authors have applied this correctly over ice covered regions and open ocean regions. With regards to the seasonal cycle of SST/SAT, this can simply be explained by the fact that SST cannot get colder in winter than -1.8C, while the SAT does not get warmer than 0C in summer over ice. However, the SST can warm substantially in summer and the SAT can get much colder in winter. This by itself explains the out of phase behaviour. While the authors have addressed some of my earlier concerns in the first review, this work does not really have a clear hypothesis and answer and I do not believe it is worthy of publication at this point.

Thanks for your comments and our responses are listed below.
1. The present work pertains to a Pliocene simulation that compares well to reconstructed data, it might make a good contribution to the literature.
2. The application of CFRAM method can take individual physical processes into account and quantify the contribution of sea ice.
3. To avoid the contamination of open ocean regions, only the ice-covered regions are examined in the revised version.
4. The fact that SST cannot get colder in winter than -1.8℃, while the SAT does not get warmer than 0℃ in summer over ice can only explain the small SST change in winter and the small SAT change in summer. The complete explanation of the out of phase behavior of SST and SAT must solve the question why the maximum SST change happens in summer and the maximum SAT change happens in winter. The former is due to albedo effect of sea ice and the latter results from insulation effect of sea ice, and both are quantitatively analyzed in this study.

**Point-to-point response to reviewers' comments**
*The comments are in black, and our answers are in blue.*
**Reviewer #2:**

**Second review of "Contribution of sea-ice albedo and insulation effects to Arctic amplification in the EC-Earth Pliocene simulation" for**
*Climate of the Past.*

**Summary:**

This manuscript examines the impact of sea ice changes on the surface air and sea temperatures in the Arctic during the Pliocene, as simulated by EC-Earth. Specifically, they examine the energy flux impact of differences in surface albedo and surface ocean insulation spatially correlated with the changes in sea ice between the Pliocene and pre-industrial eras. They found that a reduction in albedo allows for much stronger short-wave heating during May, causing the biggest SST difference between the two eras to be during August. They also found that this extra stored heat was released back to the atmosphere via enhanced surface heat fluxes due to the decrease in sea ice insulation. This resulted in SAT differences between the two eras being largest during winter, and with an inverse seasonal timing relative to SST.

**Paper recommendation:**

The authors have adequately responded to my concerns, both in the main document and in the supplement, and have substantially improved the manuscript. A few very minor issues and grammatical corrections still need to be made (which are listed below), but once they have been dealt with then I believe the paper will be ready for publication. Given this, I am recommending "**Accept after technical corrections**", and look forward to seeing the final, published version of the manuscript.

We are very grateful for the positive evaluation and detailed comments that follow. Below are our responses, and we have revised the manuscript accordingly.

**Minor issues:**

1. What units are the "SIC change" values in for figures 5 and 9?

"SIC change" is unitless as SIC is the ratio of the area of sea ice to the total area at a grid point in the ocean.

2. Line 312: I would replace "bulk heat transfer coefficient" with "bulk transfer coefficients", as the coefficient for sensible heat is not necessarily the same as the coefficient for evaporation/latent heat.

We agree and revise it in the manuscript.

**Grammatical and cosmetic issues:**

Line 12: Replace "current warming climate" with "the current warming climate".
Done.

Line 30: Replace "by National Snow" with "by the National Snow".
Done.

Line 35: Replace "results from local" with "results from the local".
Done.

Line 52: Replace "sea ice effect" with "sea ice effects".
Done.

Line 59: Replace "affect climate system" with "affect the climate system".
Done.

Line 61: Replace "refer these two effects" with "refer to these two effects".
Done.

Line 64: Replace "Community Atmospheric General Circulation Model" with "Community Atmosphere Model".
Done.

Line 67: Replace "separate contribution" with "separate contributions".
Done.

Line 81: Replace "is preindustrial" with "is a preindustrial".
Done.

Line 96: Replace "In LIM3, surface" with "In LIM3, the surface".
Done.

Line 140: Replace "is based on TOA" with "is based on the TOA".
Done.

Line 144: Replace "is decomposing radiative perturbation into individual contribution" with "is decomposing the radiative perturbation into individual contributions".
Done.

Line 151: Replace "each factors" with "each factor".
Done.

Line 152: Replace "perturbation" with "perturbations".
Done.

Line 165: Replace "of sea ice" with "of the sea ice".
Done.

Line 181: Replace "of response" with "of the response".
Done.

Line 181: Replace "degree" with "degrees".
Done.

Line 192: Replace "during Pliocene" with "during the Pliocene".
Done.

Line 203: Replace "in EC-Earth" with "in the EC-Earth".

Done.

Line 246: Replace "that albedo" with "that the albedo".

Done.

Line 252: Replace "degree of freedom" with "degrees of freedom".

Done.

Line 267: Replace "with albedo effect" with "with the albedo effect".

Done.

Line 299: Replace "individual month" with "individual months".

Done.

Line 322: Replace "similar but different" with "similar, but not exactly the same".

Done.

Line 357: Replace "by annual" with "by the annual".

Done.

Line 362: Remove one of the two "indirectly" words from this sentence.

Done.

Line 364: Replace "to insulation effect" with "to the insulation effect".

Done.

Line 377: Replace "within Arctic" with "within the Arctic".

Done.

Line 550: Replace "to albedo effect" with "to the albedo effect".

Done.

Line 597: Replace "The diagonal stripe in (c)" with "The diagonal stripe in (c) and (f)".

Done.

Line 658: Replace "vapour change to" with "vapour change due to".

Done.

Line 670: Replace "to insulation effect" with "to the insulation effect".

Done.

**Point-to-point response to reviewers' comments**

*The comments are in black, and our answers are in blue.*

**Reviewer #3:**

Review of Zheng et al.

Zheng et al. has made substantial revision to their manuscript and improved the writing and the clarity of the presentation. I appreciate the authors' detailed reply to my comments. For example, the addition of the definition of "anomaly" is essential for the interpretation of Fig. 5 and Fig. 9. Some of my questions on the linearity of the attribution method and the Pliocene experiment setup are still not well explained. I suggest the authors to add an uncertainty estimate to the attribution method. The differences in boundary conditions between the preindustrial run and the Pliocene run will introduce perturbation fluxes not included in the attribution framework. The implication for current results should be considered. I recommend a substantial-minor revision of the current manuscript before it can be accepted for publication.

We are very grateful for the positive evaluation and detailed comments that follow. Below are our responses, and we have revised the manuscript accordingly.

Major concerns:

1. The linearity of CFRAM

The author mentioned the "albedo and insulation interact in a nonlinear way" in line 72-73 and then apply an attribution method with linear assumption without giving clear estimate of the role of nonlinearity. The authors suggest the linearity of the CFRAM can be examined by comparing the total radiative perturbation and the sum of all the partial radiative perturbation terms. The provided figure has very confusing titles and does not seem to be related to the "total vs partial perturbation" comparison. Even if the figure is indeed a comparison of the total vs partial perturbations, it does not show the residual well. A comparison of the residual, total minus sum of partial perturbation, with the total is more informative of degree of nonlinearity. Maybe this can also be used as a measure of the uncertainty in the linearity assumption? Please show clear evidence to support the assumption of linearity and provide an uncertainty estimate of this assumption.

The figure below shows the total radiative perturbation (a), the sum of all the partial radiative perturbation terms (b) and the residual (c) over the Arctic Ocean. It appears that the total radiative perturbation and the sum of all the partial radiative perturbation terms are almost identical. The residual can explain about 2.6% of the variance of the total radiative perturbation. Therefore the nonlinearity might be negligible and it is reasonable to linearly decompose radiative perturbation into individual contribution with CFRAM. The validation of linearity is added in the text and the Supplement (Figure S4).

[Figure]

2. The boundary condition differences between preindustrial and Pliocene runs

I would like to thank the authors for providing clearer description of the Pliocene experiment setup. After reading the authors' clarification in the revision, I realized that it is important to note the boundary condition differences between preindustrial run and the Pliocene run. The PRISM conditions are not used in the preindustrial run. This choice is understandable for model evaluation purpose using proxy data. However, it does introduce additional uncertainties in your attribution analysis. The adoption of PRISM may introduce radiative perturbation that are not included in Eq (1). How important is this factor compared to albedo and insulation effects of sea ice?

Secondly, I understand that the authors followed the standard protocol in PlioMIP2, but a brief explanation of the justification of some setup choices are beneficial to readers who are not very familiar with the PlioMIP2. For example, the choice of equilibrium runs depends on the small $CO_2$ concentration and orbital forcing changes during this period. The application of the boundary conditions from PRISM may affect various climatic variables, such as temperature and precipitation, so the influence of boundary conditions can be captured in the changes of air temperature, cloud, water vapor and other factors. That is to say, the radiative perturbations in Eq (1) contain not only the effects of sea ice but also effects of the boundary conditions. In this study we focus on the effects of sea ice, which can explain part of the variance of the partial radiative perturbations in Eq (1). The residual variance includes the contribution of the boundary conditions and other factors, and it is difficult to obtain the contribution of the boundary conditions and compare with the effects of sea ice.

Thanks for your suggestion! A brief explanation of the justification of some setup choices are added in the "experimental design" section. That is, "These two experiments proposed in PlioMIP2 core experiments may assess the dependence of climate sensitivity on the radiative forcing and the boundary conditions."

Minor concerns:

1. The Pliocene climate "anomalies"

It is very confusing to refer to the differences between the Pliocene run and the preindustrial run as "anomalies", which is usually used to describe the deviation from a climatological mean or spatial averages. I guess this is why another reviewer is asking whether the trend is removed when calculating the "anomalies". Please try to find another way to describe it or be sure to explicitly remind the readers that the "anomalies" is defined differently here.

The definition of "anomalies" has been emphasized in Section 2.1 in the revised version to remind the readers.

2. Tables.

This is just a suggestion. The tables listed these variables for all 12 months. It might be less intimidating and easier to digest the information if they are shown in plots instead, unless there are some particular numbers that are important to the readers. This is just my opinion though.

Thanks for your suggestion! We think it might be more quantitative and easier to present some features, such as the significance of correlation coefficients if tables are used.

Specific comments:

Line 29-30: Something is not right about this statement. In addition, 2005 is not in the last decade.

The sentence has been modified in the revised version.

Line 31: "ice-free Arctic Ocean" in summer or all year around?

"ice-free Arctic Ocean" in September. It's specified in the text

Line 47-48: Is this a target for this study to produce a more significant Arctic amplification?

Koenigk et al. (2013) had suggested that the EC-Earth simulations show a strong Arctic amplification compared to most CMIP3 models. It's not a target for this study to produce a more significant Arctic amplification.

Line 52: It is not clear what the "sea ice effect" is referring to.

The "effects of albedo and insulation" is specified in the revised sentence.

Line 68: Why does the sea ice thinning lead to enhanced insulation? This is confusing. Please rephrase.

The "enhanced insulation" should be "weakened insulation" and it has been corrected in the text.

Line 72-73: So nonlinearity is important. Does this contradict the choice of linearity for attribution later?

As the result mentioned above (Major concerns 1), the nonlinearity exists but is not important in this study.

Line 118-120: This statement is confusing. It gives the impression that although it is actually late Pliocene but called mid-Pliocene because the mid-Pliocene is studied more. Do you mean this term "mid-Pliocene warm period" is frequently used for this period in literature although it is actually late Pliocene? Please rephrase.

Yes, we mean this term "mid-Pliocene warm period" is frequently used for this period in literature although it is actually late Pliocene. Thanks for your suggestion! The sentence has been modified in the revised version.

Line 146: Again, does the differences in boundary conditions contribute in this equation?

The radiative perturbations in Eq (1) contain not only the effects of sea ice but also effects of the boundary conditions. In this study we focus on the effects of sea ice, which can explain part of the variance of the radiative perturbations in Eq (1), and the boundary conditions can explain part of the residual variance.

Line 147: Is the "perturbation" here refer to the differences between the Pliocene run and the preindustrial run? Please clarify.

Yes. The "perturbation" is specified in the text for clarity.

Line 208-213: As is mentioned by another reviewer, the constant SST under sea ice is also a contributing factor here. Please add it here.

Done.

Fig. 5: Because each dot represents a grid point in the model, then the latitude can be very important for the total SW insulation and the SW changes with albedo. Will you get a different regression coefficient if subsetting the data by latitude? Can you calculate similar contribution to variance for the other factors in Eq (1)? Will they add up to 1?

To find the role of latitude, the data are divided into 4 subsets by latitude and shown below. The higher the latitude, the stronger is the response of the net shortwave radiation change caused by albedo to the change of sea ice concentration.

The contribution to variance can be calculated for other factors, but it just represents the variance of each term in the right hands of Eq (1) explained by the sea ice change, e.g. 71% for SW change due to albedo, 31% for LW change due to water vapor, so they will not add up to 1.

[Figure]

Line 260: What is the "response coefficient of albedo to SIC"?
The phrase "response coefficient of albedo to SIC" is unclear, so it is changed to "response coefficients of net shortwave flux to the albedo effect of sea ice".

Line 324-327: Following Clausius–Clapeyron, in the colder Arctic, the saturating mixing ratio of water vapor is much smaller, so the latent heating and the responses are also smaller.
Thanks for your helpful comments! The explanation is added in the revised version.

Line 336: I don't think it is conduction. Heat transfer by conduction does not rely on the wind speed or the amount of turbulence.
Conduction is efficient at the sea/ice surface when the molecules are tightly constrained, while convection occurs much more efficiently in the atmosphere by mixing. The sensible heat is transferred through convection and conduction. So the "convection" is added in the text.

[revised manuscript text omitted]

---

## Author Response (AR3)

**Editor Decision: Publish subject to technical corrections (22 Jan 2019)**

**by Ran Feng**

Comments to the Author:
Line 17: "...response analysis method (CFRAM) and an approach similar to equilibrium feedback assessment..." I believe what you meant is that CFRAM is an approach similar to equilibrium feedback assessment. If so, please replace "and" with ",".

Thanks for your comments! The climate feedback and response analysis method (CFRAM) is different from the approach similar to equilibrium feedback assessment, and they are introduced in Section 2.2 and 2.3 respectively.

[revised manuscript text omitted]